# Reticular adhesions are assembled at flat clathrin lattices and opposed by active integrin α5β1

Laura Hakanpää[1,2,3], Amr Abouelezz[1,2,3], An-Sofie Lenaerts[1,2,3], Seyda Culfa[1,2,3], Michael Algie[1,2], Jenny Bärlund[1,2,3], Pekka Katajisto[1,2,3,4], Harvey McMahon[5], and Leonardo Almeida-Souza[1,2,3]

Reticular adhesions (RAs) consist of integrin αvβ5 and harbor flat clathrin lattices (FCLs), long-lasting structures with similar molecular composition as clathrin-mediated endocytosis (CME) carriers. Why FCLs and RAs colocalize is not known. Here, we show that RAs are assembled at FCLs in a process controlled by fibronectin (FN) and its receptor, integrin α5β1. We observed that cells on FN-rich matrices displayed fewer FCLs and RAs. CME machinery inhibition abolished RAs and live-cell imaging showed that RA establishment requires FCL coassembly. The inhibitory activity of FN was mediated by the activation of integrin α5β1 at Tensin1-positive fibrillar adhesions. Conventionally, endocytosis disassembles cellular adhesions by internalizing their components. Our results present a novel paradigm in the relationship between these two processes by showing that endocytic proteins can actively function in the assembly of cell adhesions. Furthermore, we show this novel adhesion assembly mechanism is coupled to cell migration via unique crosstalk between cell-matrix adhesions.

## Introduction

Integrins are nonenzymatic dimeric transmembrane receptors that recognize ECM components. These mechanosensory proteins govern cell adhesion to the ECM maintaining correct tissue development and function, with elaborate connections to cellular homeostasis and disease (Kanchanawong and Calderwood, 2023). Ligand availability and biochemical and physical properties of the ECM determine integrin activation status, integrin clustering, and, ultimately, the formation of cellular adhesion structures (Kechagia et al., 2019).

Cells can form a variety of integrin-based adhesions. Small integrin clusters engaged to the ECM, called nascent adhesions, form on the cell periphery and establish their connection to the actin cytoskeleton via adaptor proteins such as Talin. A balancing act of traction forces and signaling molecules determines whether nascent adhesions mature into larger and molecularly more complex focal adhesions (FAs; Wehrle-Haller, 2012). In migrating cells and in the presence of the ECM component fibronectin (FN), FAs can serve as platforms for the formation of fibrillar adhesions (FBs), where FN-bound α5β1 integrins are formed along actin cables as they extend FN fibrils (Georgiadou and Ivaska, 2017). Common to all these types of cell adhesions,

their disassembly is mediated by the removal of integrin molecules from adhesion sites via endocytosis (Kechagia et al., 2019).

Recently, a novel type of integrin-based cell adhesion was discovered (Lock et al., 2018). Called reticular adhesions (RAs), these structures contain integrin αvβ5, lack the typical markers for the other adhesion types, such as Talin1 or Paxillin, and are not connected to actin stress fibers. RAs can occupy a significant portion of the substrate-facing surface of cells in culture and can significantly outlast FAs. Their physiological function is, however, not clear.

Intriguingly, RAs colocalize with large, persistent forms of clathrin structures at the cell membrane called flat clathrin lattices (FCLs; also referred to as clathrin plaques; Grove et al., 2014). The structure containing FCLs and RAs is called clathrin-containing adhesion complexes (CCAC; Lock et al., 2019; Zuidema et al., 2020). FCLs were previously considered stalled endocytic events of the clathrin-mediated endocytosis (CME) pathway. However, recent studies have changed this view and support the idea that FCLs are signaling platforms (Leyton-Puig et al., 2017; Grove et al., 2014; Alfonzo-Méndez et al., 2022). In vivo, FCLs localize to adhesive structures between bone and

[1]Helsinki Institute of Life Science, University of Helsinki, Helsinki, Finland; [2]Institute of Biotechnology, University of Helsinki, Helsinki, Finland; [3]Faculty of Biological and Environmental Sciences, University of Helsinki, Helsinki, Finland; [4]Department of Cell and Molecular Biology, Karolinska Institutet, Stockholm, Sweden; [5]MRC Laboratory of Molecular Biology, Cambridge, UK.

Correspondence to Leonardo Almeida-Souza: leonardo.almeida-souza@helsinki.fi

M. Algie's current affiliation is The New Zealand Institute for Plant and Food Research Ltd., Nelson Research Centre, Nelson, New Zealand.



osteoclasts (Akisaka et al., 2008) and are required for the organization of sarcomeres (Vassilopoulos et al., 2014).

The functional relationship between FCLs and RAs is not clear. A confounding factor in this relationship lies in the fact that although FCLs always localize to RAs, the opposite is not true. RAs can occur as large structures with FCLs covering only a fraction of their area. Moreover, integrin αvβ5 can localize to both RAs and FAs. Although details on the factors mediating integrin αvβ5 localization to FCLs are becoming clearer (Zuidema et al., 2018; Zuidema et al., 2022), why these structures co-exist, what their function is, and how cells control their formation remain a mystery.

In this study, we show that FCLs are required for the establishment of RAs. Moreover, we found that an FN-rich ECM acts as an inhibitor of RA formation. This inhibitory role of FN is mediated by the activation of integrin α5β1 localized at fibrillar adhesions. Furthermore, we show that the transition from a static to a migratory state is mirrored by the disappearance of FCLs and RAs.

## Results

### FN inhibits the formation of FCLs

While studying CME dynamics, we serendipitously observed that cells on FN appear to display fewer FCLs when compared with cells plated on non-coated glass dishes. To confirm if ECM proteins in general can influence CME, we assessed the effect of several major ECM components as well as non-ECM coatings and non-coated surfaces on the amount of FCLs. For that, dishes were coated for 16–24 h, after which cells were let to attach for 16–20 h in serum-containing medium before imaging. For quantifications, we established the metric FCL proportion, which defines the average fraction of FCLs per frame among all clathrin-coated structures detected in a 5-min movie (see Materials and methods for details). These experiments were performed using U2Os cells with an endogenously GFP-tagged α adaptin 2 Sigma-Aldrich subunit (AP2S1, hereafter referred to simply as AP2). AP2 is a widely used CME marker that faithfully mirrors clathrin dynamics (Almeida-Souza et al., 2018; Ehrlich et al., 2004; Rappoport and Simon, 2008). We used endogenously tagged cell lines throughout this study as the expression level of the AP2 complex was shown to modulate the amount of FCLs (Dambournet et al., 2018).

U2Os cells on non-coated dishes presented typical and abundant FCLs (i.e., bright, long-lived AP2-GFP marked structures; Fig. 1, A and B; Fig. S1 A; and Video 1), similar to what has been found in many cell lines (Zuidema et al., 2022; Moulay et al., 2020; Sochacki et al., 2021; Saffarian et al., 2009). Similarly, U2Os cells plated on dishes coated with the non-ECM proteins BSA and poly-L-lysine (PLL) also presented high FCL proportions (Fig. 1 B and Fig. S1 A; and Video 1). Out of the major ECM proteins tested, FN, collagen IV (Col IV), and laminin-111 (LN111) reduced FCL proportion significantly. The integrin αvβ5 ligand vitronectin (VTN) did not increase or decrease the FCL proportion when compared to non-coated dishes (Fig. 1 B and Fig. S1 A; see Discussion). Similarly, and in line with a recent study (Baschieri et al., 2018), collagen I (Col I) did not reduce

FCLs (Fig. 1 B, Fig. S1 A, and Video 1). Different concentrations of FN used for coating (10 or 20 µg/ml) did not show significant differences (Fig. 1 B).

Recently, it was described that SCC-9 cells produce more FN when plated on Col IV or LN111 (Lu et al., 2020). To probe if this is also the case for our cells, we stained FN from U2Os cells plated directly onto non-coated dishes or plated on FN, VTN, Col IV, Col I, or LN111. While U2Os cells produce little FN overnight, cells plated on Col IV produced a striking amount of FN, which assembled into elongated fibrils (Fig. 1, C and D). LN111 coating also induced FN production, but less strikingly than Col IV. Col I and VTN coating were unable to stimulate FN production (Fig. 1, C and D). These results suggest that FN is the main ECM component inhibiting FCL formation.

For many cell lines, it is common to find considerable variability in the amount of FCLs in culture. We thus decided to test if this variability is due to differential FN production within the culture. Confirming this hypothesis (and bearing in mind that U2Os secrete FN modestly; see below), we found that cells plated on non-coated dishes displaying fewer FCLs were predominantly lying on top of an FN-rich region of the culture (Fig. S1 B).

Next, we asked if the reduction in FCL proportions observed in FN-coated samples is a cell-wide effect or specific to cellular regions in direct contact with the extracellular substrate. For that, we used patterned dishes containing FN-coated regions interspersed with uncoated regions, where single U2Os-AP2-GFP cells could adhere simultaneously to both an FN- and a non-coated region. In line with a contact-dependent effect, low FCL proportions were observed in cellular regions in contact with FN whereas FCL proportion was high in cellular regions contacting non-coated surfaces (Fig. 1, E–G, and Video 2).

The abundance of an alternative splice variant of the clathrin heavy chain containing exon 31 was recently shown to increase the frequency of clathrin plaques in myotubes (Moulay et al., 2020). We thus tested if the effects we observe are also due to changes in clathrin splicing but found no differences when comparing cells plated on non-coated or FN-coated dishes (Fig. S1 C). Thus, these results show that FN is a potent inhibitor of FCLs. Moreover, FN inhibits FCLs in a contact-dependent manner locally within a single cell (Fig. 1 H).

### FN inhibits the formation of RAs in a similar manner as FCLs

As discussed in the introduction, FCLs localize to RAs. To check how ECM composition affects these structures, U2Os AP2-GFP cells plated on FN, VTN, Col IV, Col I, or LN111 and stained with the RA component integrin αvβ5 and—to be able to distinguish integrin αvβ5 on RAs or FAs—were also stained with an FA marker (p-Pax Y118; phosphorylated paxillin). Cells plated overnight without coating formed abundant RAs (Fig. 2, A and B). On FN-coated dishes, big RAs were largely absent but small "dot-like" nascent RAs were present in a few cells. Similarly, on Col IV and LN111 coatings (which stimulated FN production; Fig. 1 C), cells formed significantly fewer RAs than on non-coated dishes (Fig. 2, A and B). Coating with VTN, the ligand for integrin αvβ5 present at RAs and FAs alike (Fig. S1 C; Lock et al., 2018), did not result in more RAs (Fig. 2, A and B). Different coatings also changed the total amount of integrin αvβ5

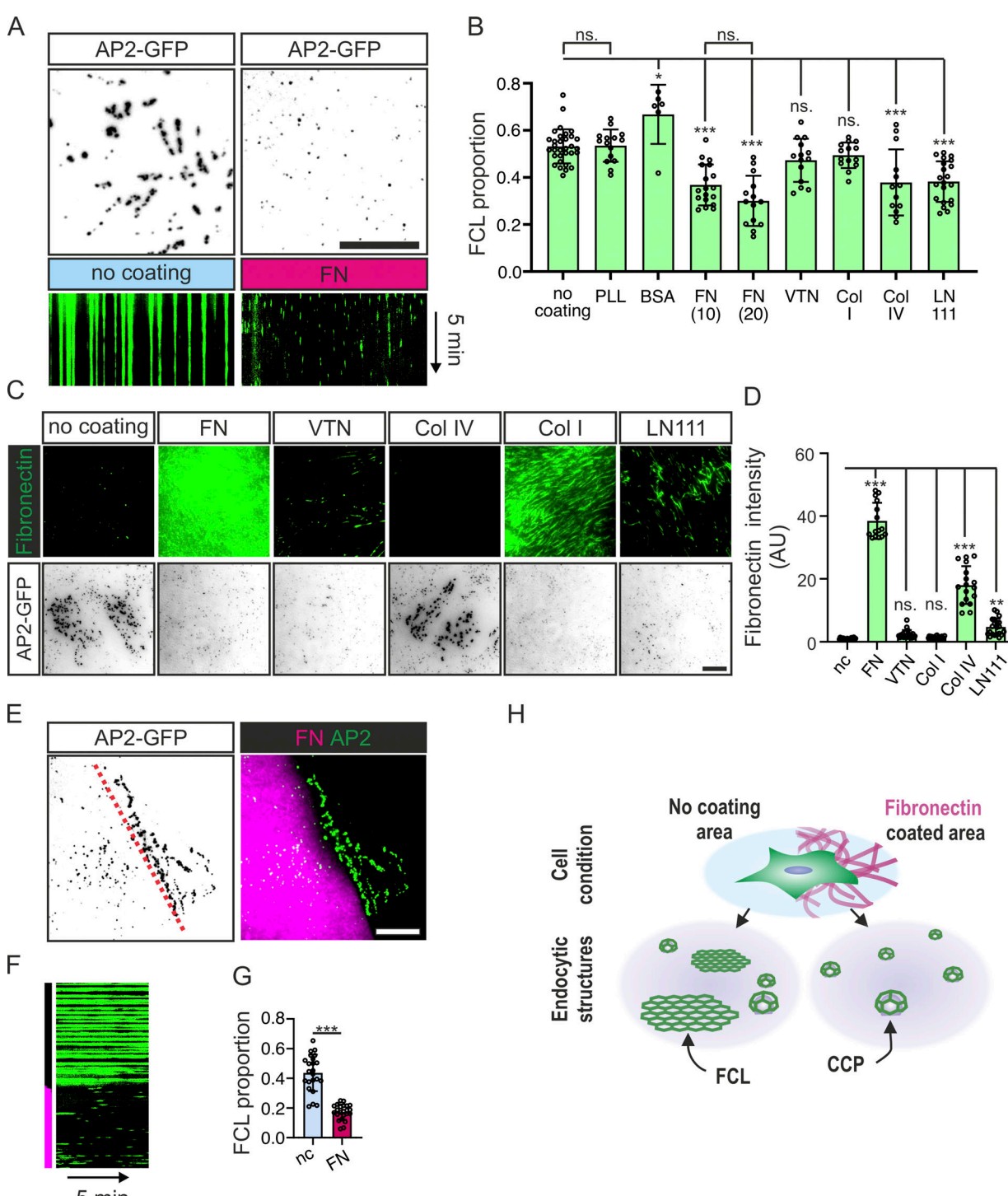

Figure 1. **FN inhibits FCL formation in a local manner. (A)** U2Os-AP2-GFP cells plated on non-coated or FN-coated dishes were imaged using total internal reflection fluorescence (TIRF) microscopy at 1-s intervals for 5 min. Images represent 15-s time projections, and kymographs represent the 5-min time-lapse videos. **(B)** FCL proportions of U2Os-AP2-GFP cells plated on PLL-, BSA-, FN-, VTN-, Col I-, Col IV-, and LN111-coated or non-coated dishes and imaged using TIRF microscopy, at 1-s intervals for 5 min. $N$ (videos): non-coated (nc) = 32, PLL 3µg/ml = 6, BSA 20 µg/ml = 15, FN 10 µg/ml = 18, FN 20 µg/ml = 14, VTN 20 µg/ml = 14, Col I 20 µg/ml = 14, Col IV 20 µg/ml = 14, LN111 10 µg/ml = 22. Videos were collected from four independent experiments, except for PLL, from two experiments. One-way ANOVA with Tukey's multiple comparison test. $F_{(8, 183)} = 19.11$, $P < 0.0001$. **(C)** U2Os-AP2-GFP cells plated on FN, VTN, Col I, Col IV, LN111 (all 10 µg/ml) -coated or non-coated dishes and stained for FN. Representative TIRF images. **(D)** Quantification of FN fluorescent intensity from samples in C. $N$ (images): FN = 15 images, VTN/Col I/LN111/non-coated = 21 images; Col IV = 17 images. Results were obtained from one representative experiment. Similar results were observed in four individual experiments. $F_{(5, 110)} = 338.5$, $P < 0.0001$. **(E)** U2Os-AP2-GFP cells were plated on FN/glass patterned imaging dishes and imaged with TIRF microscopy at 1-s intervals for 5 min. Representative images are 15-s time projections. **(F)** Representative 5-min kymograph of time-lapse videos in E. **(G)** Quantification of FCL proportions in E. $n$ = 23 videos, collected from four individual experiments. Two-tailed Student's $t$ test, $P < 0.001$. **(H)** A schematic illustration of the results shown in this figure. Data are the mean ± SD, ns. non-significant P value; * P value < 0.05; P value < 0.01; *** P value < 0.001. Scale bars, 10 µm.

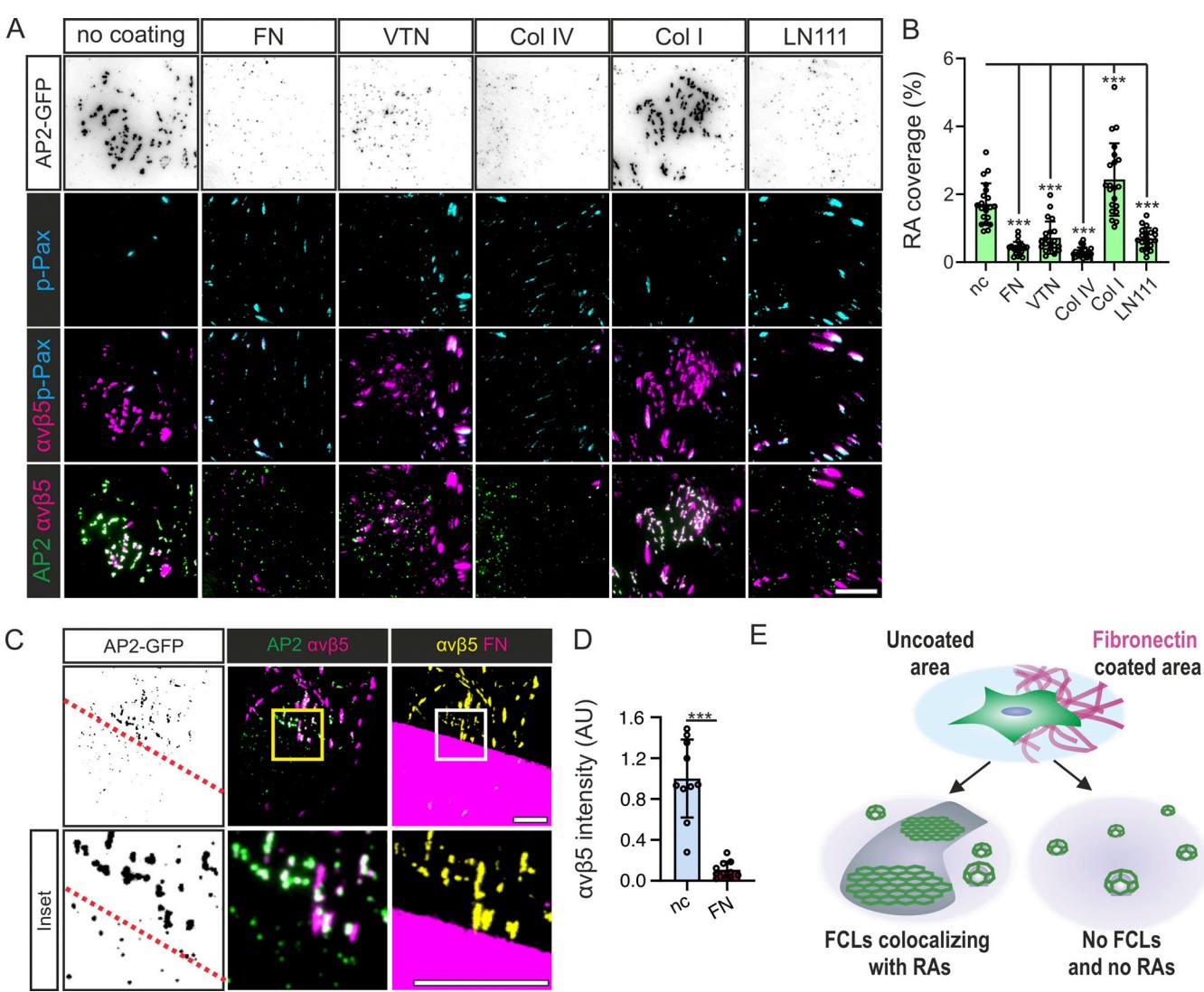

Figure 2. **FN inhibits RA formation in a local manner. (A)** U2Os-AP2-GFP cells plated on FN, VTN, Col I, Col IV, LN111 (all 10 µg/ml) -coated or non-coated dishes, stained for p-Pax and integrin αvβ. Representative TIRF images. **(B)** Analysis of RA coverage from samples in A. $N$ = 21 from two independent experiments. $F(5, 120)$ = 48.05, P < 0.0001. **(C)** U2Os-AP2-GFP cells were plated on FN/glass patterned imaging dishes overnight, stained for integrin αvβ5 and p-Pax, and imaged using TIRF microscopy; representative TIRF images. **(D)** Quantification of integrin αvβ5 fluorescent intensity on FN- and glass-side of the pattern; $n$ = 10 from one representative experiment. Similar results were observed in five similar experiments. Two-tailed Student's $t$ test, P < 0.01. **(E)** A schematic illustration of the results shown in this figure. Data are the mean ± SD, ns. non-significant P value; *** P value < 0.001. Scale bars, 10 µm; insets, 5 µm.

on the bottom surface of cells (Fig. S1 D). However, they did not follow a clear relationship with the amount of RAs. To quantify differences in RA amounts in cells, we developed a metric called RA coverage, which measures the fraction of the area of the cell covered by integrin αvβ5 signal (excluding FAs). RA coverage serves as a good metric to distinguish between large and nascent RAs and, crucially, it shows a clear correspondence between RA content and both FCL proportion and FN abundance in the ECM (see Fig. 1, B–D, and Fig. 2 B).

Next, we used our substrate patterning strategy to check if the local FN effects on FCLs were also similar for RAs. Strikingly, cells plated on patterned FN revealed that RAs, akin to FCLs, were completely inhibited on cellular regions in contact with FN. Cellular regions in contact with non-coated surfaces

displayed many FCLs colocalizing with RAs while regions in contact with FN presented no RAs or FCLs (Fig. 2 C). Interestingly, in these patterned substrates, most of the integrin αvβ5 signal segregated to non-coated regions forming typical RAs (Fig. 2, C and D). This contrasts with cells plated in fully coated dishes (Fig. 2 A), where integrin αvβ5 can be seen in both RAs and FAs. Hence, the inhibitory effects of FN on FCLs affect RAs in a similar manner (Fig. 2 E).

**The effect of FN on FCLs and RAs is clear in various cell lines**
Next, we checked if the effects we see in U2Os cells are also true for other cell lines. To avoid problems of overexpression, we endogenously tagged AP2 with either Halo tag or GFP in various human cell lines: HeLa (epithelial, cervical carcinoma), MCF7

(epithelial, breast cancer), HDF (dermal fibroblast, noncancerous), Caco2 (epithelial, colon carcinoma), and hMEC (human mammary epithelial cells). These cell lines presented a large variation in the amount of FCLs and the morphology of RAs. Importantly, these cells could be divided into two groups in terms of endogenous FN secretion, and this division clearly correlated with the amount of FCLs and RAs (Fig. 3, A and B). U2Os, HeLa, and MCF7 composed the group of low-FN-secretion cells. U2Os form large RAs on non-coated dishes, whereas HeLa formed multiple dot-like nascent RAs (which colocalized with FCLs), with bigger RAs found more seldom (Fig. 3 A). MCF7 cells formed many FCLs and large RAs covering almost the entire cell area (Fig. 3 A). None of the high FN-secreting cell lines (HDF, Caco2, and hMEC) formed large RAs (Fig. 3, A and B). In these high FN-producing cells, small FCL/RA dots were often found in areas with less deposited FN (Fig. 3 A).

We next evaluated the response of these cell lines to FN precoating. In low-FN-producing cell lines (U2Os, HeLa, and MCF7), RA coverage dropped significantly (Fig. 3, C–E). Among the high-FN-production cell lines, only Caco2 reduced its RA coverage on FN-coated dishes (Fig. 3, C and D). As expected, HDF and hMEC, which had low RA coverage without coating, did not show a significant response to FN coating (Fig. 3, C–E).

To evaluate if changes in the amount of RAs are reflected in the amount of FAs, we measure FA coverage (i.e., the cellular area covered by FAs) from U2Os, Hela, and MCF7. We did not observe any clear difference in the coverage of FAs for these three cell lines plated on uncoated or FN-coated dishes (Fig. S2 A). These results suggest that the dynamics of integrin αvβ5 in FAs and RAs are controlled independently.

For all experiments so far, we used media supplemented with serum, which is known to contain ECM components, including FN. Given that our cells are left to attach overnight in this media, it would be reasonable to expect that the FN present in serum would coat the dishes and completely mask our results. To test why this does not seem to happen (Fig. 1 C and Fig. 3 A), we compared the amount of FN deposited on the glass surface in different conditions: dishes were coated for 24 h with 10, 5, and 1 µg/ml of FN (diluted in PBS), 100% FBS, media with 10% FBS, and PBS as a control. After coating, U2Os cells were plated and left to attach for 16 h before being fixed and stained for FN. Surprisingly, our results revealed that very little FN was deposited on glass in dishes "coated" with full media or pure FBS (Fig. S1, F and G). These results are in line with similar experiments performed 30 years ago (Steele et al., 1992). We hypothesize that this phenomenon occurs due to the high concentrations of BSA in serum (40 mg/ml), which rapidly saturates the surface of culture dishes, thereby acting as a blocking agent for the binding of serum FN.

Taken together, the presence of FN controls the formation of RAs and FCLs in a very similar manner and in different cell lines (Fig. 3 F), suggesting a common and general mechanism for the establishment of these structures.

**The CME machinery is essential for RA formation**
Next, we set out to dissect the relationship between the formation of FCLs and RAs. It has been shown that integrin αvβ5 is required for the establishment of FCLs (Baschieri et al., 2018; Zuidema et al., 2018). We confirmed this observation by silencing integrin β5 from U2Os AP2-GFP cells plated on non-coated dishes and, indeed, they displayed a significantly lower FCL proportion compared with control cells (Fig. S2, B and C). Further, while integrin β5-silenced cells were unable to form RAs, they did form FAs (Fig. S2, D and E). The requirement of integrin β5 for FCL formation was further confirmed using Cilengitide, the inhibitor for integrin αvβ5 (Desgrosellier and Cheresh, 2010), as the treatment led to a rapid disassembly of FCLs and RAs (Fig. S2, F and G).

While all FCLs colocalize to RAs, FCL-free areas of larger RAs are rather common (e.g., Fig. 2 A; Fig. 3, A–C; and Fig. S2 D), which may give the impression that FCLs are formed on preexisting RAs. Nevertheless, the fact that both structures are inhibited independently by FN suggests a deeper relationship and led us to ask if RAs can exist without the CME machinery. To answer this question, we quantified the RA coverage in U2Os-AP2-GFP cells silenced for the clathrin adaptor AP2 complex subunits α1 (AP2A1) or Sigma-Aldrich 1 (AP2S1) in cells plated on non-coated dishes, a condition where we observe large RAs. Consistent with an important role played by the CME machinery in RA formation, AP2A1- or AP2S1-silenced cells (easily recognizable as cells with little to no AP2-GFP signal) did not display RAs. Instead, integrin αvβ5 localized to FAs (Fig. 4, A and B).

To confirm these results, we expressed the AP180 C-terminal fragment (AP180ct), which acts as a strong dominant negative of CME (Ford et al., 2001). AP180ct-positive U2Os-AP2-GFP cells plated on non-coated dishes displayed low AP2 signal at the membrane and, akin to AP2-silenced cells, RAs were largely absent with integrin αvβ5 localized to FAs, whereas AP180ct-negative cells displayed typical FCLs and RAs (Fig. 4, C and D). Thus, the CME machinery is required for the formation of RAs (Fig. 4 E).

Next, we set out to visualize the dynamics of AP2 during RA formation. For that, we generated a double U2Os knock-in cell line AP2-GFP and integrin β5 (ITGB5)-mScarlet. RAs are remarkably stable structures (Lock et al., 2018) and their de novo formation is rare, making it rather difficult to capture such events. To minimize this challenge, we optimized the conditions for Cilengitide treatment to disassemble RAs followed by a washout, when RAs could start reforming (Fig. S3, A and B). Using these washout conditions, we were able to capture events showing that the formation and growth of ITGB5-positive structures are accompanied by the formation of FCLs (Fig. 5 A; Fig. S3, C and D; and Videos 3 and 4). Also, in events where we could not clearly detect the extension of a mature RA (reminiscent of the dot-like structures we see in many cells), we noticed that the establishment of an FCL was typically accompanied by an increase in ITGB5 fluorescence (Fig. 5 B; Fig. S3, C and D; and Video 3). Importantly, ITGB5-positive structures, which did not colocalize with an FCL, rapidly disappeared. In many cases, this disappearance was preceded by bona fide CME events (short lived AP2-GFP signals), likely representing CME-mediated ITGB5-cluster disassembly (Fig. 5 B and Fig. S3 D).

Taken together, our results show that the relationship between FCLs and RAs is beyond a simple colocalization. In fact,

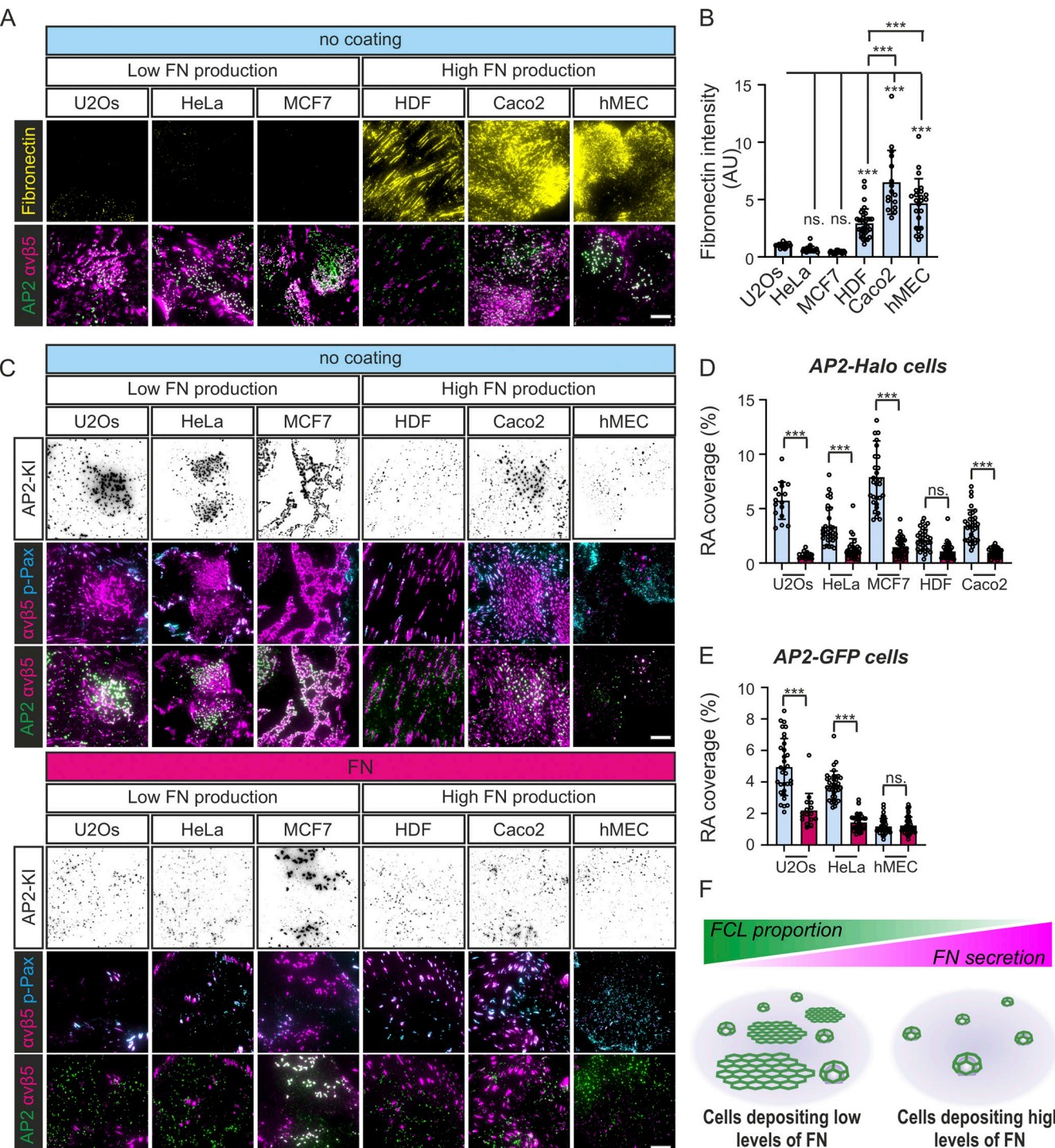

Figure 3. **FCLs and RAs presence correlates with FN production in multiple cells lines. (A–E)** The following knock-in cell lines were used in this figure: U2Os-AP2-GFP, HeLa-AP2-GFP, hMEC-AP2-GFP, U2Os-AP2-halo, HeLa-AP2-halo, MCF7-AP2-halo, HDF-AP2-halo, and Caco2-AP2-halo. **(A)** Cell lines indicated were plated to non-coated (nc) dishes, allowed to settle overnight, and stained for integrin αvβ5 and FN. Representative TIRF images. **(B)** Analysis of FN fluorescent intensity from samples in A. *N* (images): U2Os = 21, HeLa = 20, MCF7 = 23, HDF = 33, Caco2 = 16, hMEC = 23, from two independent experiments. $F_{(5, 130)}$ = 56.16, P < 0.0001. **(C)** Cells were plated on FN-coated or nc dishes, allowed to settle overnight, and stained for integrin αvβ5 and p-Pax. Representative TIRF images. **(D)** Analysis of RA coverage from samples in C. Cell lines with an AP2 halo tag, *n* (images): U2Os (nc/FN) = 20, HeLa (nc) = 33, HeLa (FN) = 29, MCF7 (nc) = 29, MCF7 (FN) 32, HDF (nc/FN) 32, Caco2 (nc/FN) = 32. Data were obtained from two individual experiments and similar results were observed in four individual experiments. $F_{(9, 273)}$ = 64.96, P < 0.0001. **(E)** Analysis of RA coverage from samples in C. Cell lines with an AP2 GFP tag, *n* (images): U2Os (nc) = 31, U2Os (FN) = 18, HeLa (nc) = 35, HeLa (FN) = 38, and hMEC (nc/FN) = 38. Data were obtained from two individual experiments and similar results were observed in four individual experiments. $F_{(5, 189)}$ = 89.97, P < 0.0001. **(F)** A schematic illustration of the results shown in this figure. Data are the mean ± SD, ns. non-significant P value; *** P value < 0.001. Scale bars, 10 μm.

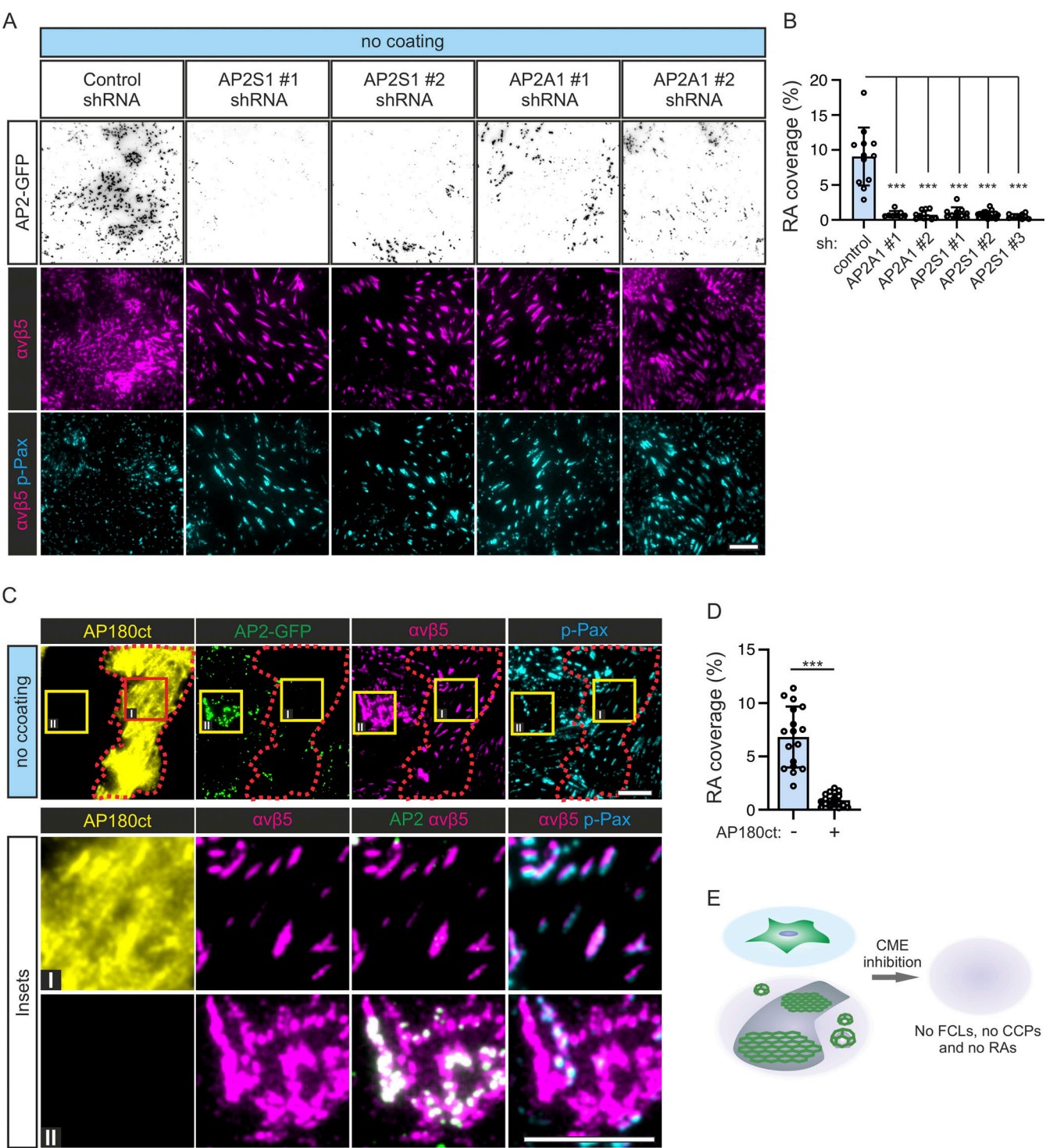

Figure 4. **Inhibition of CME prevents RA formation. (A)** U2Os-AP2-GFP cells silenced for AP2A1, AP2S1, or control (scrambled shRNA) were plated on non-coated dishes and stained for integrin αvβ5 and p-Pax. Representative TIRF images. **(B)** Analysis of RA coverage from samples in A. *N* (images): control = 12, shAP2A1#1 = 8, shAPA1#2 = 11, shAPS1#1 = 10, shAPS1#2 = 19, shAPS1#3 = 10. Data were obtained from two individual experiments and similar results were observed in five independent experiments. One-way ANOVA with Tukey's multiple comparison, $F(5, 64) = 43.11$, P < 0.001. **(C)** U2Os-AP2-GFP cells over-expressing Ap180 ct were plated on non-coated dishes and stained for integrin αvβ5 and p-Pax. Representative TIRF images. **(D)** Analysis of RA coverage from samples in C; *n* = 17 from one representative experiment and similar results were observed in four individual experiments. Two-tailed Student's *t* test, P < 0.0001. **(E)** A schematic illustration of the results shown in this figure. Data are the mean ± SD, *** P value < 0.001. Scale bars, 10 μm; insets, 5 μm.

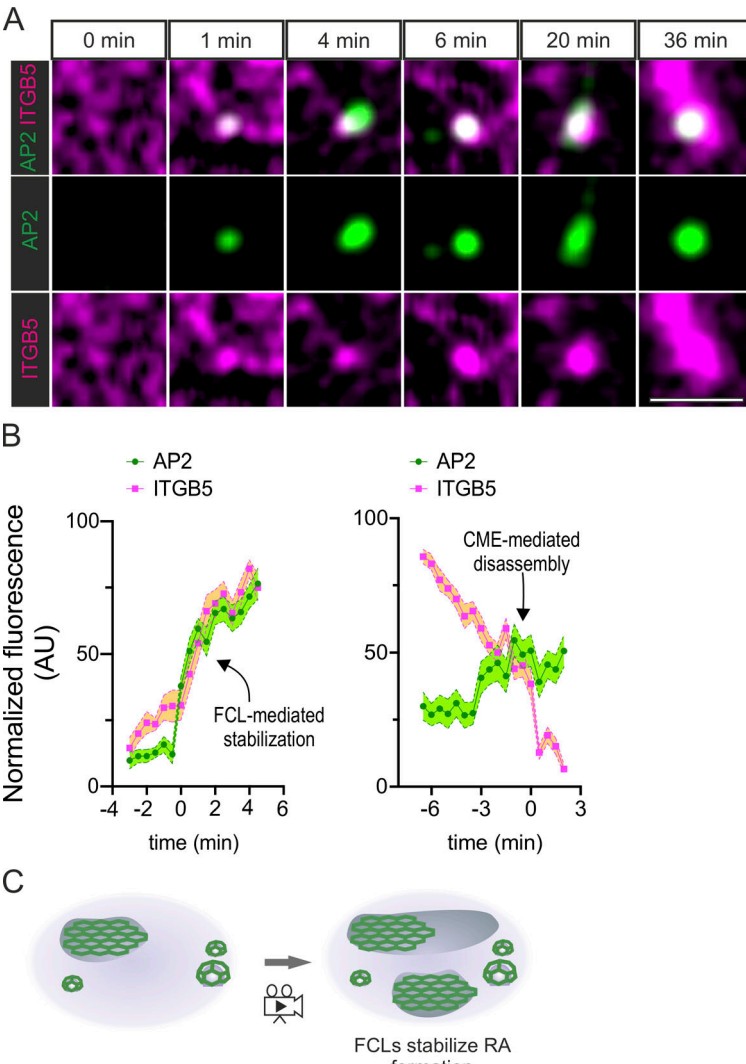

**Figure 5.** **RAs are formed at FCLs. (A)** U2Os-AP2-ITGB5-mScarlet cells plated on glass were treated with Cilengitide (10 µM) for 15 min, washed twice with fresh medium, and imaged live using TIRF microscopy at two frames per minute. Representative frames showing the growth of an RA from an FCL. **(B)** Analysis of AP2 and integrin αvβ5 (ITGB5) intensity over time. Left: FCL and αvβ5 clustering during RA assembly (n = 16 events). In this graph, time zero is defined as the arrival of the AP2 signal Right: ITGB5 clusters not stabilized by FCLs disassemble and are removed from the membrane by CME-mediated disassembly (n = 24 events). In this graph, time zero is defined as the disappearance of the ITGB5 signal. Events have been collected from four 1-h long time-lapse acquisitions. Mean ± SEM. **(C)** A schematic illustration of the results is shown in this figure. Scale bar, 5 µm.

our data reveal strict codependency, where FCLs are required for the stabilization and growth of integrin αvβ5 clusters, thereby establishing RAs, and vice versa, with integrin αvβ5 being required for the formation of FCLs (Fig. 5 C and Fig. S2).

**The inhibitory effect of FN on FCL and RA formation is mediated by integrin α5β1**
To understand the mechanism controlling the coassembly of FCLs and RAs, we turned our attention back to FN. While integrin αvβ5 binds to VTN at FAs and RAs, the major FN receptor is integrin α5β1 (Humphries et al., 2006). First, we acutely interfered with integrin β1 binding to FN using the function-blocking antibody mab13. U2Os-AP2-GFP cells seeded on FN-coated dishes were treated with mab13 and monitored for the acute formation of FCLs and RAs. Over the time course of 45 min, mab13 induced the relocalization of integrin αvβ5 from FAs to small, newly formed RAs (Fig. 6, A and B). Further supporting the role of FCLs in RA assembly, these newly formed RAs completely colocalized with FCLs (bright AP2 signals; Fig. 6, A–C). A similar experiment followed by live-cell imaging confirmed these results and showed a gradual increase in FCL proportions after mab13 treatment (Fig. 6 D). Mab13 treatment

had no effect on existing FCLs and RAs formed on non-coated dishes (data not shown).

In line with these results, integrin β1 silencing in U2Os-AP2-GFP cells plated on FN resulted in a high FCL proportion and large, prominent RAs (Fig. 7, A–D, and Fig. S4, A–C). Despite the striking increase of integrin αvβ5 on the bottom surface of silenced cells, this increase was not reflected in expression levels, indicating that the stimulation of RA formation leads to a change in the trafficking of this integrin dimer (Fig. S4 C).

A significant increase in RAs was also seen in cells silenced for integrin α5, the α subunit which pairs with integrin β1 for FN binding (Fig. 7, C–E, and Fig. S4, D and E). Taken together, these results show that the inhibitory activity of FN on RAs and FCLs occurs via the activation of integrin α5β1 (Fig. 7 F).

**Activation of integrin α5β1 at FBs controls RA and FCL formation**
When bound to FN, integrin α5β1 translocates centripetally from FAs to form elongated structures called FBs along actin stress fibers. This movement generates long FN fibrils in a process called FN fibrillogenesis and is mediated by the cytoskeleton scaffolding protein Tensin1 (Pankov et al., 2000). To determine

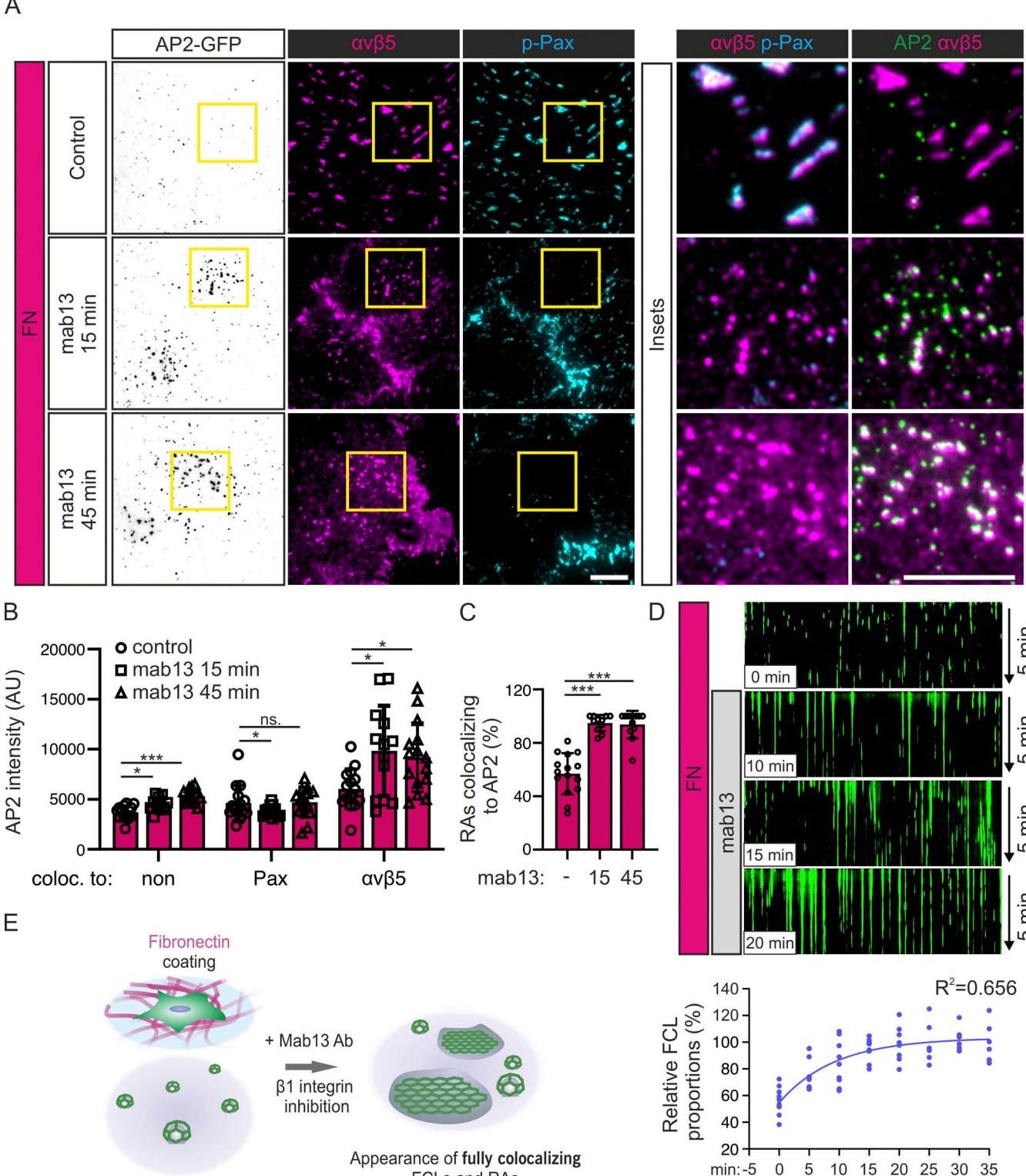

Figure 6. **Integrin β1 blocking stimulates FCL and RA formation. (A)** U2OS-AP2-GFP cells plated on FN were treated with integrin β1 blocking antibody mab13 (0.3 µg/ml) for 15 and 45 min (or vehicle for 45 min = control) and stained for integrin αvβ5 and p-Pax. Representative TIRF images. **(B)** Analysis of AP2 signal colocalizing with no markers (non), p-Pax, or integrin αvβ5 over time. $N$ (images): control = 18, mab13 15 min = 13, mab13 45 min = 15, from one representative experiment. Similar results were observed in four independent experiments. One-way ANOVA with Tukey's multiple comparison. $F_{(2, 119)}$ = 46.68, P < 0.0001. **(C)** Analysis of fraction of integrin αvβ5 colocalizing with AP2 over time from samples in A. $N$ (images): control = 15, mab13 15 min = 11, mab13 45 min = 12, from one representative experiment. Similar results were observed in four independent experiments. One-way ANOVA with Tukey's multiple comparison. $F_{(2, 35)}$ = 45.81, P < 0.0001. **(D)** U2OS-AP2-GFP cells plated on FN were treated with mab13 (0.3 µg/ml) and imaged using TIRF microscopy. 5-min time-lapses with 1-s intervals starting at 0 min (no mab13) and every 5 min after mab13 addition, until 35 min, were acquired. $N$ (videos): 0 min

= 11, 5 min = 11, 10 min = 9, 15 min = 8, 20 min = 8, 25 min = 6, 30 min = 7, 35 min = 6. Videos were acquired from two independent experiments. Similar results were observed in five individual experiments. One-way ANOVA with Tukey's multiple comparison. $F(7, 29) = 8.893$, $P < 0.0001$. **(E)** A schematic illustration of results shown in this figure. Data are the mean ± SD, ns. non-significant P value; * P value<0.05, *** P value < 0.001. Scale bars, 10 µm; insets, 5 µm.

which type of adhesion structure active integrin α5β1 localizes to under our experimental conditions, we plated U2Os-AP2-GFP cells on FN and non-coated dishes and stained them with an active integrin β1-specific antibody (12G10) and Tensin1 or p-Pax to mark FBs or FAs, respectively. The staining revealed that in the FN-coated dishes, active integrin β1 was colocalizing with FBs (Tensin1; Fig. 8 A). As expected, active integrin β1 and Tensin1-positive adhesions were largely absent in non-coated dishes (Fig. 8 A).

Next, to determine which active integrin β1 pool is more important for the inhibition of FCLs and RAs, we silenced FAs and FB components on U2Os-AP2-GFP cells and plated them on FN. In accordance with the higher accumulation of active integrin β1 in FBs, silencing of Tensin1 led to a marked increase of RAs and FCLs accompanied by a reduction in the presence of active integrin β1 on the membrane (evidence by 12G10 antibody staining; Fig. 8, B and C; and Fig. S5, A–C). Silencing of the FA component Talin-1 also led to increased RAs and FCLs and a reduction of active integrin β1 on the membrane (Fig. S5, D–F). Given the strong phenotype on Tensin1 knockdown, this result was expected as FAs are precursors of FBs.

FB formation indicates activated and migratory cell phenotypes. Indeed, active sliding of integrin α5β1 and Tensin1 bound to FN along central actin stress fibers increases traction forces (Georgiadou and Ivaska, 2017; Pankov et al., 2000) and is required in cell migration during development and cancer metastasis (Efthymiou et al., 2020; Schwarzbauer and DeSimone, 2011). If the extension of active integrin α5β1 into FBs is indeed required for the inhibition of FCLs and RAs, we hypothesized that physical confinement of cells—which inhibits cell migration—would also inhibit the sliding of FB from FAs. In turn, the absence of integrin α5β1 in FBs would favor FCLs and RAs, even if cells were plated on an FN-rich matrix. To test this possibility, we turned to single-cell micropatterns. In contrast to the patterned coatings we used in Figs. 1 and 2, these micropatterns do not allow cells to attach outside the defined areas on a coverslip. Given the small size of these areas (1,100 µm²), cells are laterally confined. U2Os-AP2-GFP cells were plated on slides with arrow- and H-shaped micropatterns either precoated with FN or not and stained for integrin αvβ5 and p-Pax and imaged to measure RA coverage. In addition, to measure integrin β1 activation, U2Os-AP2-GFP-ITGB5-mScarlet cells were plated similarly and stained for active integrin β1. Supporting our hypothesis, we could detect clear FCL and RAs in FN-coated micropatterns (Fig. 9, A and B; and Fig. S5 K). On arrows, FCLs and RAs developed on the shaft of the pattern rather than the arched area. In the H-patterns, FCLs and RA developed all over the pattern. Cells on non-coated patterns made large FCLs and RAs often extending throughout the pattern (Fig. 9, A and B). Crucially, the RA coverage was not significantly different between coated or non-coated patterns (Fig. 9 B). As expected, staining with active integrin β1 (12G10)

showed a clear difference in signal between FN-coated and non-coated patterns (Fig. 9 C and Fig. S5 L). Importantly, further supporting a need for FB formation to inhibit FCLs and RAs, 12G10 signal was not organized as elongated, central FBs but rather confined at the cell periphery (Fig. S5 K). Thus, the inhibitory role of FN on FCLs and RA formation occurs primarily via the activation of integrin α5β1 on Tensin1-positive fibrillar adhesions.

### The disassembly of FCLs and RAs is coupled to cell migration

As physical restriction favored FCLs and RAs, we wondered if inducing migration will have the opposite effect. To test this hypothesis, we monitored FCLs and RAs in a classic wound healing assay. U2Os-AP2-GFP-ITGB5-mScarlet cells were plated on non-coated dishes and allowed to grow to full confluency for 2 d. Cultures were then wounded and cells were allowed to migrate. At 0 min (i.e., just after wounding), FCLs and RAs were abundant and equally distributed at the edge and away from the wound (Fig. 9 D). Within 80 min, the cells at the migration front had lost most of their FCLs and RAs, while cells further away from the edge maintained their FCLs and RAs (Fig. 9 D). At 4 h, as the migratory front grew larger, the loss of FCL and RAs also extended away from the wound (Fig. 9 D). In full accordance with the results we presented above, the disappearance of FCLs and RAs was preceded by the increase in FN secretion by the cells at the edge of the wound (Fig. 9, E and F). Together, these results place the resolution of FCLs and RAs as an intrinsic part of the cascade of events triggering cell migration (Fig. 9 G).

## Discussion

The extracellular environment is a key regulator of cellular physiology with integrins playing a key role translating the chemical composition of the extracellular milieu into intracellular signals. Among various mechanisms controlling integrin function, integrin trafficking via endocytosis and exocytosis plays a major role (Moreno-Layseca et al., 2019). Thus far, the relationship between integrin-based matrix adhesions and endocytosis has been considered primarily antagonistic, with endocytosis playing a role in the disassembly of said adhesive structures (Ezratty et al., 2009; Moreno-Layseca et al., 2019). Here, we provide evidence, for the first time, of a constructive relationship between the endocytic machinery and cellular adhesions, where the CME machinery, in the form of FCLs, is key for the formation of integrin αvβ5 RAs. Moreover, we show that RA formation is counteracted by the activation of a distinct integrin heterodimer, α5β1, in distinct adhesion structures, FBs, revealing an interesting mechanism of interadhesion crosstalk.

Our results support the idea that FCLs and RAs are two sides of the same structure (Lock et al., 2019; Zuidema et al., 2020). Previous studies have demonstrated the importance of integrin αvβ5 at RAs in the formation of FCLs (Zuidema et al., 2018; Zuidema et al., 2022; Lock et al., 2019). Here, we show that this

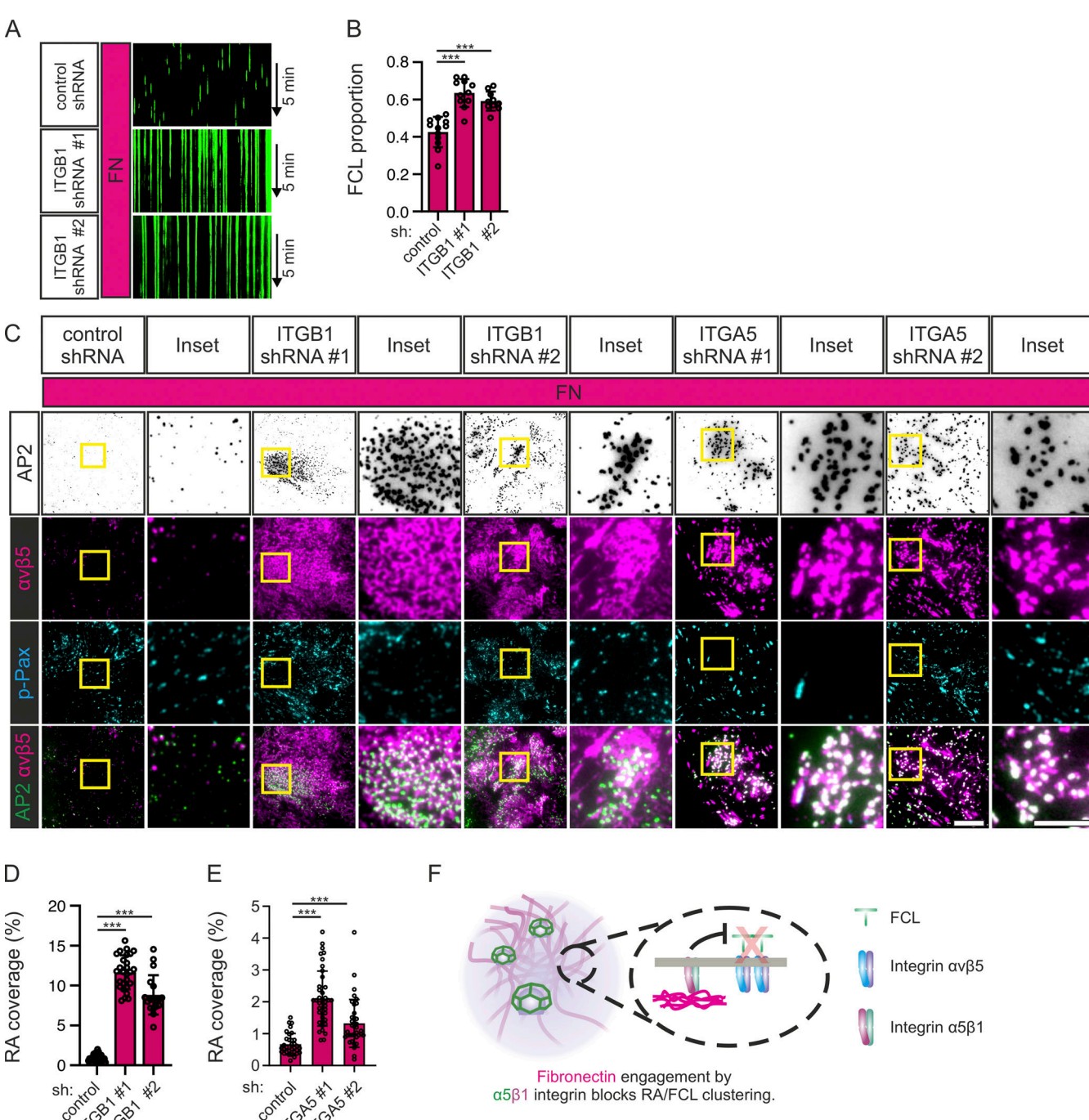

Figure 7. **Depletion of integrin β1 or Integrin α5 promotes FCL and RA formation. (A)** U2Os-AP2-GFP cells silenced for integrin β1 with two different shRNAs (shITGB1 #1, #2) or control shRNA were plated on FN-coated dishes and stained for p-Pax and imaged using TIRF microscopy at 1-s intervals for 5 min. Representative 5-min kymographs. **(B)** Analysis of FCL proportions from samples in A. $N$ (videos): shScr = 12, shITGB1 #1 = 11, shITGB1 #2 = 10, from three individual experiments. Tukey's multiple comparison, $F(2, 30) = 27.81$, P < 0.001. **(C)** U2Os-AP2-GFP cells silenced for integrin β1 with two different shRNAs (shITGB #1, #2), or integrin α5 with two different shRNAs (shITGA5 #1, #2), or control shRNA, were plated on FN-coated dishes and stained for integrin αvβ5 and p-Pax. Representative TIRF images. **(D)** Analysis of RA coverage from integrin β1 silenced samples in C. $N$ (images): control shRNA = 30, shITGB1 #1 = 27, shITGB1 #2 = 18, from two independent experiments; similar results were observed in four independent experiments. One-way ANOVA with Tukey's multiple comparison, $F(2, 72) = 276.2$, P < 0.0001. **(E)** Analysis of RA coverage from integrin α5 silenced samples in E. $N$ (images): control shRNA = 33, shITGA5 #1 = 40, shITGA5 #2 = 37, from three individual experiments. One-way ANOVA with Tukey's multiple comparison, $F(2, 107) = 44.46$, P < 0.0001. **(F)** A schematic illustration of the results shown in this figure. Data are the mean ± SD, *** P-value < 0.001. Scale bars, 10 μm; insets, 5 μm.

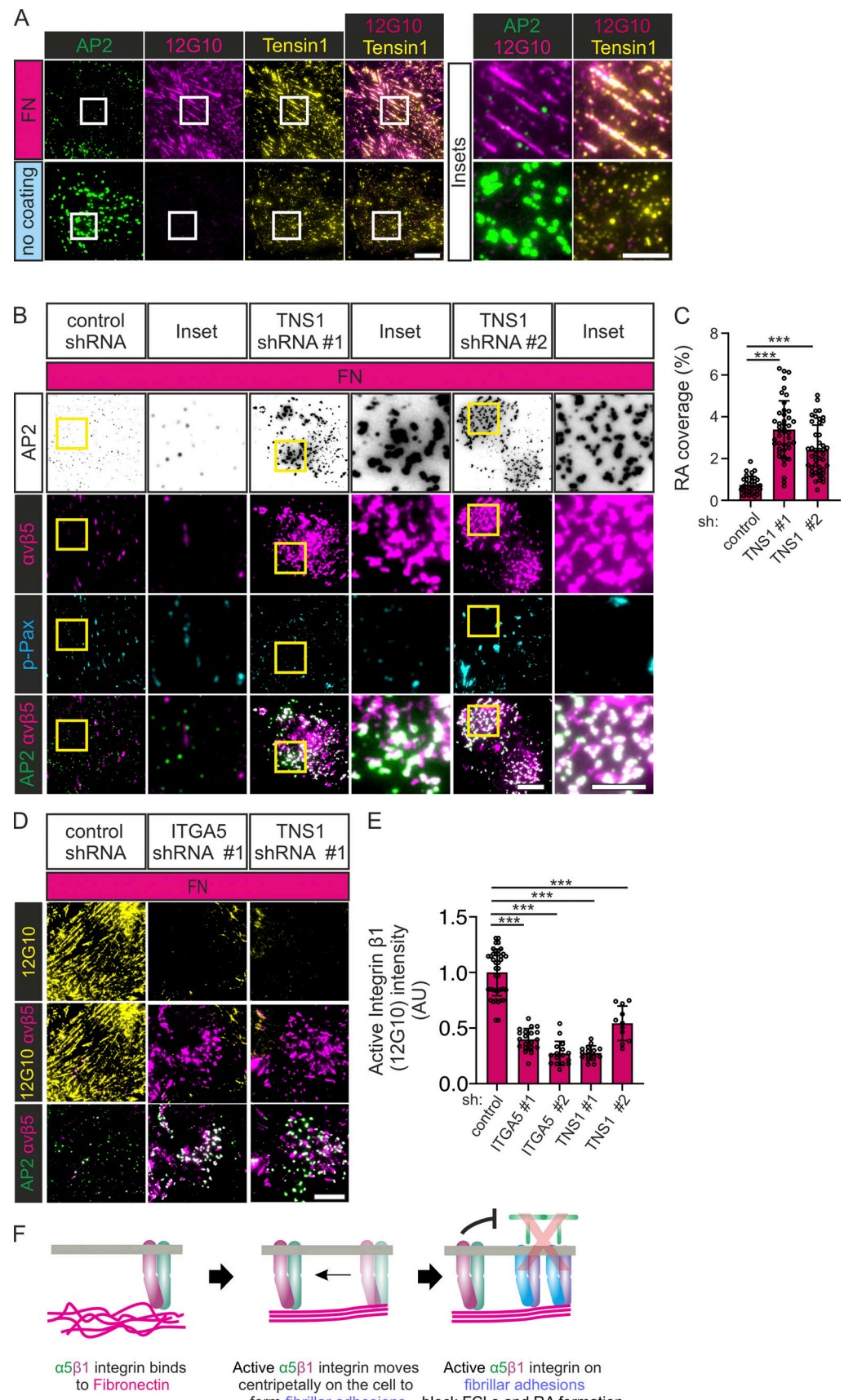

Figure 8. **Active integrin α5β1 at fibrillar adhesions inhibit FCL and RA formation. (A)** U2Os-AP2-halo cells were plated to FN-coated or non-coated dishes and stained for Tensin1 and active integrin β1 12G10. Representative TIRF images. **(B)** U2Os-AP2-GFP-ITGB5-mScarlet cells silenced for Tensin1 with

two different shRNAs (shTNS1 #1, #2) or control shRNA were plated on FN-coated dishes and stained for p-Pax. Representative TIRF images. **(C)** Analysis of RA coverage from samples in B. *N* (images): shScr control = 35, shTNS1 #1/#2 = 44, from three independent experiments. One-way ANOVA with Tukey's multiple comparison, $F_{(2, 120)}$ = 56.26, P < 0.0001. **(D)** U2Os-AP2-GFP-ITGB5-mScarlet cells silenced for Tensin1 with two different shRNAs (shTNS1 #1, #2) or control shRNA were plated on FN-coated dishes and stained for active integrin β1 (12G10 antibody). Representative TIRF images. **(E)** Analysis of 12G10 fluorescent intensity from samples in D. *N* (images): shScr = 20, shTNS1 #1 = 15, shTNS1 #2 = 11, from one representative image. Similar results were observed in three individual experiments. One-way ANOVA with Tukey's multiple comparison, $F_{(2, 43)}$ = 87.81, P < 0.0001. **(F)** A schematic illustration of the results shown in this figure. Data are the mean ± SD, *** P value < 0.001. Scale bars, 10 μm; insets, 5 μm.

---

relationship is also crucial in the other direction, with FCLs being required for the formation of integrin αvβ5 RAs. Therefore, we believe the previously suggested term clathrin containing adhesion complexes (or CCAC for short) is a more appropriate terminology to refer to these structures.

### The mechanism of RA formation by FCL and αvβ5 co-assembly

We observed that RA formation events are rare, which led us to use non-physiological conditions—a Cilengitide washout experiment—to detect them. Therefore, the physiological trigger leading to the formation of FCLs and establishment of RAs remains to be understood. VTN, the ligand for integrin αvβ5, could be considered a good candidate. However, as we show in Fig. 2 E and as reported by others (Zuidema et al., 2022), integrin αvβ5 binds VTN equally on FAs and RAs. While it is clear that the presence of VTN is important as an extracellular tether for the formation of integrin αvβ5 adhesions (Zuidema et al., 2022; Zuidema et al., 2018; Lock et al., 2018), the switch between these adhesion types is likely an inside-out mechanism. We did not detect an increase in integrin αvβ5 RAs on VTN-coated dishes (Fig. 1 B). This could seem counterintuitive, but VTN, which was initially called a "serum spreading factor" (Hayman et al., 1983), is readily secreted by cells during attachment. Therefore, we advise caution when making conclusions based on the results of non-coated and VTN-coated dishes on the role of this ECM component on RA coverage. Further work is necessary to shed light on this issue.

Recent evidence showed that EGFR activation led to the enlargement of FCLs in an integrin β5 phosphorylation-dependent manner (Alfonzo-Méndez et al., 2022), pointing to a possible mechanism for the initial coassembly of FCLs and RAs. This possibility is further reinforced by the fact that the relationship between growth factor receptors and integrins has been established in multiple contexts (Ivaska and Heino, 2011).

Another key unknown aspect of RA formation by FCL and αvβ5 coassembly concerns how these structures can be molecularly differentiated from canonical endocytic events. The connection between integrin αvβ5 located in RAs and FCLs occurs primarily via the endocytic adaptors ARH and NUMB (Zuidema et al., 2018). Importantly, these adaptors also participate in integrin endocytosis (Ezratty et al., 2009; Nishimura and Kaibuchi, 2007), suggesting that other mechanisms may be required to define the identity of FCLs.

Recently, a correlation was found between the presence of clathrin plaques and an alternatively spliced isoform of clathrin containing exon 31 in myotubes (Moulay et al., 2020). We did not detect any changes in clathrin splicing in our experimental system, which was not surprising given the effects we see are contact-dependent and could not be explained by transcriptional changes. In addition, we cannot ensure that the clathrin plaques detected in myotubes are equivalent to the FCLs we observe here. Nonetheless, it is possible that the abundance of the exon 31-positive clathrin isoform works as a dial that changes the probability, speed, or efficiency by which cells form FCLs.

### Another unusual function of the clathrin machinery

In addition to its endocytic function, the clathrin machinery has been shown to participate in other processes. For example, clathrin helps to stabilize the mitotic spindle by binding to microtubules (Royle, 2012) and, during *E. coli* infection, the CME machinery is co-opted to form a clathrin-based actin-rich adhesive structure for the bacteria called pedestal (Veiga et al., 2007). Furthermore, a clathrin/AP2 tubular lattice was recently described to envelop collagen fibers during cell migration (Elkhatib et al., 2017). The results we present here add to this list of non-endocytic functions of CME components with an important twist. FCLs can also be disassembled into individual endocytic events (Lampe et al., 2016; Tagiltsev et al., 2021; Maupin and Pollard, 1983), providing an elegant and efficient mechanism for cells to switch the same machinery from an adhesion assembly to an adhesion disassembly function.

### Inhibition of CCAC and its relationship to cell migration

In addition to defining FCLs as key factors in the establishment of CCACs, our work has also revealed many interesting aspects of the inhibition and disassembly of these structures. We show that activation of integrin α5β1 by FN and the capacity of this integrin heterodimer to form FBs are both essential for the inhibition and disassembly of CCAC. In a classical wound healing assay, we observed that as cells start to migrate they secrete FN, leading to the disappearance of CCACs. However, using laterally confined cells that cannot form FBs, we observed that the mere presence of FN is not enough to inhibit CCACs. A recent study showed that high levels of activated myosin light chain (p-MLC) correlated with integrin αvβ5 localizing to FAs over RAs (Zuidema et al., 2022). Moreover, overexpression of a constitutively active RhoA mutant in a cell line with low p-MLC levels promoted integrin αvβ5 localization to FAs (Zuidema et al. 2022). Interestingly, expression of the very same RhoA mutant leads to increased FN secretion (Danen et al., 2002). As Integrin α5β1-mediated FN fibrillogenesis is required for optimal activation of the RhoA-MLC pathway, which in turn increases actin stress fiber-based migration along FBs (Gagné et al., 2020; Huveneers et al., 2008; Danen et al., 2002), these findings perfectly complement our data. Together, these results suggest that the disappearance of CCACs requires FN activation of integrin α5β1 followed by the activation of actin contractility during the cell migration process. Whether these two factors (FN and

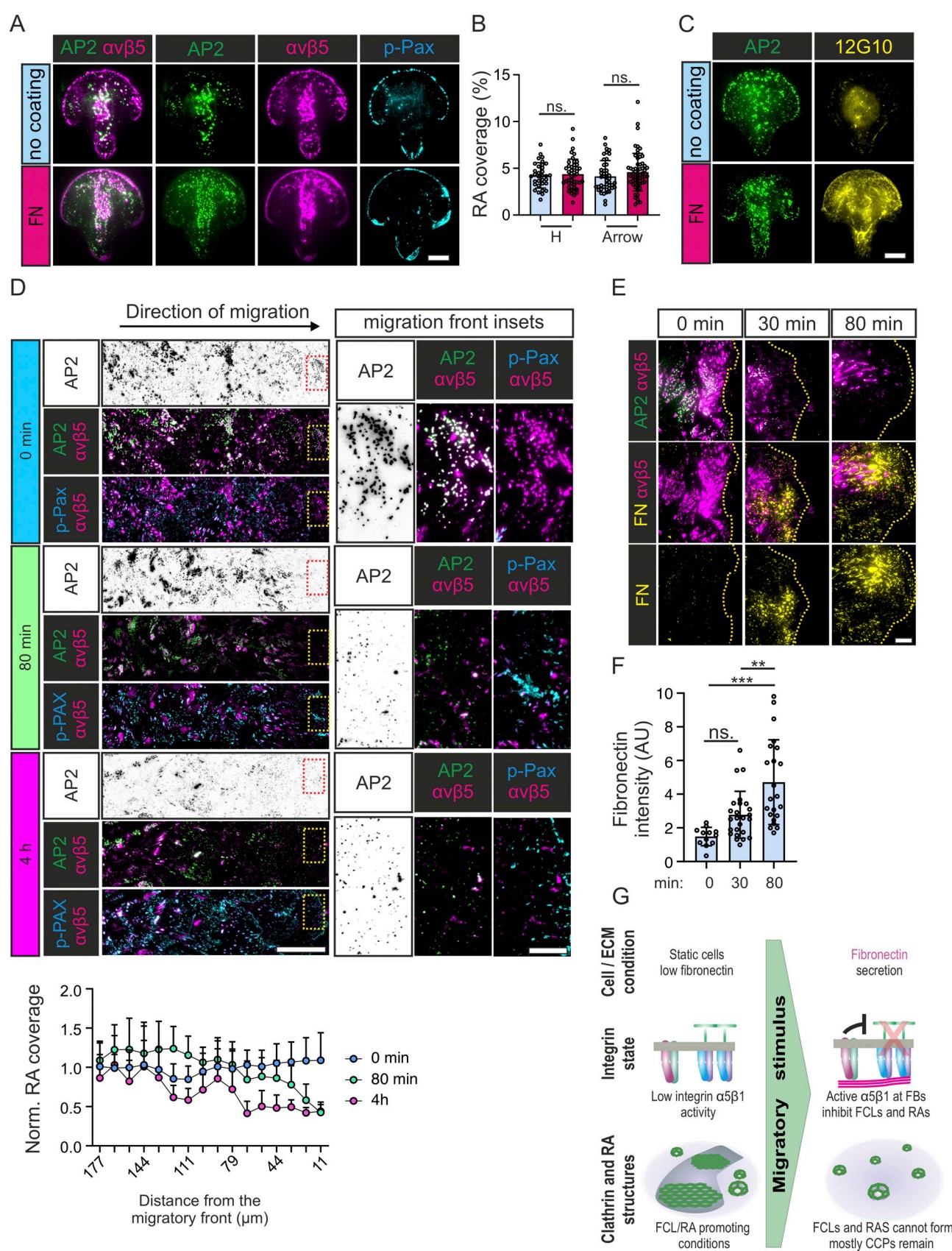

Figure 9. **RA disassembly is coupled to cell migration. (A)** U2OS-AP2-GFP cells were grown on FN-precoated or non-coated micropatterns (1,100 mm²) and stained for integrin αvβ5 and p-Pax. Representative TIRF images. See Fig. S5 K for the same staining of H-shaped patterns. **(B)** Analysis of RA coverage from samples in A. *N* (images): Arrow non-coated = 28, Arrow FN = 27, H non-coated = 25, H FN = 20 from one representative image. One-way ANOVA with Tukey's

multiple comparison, $F_{(2, 79)}$ = 38.09, P < 0.0001. **(C)** U2Os-AP2-GFP-ITGB5-mScarlet cells were grown on FN-coated or non-coated micropatterns (1,100 mm²) and stained for active integrin β1 12G10. Representative TIRF images. See Fig. S5 L for the same staining with H-shaped patterns. **(D)** Top: U2Os-AP2-GFP-ITGB5-mScarlet cells were grown on non-coated dishes for 2 d, wounded, and let to migrate for 80 min or 4 h in fresh complete medium, and stained for p-Pax. Representative TIRF images (stitched tile of five side-by-side fields of view). The wound is exactly at the right edge of the images. Bottom: Analysis of normalized RA coverage from tiles in D. RA coverage was calculated in 11-µm-wide sliding windows from the wound edge. N (tiles): 0 min = 48, 80 min = 36, 4 h = 15. One-way ANOVA with Tukey's multiple comparison, $F_{(2, 79)}$ = 38.09, P < 0.0001. **(E)** U2Os-AP2-GFP-ITGB5-mScarlet cells were grown on non-coated dishes for 2 d, wounded, and let to migrate for 30 min or 80 min in fresh complete medium, and stained for FN. Representative TIRF images from the migrating front (indicated as yellow lines). **(F)** Analysis of FN fluorescent intensity from samples in E. N (images): 0 min = 10, 30 min = 25, 80 min = 22. One-way ANOVA with Tukey's multiple comparison, $F_{(2, 56)}$ = 13.77, P < 0.0001. **(G)** A schematic illustration of results shown in this figure. Data are the mean ± SD, ** P value < 0.01, *** P value < 0.001. Scale bars, 10 µm, except in D (50 µm) and inset (10 µm).

contractility) are always required for CCACs disassembly will require further investigation. Similarly, whether the disassembly of CCACs occurs actively or is a mere consequence of a non-permissive environment for the de novo formation of new adhesions is still unknown.

Given the fact that RAs are long-lasting cellular adhesion structures, it is tempting to hypothesize that these structures act as a "parking brake" for a cell. As the cell is triggered to migrate, this brake needs to be released for efficient cell movement. This process would be analogous to the loss of cell–cell contacts, which happens during epithelial to mesenchymal transition (Kalluri and Weinberg, 2009), but instead of happening between cells, it would happen between the cell and the ECM. Therefore, we propose that disassembly of RAs is an intrinsic process during cell migration.

We showed that FN regulates CCAC assembly in all the cell lines we tested. However, how these in vitro findings will operate in vivo is still unknown. Even though the ECM composition in tissues is complex, the FN effect on CCAC formation is local and strictly contact dependent, which opens the possibility that, in vivo, tissues may use focal changes in ECM composition to control these structures.

## Materials and methods
### Cell culture and reagents
U2Os, U2Os-AP2-GFP, U2Os-AP2-halo, U2Os-AP2-GFP-ITGB5-mScarlet, HeLa-AP2-GFP, and HeLa-AP2-halo were cultured in MEM supplemented with 10% FBS (Gibco) and penicillin–streptomycin (100 U/ml, Thermo Fisher Scientific). HDF-AP2-halo and Caco2-AP2-halo were cultured in DMEM supplemented with 10% FBS (Gibco) and penicillin–streptomycin (100 U/ml, Thermo Fisher Scientific). MCF7-AP2-halo was cultured in DMEM supplemented with 10% FBS (Gibco) and penicillin–streptomycin (100 U/ml, Thermo Fisher Scientific), 2 mM glutamine, 5 µg/ml human insulin, and 1 µM sodium pyruvate. hMEC-AP2-GFP was cultured in MEGM complete medium (Lonza).

The following primary antibodies were used: anti-human integrin β1 clones 12G10 (NB100-63255; Novus bio), mAb13 (552828; BD), total integrin β1 (MAB2252; Millipore), anti-human integrin αvβ5 clone 15F11 (MAB2019Z; Millipore), anti-human integrin α5 clone SNAKA51 (AF1846; R&D), anti-human Tensin1 (SAB4200283; Sigma-Aldrich), anti-human p-paxillin Y118 (69363; Cell Signaling), anti-human-Talin1 (T3287; Sigma-Aldrich), anti-α-tubulin (sc-32293; SantaCruz Biotechnology),

and anti-GAPDH (G9545; Sigma-Aldrich). Corresponding secondary antibodies raised against rabbit or mouse IgG were purchased from Jackson Immunoresearch.

### Coating of imaging dishes
To compare the effects of major ECM proteins on AP2 lifetimes, the glass coverslip areas (14 mm diameter) of imaging dishes (Mattek) were precoated with 10 or 20 µg/ml (300 µl in PBS) of the following ECM proteins: recombinant human FN (341631; Merck), recombinant human VTN (140-09; PeproTech), Col IV (Santa cruz, se-29010), Col I (C3867; Sigma-Aldrich), and LN111 (#LN111; Biolamina). Coatings were incubated overnight at +37°C, except Col I, which was incubated at RT overnight, and LN111 was incubated at +4°C overnight. Alternatively, as non-ECM protein controls, 1% BSA (A34785; Thermo Fisher Scientific) or 0.3% PLL (152690; MP Biochemicals) were used to coat the dishes overnight at +37°C. Throughout this study, the standard FN coating was always performed similarly, 10 µg/ml (300 µl in PBS), overnight at +37°C.

### FN patterning
To study local vs. global effects of FN, FN was mixed with 50 ng/ml of Alexa647-labeled BSA and used to precoat the imaging dishes overnight at +37°C. The coated surface was subsequently scratched with a needle to allow partial reappearance of non-coated surface. After scratching, the dishes were heavily rinsed with PBS. 20,000 U2Os-AP2-GFP cells were seeded on patterned imaging dishes to ensure sufficient single-cell attachment to border areas.

### Overexpression of mammalian proteins
The clathrin inhibitor AP180 C-terminal fragment (AP180 ct; amino acids 516–898) cDNA, from rat origin, was described previously (Ford et al., 2001). This construct was cloned into Gateway-compatible pCI vectors containing an N-terminal monomeric EGFP using the Gateway system.

Transient transfections were carried out with PEI MAX transfection reagent (24765-1; Polysciences) using 70% confluent U2Os cells.

### Genetic engineering of cell lines
#### Generating the U2Os-AP2-GFP and hMEC-AP2-GFP cell lines
Three gRNA sequences (Integrated DNA Technologies) were designed using the Welcome Sanger Institute Genome online editing tool (https://wge.stemcell.sanger.ac.uk//). gRNAs were cloned into pSpCas9(BB)-2A-Puro (PX459) V2.0 (gift from. Feng

Zhang, #62988; Addgene) using BbsI sites and confirmed by Sanger sequencing.

gRNAs were transfected with the donor template for homologous recombination and the most effective gRNA (5′-TGC TACAGTCCCTGGAGTGA-3′), judged by the percentage of fluorescent cells by FACS, was used for single-clone selection, genotyping, and confirmation by microscopy.

The donor template sequence was: 5′-GGCCAGCATCCTGGG GGGCCTCGTCTCACCCCAGGGTCTCCCCTCACACAGGTTTAC ACGGTCGTGGACGAGATGTTCCTGGCTGGCGAAATCCGAGAG ACCAGCCAGACGAAGGTGCTGAAACAGCTGCTGATGCTACAG TCCCTGGAGGGAAGTGCATCTGGGAGCTCAGGCGCTAGTGGT TCAGCGAGCGGGGTGAGCAAGGGCGAGGAGCTGTTCACCGGG GTGGTGCCCATCCTGGTCGAGCTGGACGGCGACGTAAACGGC CACAAGTTCAGCGTGTCCGGCGAGGGCGAGGGCGATGCCACC TACGGCAAGCTGACCCTGAAGTTCATCTGCACCACCGGCAAG CTGCCCGTGCCCTGGCCCACCCTCGTGACCACCCTGACCTAC GGCGTGCAGTGCTTCAGCCGCTACCCCGACCACATGAAGCAG CACGACTTCTTCAAGTCCGCCATGCCCGAAGGCTACGTCCAG GAGCGCACCATCTTCTTCAAGGACGACGGCAACTACAAGACC CGCGCCGAGGTGAAGTTCGAGGGCGACACCCTGGTGAACCGC ATCGAGCTGAAGGGCATCGACTTCAAGGAGGACGGCAACAT CCTGGGGCACAAGCTGGAGTACAACTACAACAGCCACAACGTC TATATCATGGCCGACAAGCAGAAGAACGGCATCAAGGTGAAC TTCAAGATCCGCCACAACATCGAGGACGGCAGCGTGCAGCT CGCCGACCACTACCAGCAGAACACCCCCATCGGCGACGGCCCC GTGCTGCTGCCCGACAACCACTACCTGAGCACCCAGTCCAAG CTGAGCAAAGACCCCAACGAGAAGCGCGATCACATGGTCCTG CTGGAGTTCGTGACCGCCGCCGGGATCACTCTCGGCATGGAC GAGCTGTACAAGTGAGGGCAGGCGAGCCCCACCCCGGCCCCGGCC CCTCCTGGACTCGCCTGCTCGCTTCCCCTTCCCAGGCCCGTG GCCAACCCAGCAGTCCTTCCCTCAGCTGCCTAGGAGGAAGGG ACCCAGCTGGGTCTGGGCCACAAGGGAGGAGACTGC-3′, where C-terminal tagging with GFP is in green (codon-optimized) and short linker in purple. 150-bp homology arms (orange) were incorporated via PCR amplification from a synthesized (IDT), codon-optimized monomeric EGFP.

PCR product was purified and 150 ng was used directly for transfection together with gRNAs. 70–80% confluent 24-well plates of U2Os cells were transfected with 2 µg PEI (1 µg/ml), 150 ng of plasmid, and 150 ng of the PCR product. In addition, hMEC-AP2-GFP were treated with 1 µM DNA-PKc inhibitor NU7441 for 48 h after transfection. 2 d after transfection, cells were treated with puromycin (1 µg/ml) to enrich for successfully transfected cells. After expansion, GFP-positive cells were sorted by FACS and single clones were expanded and genotyped.

### Generating the U2Os-AP2-halo and HeLa-AP2-halo cell lines
U2Os-AP2-halo and HeLa-AP2-halo cell lines were generated with the same protocol as the U2Os-AP2-GFP cell line.

The donor template sequence was: 5′-*GGCCAGCATCCTGGG GGGCCTCGTCTCACCCCAGGGTCTCCCCTCACACAGGTTTACACG GTCGTGGACGAGATGTTCCTGGCTGGCGAAATCCGAGAGACCAGC CAGACGAAGGTGCTGAAACAGCTGCTGATGCTACAGTCCCTG GAG***GGAAGTGCATCTGGGAGCTCAGGCGCTAGTGGTTCAGCG AGCGGG**GCAGAAATCGGTACTGGCTTTCCATTCGACCCCCAT TATGTGGAAGTCCTGGGCGAGCGCATGCACTACGTCGATGTT

GGTCCGCGCGATGGCACCCCTGTGCTGTTCCTGCACGGTAAC CCGACCTCCTCCTACGTGTGGCGCAACATCATCCCGCATGTT GCACCGACCCATCGCTGCATTGCTCCAGACCTGATCGGTATG GGCAAATCCGACAAACCAGACCTGGGTTATTTCTTCGACGAC CACGTCCGCTTCATGGATGCCTTCATCGAAGCCCTGGGTCTG GAAGAGGTCGTCCTGGTCATTCACGACTGGGGCTCCGCTCTG GGTTTCCACTGGGCCAAGCGCAATCCAGAGCGCGTCAAAGGT ATTGCATTTATGGAGTTCATCCGCCCTATCCCGACCTGGGAC GAATGGCCAGAATTTGCCCGCGAGACCTTCCAGGCCTTCCGC ACCACCGACGTCGGCCGCAAGCTGATCATCGATCAGAACGTT TTTATCGAGGGTACGCTGCCGATGGGTGTCGTCCGCCCGCTG ACTGAAGTCGAGATGGACCATTACCGCGAGCCGTTCCTGAAT CCTGTTGACCGCGAGCCACTGTGGCGCTTCCCAAACGAGCTG CCAATCGCCGGTGAGCCAGCGAACATCGTCGCGCTGGTCGAA GAATACATGGACTGGCTGCACCAGTCCCCTGTCCCGAAGCTG CTGTTCTGGGGCACCCCAGGCGTTCTGATCCCACCGGCCGAA GCCGCTCGCCTGGCCAAAAGCCTGCCTAACTGCAAGGCTGTG GACATCGGCCCGGGTCTGAATCTGCTGCAAGAAGACAACCCG GACCTGATCGGCAGCGAGATCGCGCGCTGGCTGTCGACGCTC GAGATTTCCGGCTGAGGGCAGGCGAGCCCCACCCCGGCCCCGGC CCCTCCTGGACTCGCCTGCTCGCTTCCCCTTCCCAGGCCCGTG GCCAACCCAGCAGTCCTTCCCTCAGCTGCCTAGGAGGAAGGGACC CAGCTGGGTCTGGGCCACAAGGGAGGAGACTGC*-3′, where C-terminal tagging with halo is underlined (codon-optimized) and flexible linker region (GSASGSSGASGSASG) is bold. 150-bp homology arms (italic) were incorporated via PCR amplification from a synthesized (IDT), codon-optimized monomeric halo tag.

### Generating the HeLa-AP2-GFP, MCF7-AP2-halo, HDF-AP2-halo, and CAco2-AP2-halo cell lines
The most effective gRNA (5′-TGCTACAGTCCCTGGAGTGA-3′) was ordered as single guide RNA (sgRNA) from Synthego, and Cas9 protein (purified in the lab) was used instead of plasmid. The donor templates for these cell lines to insert either EGFP or Halo tag were the same as above, respectively.

$2 \times 10^5$ cells were combined with 4 µg Cas9 protein, 45 pmol sgRNA, and 300 ng purified PCR product into a nucleofection cuvette. The nucleofection protocols were the following: MCF7-AP2-halo: EN130, buffer SF (Lonza #V4XC-2031); HDF-AP2-halo: EO114, buffer SF (Lonza #V4XC-2031); Caco2-AP2-halo: DG113, buffer SF (Lonza #V4XC-2031); HeLa-AP2-GFP: CN114, buffer SF (Lonza #V4XC-2031).

Cell lines were then treated with 1 µM DNA-PKc inhibitor NU7441 for 48 h after nucleofection.

### Generating the U2Os-AP2-GFP-ITGB5-mScarlet cell line
This cell line was produced by the same protocol as the U2Os-AP2-GFP cells with the following changes:

The gRNA sequence was 5′-CAAATCCTACAATGGCACTG-3′ and the donor template was: 5′-*GGTTTGAGTGTGTGAGCTAAC ATGTGTCCTCATCCTCTTCCCCGCCGTGTTCTGTAGGCTTCA AATCCATTATACAGAAAGCCTATCTCCACGCACACTGTGGAC TTCACCTTCAACAAGTTCAACAAATC**A**TA**T**AA**C**GGCACTGT**T** GAC**GGAAGTGCATCTGGGAGCTCAGGCGCTAGTGGTTCA GCGAGCGGG***GTGAGCAAGGGCGAGGCAGTGATCAAGGAGTTCATG CGGTTCAAGGTGCACATGGAGGGCTCCATGAACGGCCACGAGTTC GAGATCGAGGGCGAGGGCGAGGGCCGCCCCTACGAGGGCACCCAG*

*ACCGCCAAGCTGAAGGTGACCAAGGGTGGCCCCCTGCCCTTCTCC
TGGGACATCCTGTCCCCTCAGTTCATGTACGGCTCCAGGGCCTTC
ATCAAGCACCCGCCGACATCCCCGACTACTATAAGCAGTCCTT
CCCCGAGGGCTTCAAGTGGGAGCGCGTGATGAACTTCGAGGACGGC
GGCGCCGTGACCGTGACCCAGGACACCTCCCTGGAGGACGGCACC
CTGATCTACAAGGTGAAGCTCCGCGGCACCAACTTCCCTCCTGA
CGGCCCCGTAATGCAGAAGAAGACAATGGGCTGGGAAGCATCCACC
GAGCGGTTGTACCCCGAGGACGGCGTGCTGAAGGGCGACATTAAG
ATGGCCCTGCGCCTGAAGGACGGCGGCCGCTACCTGGCGGACTTC
AAGACCACCTACAAGGCCAAGAAGCCCGTGCAGATGCCCGGCGC
CTACAACGTCGACCGCAAGTTGGACATCACCTCCCACAACGAGGAC
TACACCGTGGTGGAACAGTACGAACGCTCCGAGGGCCGCCACTCC*
ACCGGCGGCATGGACGAGCTGTACAAG<u>TAA</u>TGTTTCCTTCTCCGA
GGGGCTGGAGCGGGGATCTGATGAAAGGTCAGACTGAAACG
CCTTGCACGGCTGCTCGGCTTGATCACAGCTCCCTAGGTAGG
CACCACAGAGAAGACCTTCTAGTGAGCCTGGGCCAGGAGCCC
ACAGTGCCT-3′, where **A** = silent mutations in 5′ homology arm, flexible linker region (GSASGSSGASGSASG) is bold, and mScarlet is italic.

## Lentiviral shRNA production and transduction
Lentiviruses for shRNA production were produced using packaging plasmids pCMVR and pMD2.g and specific shRNAs in pLKO.1 vector as follows: 80% confluent HEK293T cells in DMEM supplemented with 10% FBS and 100 U penicillin–streptomycin were transfected using PEI MAX transfection reagent. 5 h later, the medium was changed to DMEM supplemented with 4% FBS and 25 mM Hepes. Media containing lentiviral particles were harvested after 48 and 72 h, filtered (0.45 μm), aliquoted, and stored at –80°C.

U2Os-AP2-GFP cells were transduced with lentiviral media expressing respective shRNAs in the presence of Polybren 8 μg/ml (TR-1003; Sigma-Aldrich) for 5 h and replaced with culture medium. 48 h later, puromycin (1 μg/ml) was added for 24 h to allow selection of transduced cells. Experiments targeting AP2 subunits were performed otherwise similarly but without puromycin selection. All shRNA-silenced cell lines were replated on non-coated glass-bottomed imaging dishes 1 d prior to imaging.

The following sequences were targeted:

ITGB5 shRNAs: TRCN0000057744 (5′-GCATCCAACCAGATG GACTAT-3′) and TRCN0000057745 (5′-GCTGTGCTATGTTTC TACAAA-3′).
ITGB1 shRNAs: TRCN0000029644 (5′-CCTGTTTACAAGGAG CTGAAA-3′) and TRCN0000029645 (5′-GCCTTGCATTACTGC TGATAT-3′).
AP2S1 shRNAs: TRCN0000060263 (5′-GACGCCAAACACACC AACTTT-3′), TRCN0000060266 (5′-GTGGAGGTCTTAAACGAA TAT-3′), and TRCN0000060267 (5′-CACAACTTCGTGGAGGTC TTA-3′).
AP2A1 targeting shRNAs: TRCN0000065108 (5′-GCTGAATAA GTTTGTGTGTAA-3′) and TRCN0000065109 (5′-GCACATTGA CACCGTCATCAA-3′)
Integrin α5 (ITGA5) targeting shRNAS: TRCN0000029651 (5′-CCACTGTGGATCATCATCCTA-3′) and TRCN0000029652 (5′-CCTCAGGAACGAGTCAGAATT-3′).

Tensin1 targeting (TNS1) shRNAs: TRCN0000002953 (5′-GAG GATAAGATTGTGCCCATT-3′) and TRCN0000002954 (5′-CCC AAAGAAGGTACGTGCATT-3′).
Talin1 targeting (TLN1) shRNA: TRCN0000123106 (5′-GCCTCA GATAATCTGGTGAAA-3′)

## Clathrin exon 31 analysis
Cell culture dishes were coated with 10 μg/ml of FN in a cell culture incubator or left uncoated. The next day, U2Os cells were plated to reach confluency in 24 h. RNA extraction, cDNA synthesis, and PCR followed those described by Moulay et al. (2020) with minor variations. Total RNA was extracted from cells using TRIzol reagent with an additional acidic phenol (pH 5.4) extraction step to remove genomic DNA contamination. cDNA synthesis from 1 μg total RNA was carried out using Maxima H Minus Reverse Transcriptase (Thermo Fisher Scientific Scientific) and oligo dT12–18 (Life Technologies). No enzyme reactions were included to confirm that no genomic DNA was present. PCR was performed using Phusion High-Fidelity DNA polymerase (Thermo Fisher Scientific Scientific) with no other variations from Moulay et al. (2020). Primers used were F′ TGC CCTATTTCATCCAGGTCA and R′ ATGGGTTGTGTCTCTGTAGC. Gel images were acquired using a GelDoc XL (Bio Rad).

## Western blots
U2Os were silenced for respective proteins, as mentioned above, and lysed into 150 mM NaCl, 50 mM Tris, pH 8, and 1% NP-40 (Sigma-Aldrich). Protein concentrations were measured with the Pierce BCA kit (Thermo Fisher Scientific) and boiled in Laemmli loading buffer with 10% β-mercaptoethanol. Equal amounts of protein lysates were loaded onto SurePAGE Bis-Tris 4–20% gels (Genscript) and transferred either to 0.2 μm polyvinylidene difluoride (PVDF) or 0.45 μm nitrocellulose (GE Healthcare) membranes. Membranes were blocked in 5% BSA or skimmed milk for 1 h, incubated with primary antibodies at +4°C overnight, and HRP-conjugated secondary antibodies (1706516, 1721019; Biorad) for 1 h in RT. HRP was activated with Supersignal West Femto or Pico reagents (Thermo Fisher Scientific) and bands were detected with ChemiDoc XRS+ (BioRad).

## Microscopy
All live videos and images from fixed samples were acquired with the ONI nanoimager microscope equipped with 405, 488, 561, and 647 lasers, an Olympus 1.49NA 100× super achromatic objective, and a Hamamatsu sCMOS Orca flash 4 V3 camera. Image acquisition software used was NimOS. Fixed samples were imaged in RT and live imaging was carried out at +37°C. Used fluorochromes were GFP, mScarlet, JF-646, Alexa-488, Alexa-594, and Alexa-647.

## Live time-lapse TIRF imaging
### Endogenous AP2 lifetime monitoring
35,000 U2Os-AP2-GFP cells were plated on precoated/non-coated areas of the dish resulting in ~70–80% confluence 16–20 h later, at the onset of imaging. Alternatively, shRNA-silenced cell lines were plated. After overnight culture, 25 μM

Hepes was added and samples were subjected to live TIRF imaging in a preheated +37°C chamber.

The ONI Nanoimager microscope set to TIRF angle was used to acquire AP2 lifetimes at the cell membrane from 300 frames (1 frame/s) with an exposure time of 330 ms. Each video represents endocytic events from two to three cells (total field of view).

## Acute manipulation of integrin activity
### Integrin β1 blocking

Acute modulation of ligand binding activity for integrin β1 was achieved using the function-blocking antibody mab13 (0.3 μg/ml). U2OS-AP2-GFP cells were plated on FN, as explained above, and 16–20 h later subjected to live TIRF imaging. 0-min sample has no mab13 added to control baseline FCL proportions. Immediately after mab13 addition, 5-min time lapses were continuously collected until 35 min; control videos (time point 0) had no mab13 added.

### Integrin β5 blocking

To acutely induce the inhibition of integrin αvβ5, we used the small molecular inhibitor Cilengitide (HY-16141; MedChem Express, 10 μM). U2OS-AP2-GFP cells plated on non-coated imaging dishes were treated with Cilengitide for 15 or 45 min, fixed, stained, and imaged with the ONI Nanoimager microscope at TIRF angle and analyzed for the resulting reduction of RA coverage.

### Cilengitide washout

U2OS-AP2-GFP-ITGB5-mScarlet cells were plated on non-coated imaging dishes and 1 d later confluent monolayers were treated with 1 μM Cilengitide for 15–25 min, during which most FCLs and RAs were dissociated from the cell membrane. Samples were then washed twice and immediately subjected to live TIRF imaging to detect the de novo formation of FCLs and RAs. 1-h time-lapses were acquired with the ONI Nanoimager microscope at TIRF angle, at 30 s intervals, with an exposure time of 100 ms for AP2 and 300 ms for integrin β5.

## Immunofluorescence experiments
### Immunofluorescent staining and imaging

For immunofluorescence experiments, cells were fixed with 4% paraformaldehyde–PBS for 15 min in a +37°C incubator, washed with PBS, and blocked with 1% BSA-PBS. Primary antibodies diluted in 1% BSA–PBS were incubated for 1 h, samples were washed with PBS, and secondary antibodies diluted in 1% BSA–PBS were let to bind for 30 min. Samples were imaged with the ONI Nanoimager microscope using a TIRF angle and exposure times of 500 or 1,000 ms.

## Manipulation of CME machinery to study RA formation

Clathrin assembly at the cell membrane was reduced by silencing two subdomains of AP2 or by overexpressing AP180 ct, which acts as a dominant negative for AP2. U2OS AP2-GFP cells silenced for AP2A1 shRNA #1 and #2 or AP2S1 shRNA #1, #2, and #3, and control shRNA, and plated on non-coated imaging dishes were fixed and stained for integrin αvβ5 and FA marker p-Pax.

Alternatively, AP180 ct was overexpressed in U2OS AP2-GFP cells, and cells were plated on non-coated imaging dishes, fixed the next day, and stained for integrin αvβ5 and p-Pax. RA formation (integrin αvβ5 adhesions without FA marker) and FAs (integrin αvβ5 colocalizing with FA marker) were imaged using TIRF microscopy.

## Manipulation of integrin activity and availability

We performed time-series experiments to block integrin β1 active conformation with the mab13 antibody or to inhibit integrin αvβ5 with Cilengitide. Experiments utilizing mab13 were performed with U2OS-AP2-GFP cells plated on FN-coated dishes to disfavor the preformation of FCLs and RAs. Mab13 (0.3 μg/ml) was added to replicate samples and fixed 15, 30, or 45 min later. Similarly, U2OS-AP2-GFP cells plated on non-coated dishes to favor the preformation of FCLs and RAs were treated with Cilengitide. Samples were stained for integrin αvβ5 and FA marker p-Pax and imaged using TIRF microscopy.

## Cilengitide washout

Cilengitide washout experiments were developed to study acute reappearance of FCLs and RAs in cell cultures. Cells were treated for 15–25 min with 1 μM Cilengitide (or DMSO as control) and washed two times. Unwashed controls were collected. Washed samples were let to recover for indicated time points, fixed, stained for p-Pax, subjected to TIRF imaging, and analyzed for the reappearance of FCLs and RAs (RA coverage).

U2OS-AP2-GFP cells silenced for integrin β1, integrin β5, integrin α5, or Tensin1 and controls were plated either on FN-coated dishes (control shRNA and ITGB1 shRNAs, ITGA5 shRNAs, or TNS1 shRNAs) or on non-coated dishes (control shRNA and ITGB5 shRNAs) and fixed the next day, 16–20 h later. Samples were stained for integrin αvβ5 and FA marker p-Pax and imaged using TIRF microscopy.

## Micropatterning

U2OS-AP2-GFP or U2OS-AP2-GFP-ITGB5-mScarlet cells were seeded onto micropatterned glass coverslips (CYTOO) either precoated with FN 10 μg/ml or left uncoated. Excess amount (60,000) of cells were plated and monitored for attachment to the patterns. The cells plated on FN had attached in 1 h and the cells plated on non-coated micropatterns had attached in 4 h. After attachment, excess cells were carefully rinsed and the samples were fixed 16–20 h later, stained, and subjected to immunofluorescence analysis with TIRF microscopy.

## Wound healing stimulated migration

U2OS-AP2-GFP-ITGB5-mScarlet cells were plated confluent onto non-coated imaging dishes for 2 d. Fully confluent monolayers were wounded with a micropipette tip, washed twice with fresh complete medium, and let to migrate. 0-min samples were collected directly after wounding. After fixing, samples were stained and subjected to TIRF imaging. Tile images (5 × 1), starting at the edge of the wound, were acquired with a 25% overlap and stitched using the pairwise stitching plug-in Image J.

## Image analyses
### CME lifetime analyses (FCL proportion)
To track CME events and measure lifetimes we used "u-track 2.0" multiple-particle tracking MATLAB software at default settings (Jaqaman et al., 2008). For all experiments, "*n*" refers to a movie, which contained two to four cells. To determine the proportion of FCLs, we used the output from u-track to count the number of pits (events lasting longer than 20 s and shorter than 120 s) and the number of FCLs (events lasting longer than 120 s, as described in Saffarian et al., 2009) in all frames. We took a conservative approach to identify bona fide CME events (i.e. clathrin coated pits—CCPs), where events that were present at the start or lasted beyond the end of the movies were not counted as CCPs. This approach artificially led to higher FCL proportions in the first and final 120 movie frames. Therefore, FCL proportions are presented as the average FCL proportion from frames 120–175 for each movie.

### Other analyses
With the exception of FCL proportions, all image analyses were performed using ImageJ. Simple fluorescence measurements were done manually. Others were performed using custom scripts as shown below.

### RA (and FA) coverage
Individual cells were marked, and ITGB5 (or αvβ5) and p-Pax channels were transformed into binary masks using the Robust Automatic Threshold Selection function. The Robust Automatic Threshold Selection parameters used for segmentation ("noise," "lambda," and "min") were defined by visual inspection for each experiment. The mask of the p-Pax channel was then subtracted from the mask of the ITGB5 (or αvβ5) channel using the "image calculation" function. This calculation results in a mask containing only the RAs (i.e., ITGB5 [or αvβ5] signals not colocalizing with p-Pax). RA coverage was then calculated by dividing the area covered by RAs in the RA mask (integrated density/255) by the area of each marked cell (or cells) in the image. An illustration of this method is shown in Fig. S6. Data is presented as a percentage of the cell area covering RAs.

Focal adhesion coverage (Fig. S2 A) was calculated by dividing the area of the p-Pax mask by the area of each marked cell (or cells) in the image.

For the wound healing experiment, a line on the migration front of each image was manually drawn. This line was then used as a reference to automatically draw a box 100 pixels in width (11.7 μm). RA coverage (as above) was calculated for this box, which was then moved inward in the culture in 50-pixel steps, where the RA coverage analysis was repeated. Values are normalized to the average RA coverage on the three innermost areas in the culture.

### AP2-ITGB5 dynamics
Events showing the appearance of both AP2 and ITGB5 were identified by visual inspection of videos. For the generation of graphs, we selected only events where we could unambiguously ensure that significant FCLs and ITGB5 signals were not present in the region for at least 3 min. For the events where an RA was established (Fig. 5 B, left), time zero was defined as the frame where AP2 signal appeared, and fluorescence intensity from a 10 × 10 μm region around each event was measured for 3 min before (six frames) and 5 min after (10 frames). For the events where ITGB5 clusters did not result in established RAs (Fig. 5 B, right), time zero was defined as the frame where the ITGB5 signal disappeared. Fluorescence was normalized to the highest value in these frames.

### AP2 intensity per colocalization status
AP2, ITGB5, and p-Pax channels were segmented using the Robust Automatic Threshold Selection function (similarly to what was done in the RA coverage experiments). Individual AP2 spots were identified as regions of interest (ROIs) using the "analyze particles" function. The fluorescence intensity for each AP2 spot (i.e., ROI) was then measured from the original image. Using the same AP2 ROIs in the ITGB5 and p-Pax binary masks, we then classified each AP2 spot for their colocalization with either marker. In this case, an AP2 ROI with a non-zero signal in either mask was considered colocalizing. We used full images for these analyses.

### RAs colocalizing to AP2
Individual cells were marked and AP2, ITGB5, and p-Pax channels were segmented using the Robust Automatic Threshold Selection function. RAs were defined as ITGB5 signals not colocalizing with p-Pax. In the conditions used for these experiments (FN + mab13), RAs were primarily individual spots. The colocalization of RAs to AP2 was classified by measuring the intensity of each RA region at the segmented AP2 channel.

### FN intensity vs. AP2 intensity
AP2, ITGB5, and P-Pax channels were segmented using the Robust Automatic Threshold Selection function. Each segmented AP2 spot had the fluorescence intensity measured from the original image. A 3 μm × 3 μm region was drawn around each AP2 spot and used to measure the intensity of FN from the original image. Data are presented as the fluorescence for each AP2 spot.

### Statistics
Figure legends state the exact *n*-values and individual repeats used in analyses. For multiple comparisons, one-way ANOVA was performed followed by Tukey's multiple comparison. Pairwise comparisons were performed using two-tailed Student's *t* test with two-tailed distributions. Data distributions were assumed to be normal, but this was not formally tested. All graphs and statistical calculations were performed with GraphPad Prism 9.

### Online supplemental material
Fig. S1 shows that FN-rich ECM inhibits the formation of FCL and RA. Refers to Fig. 1. Fig. S1 A shows representative stills and kymographs from live TIRF imaging analyzed in Fig. 1 B. Fig. S1 B shows the distribution of FCL and CCP according to FN intensity secreted by U2Osc cells. Fig. S1 C shows how clathrin heavy chain exon31 is expressed in U2Os plated on non-coated or

FN-coated dishes. Fig. S1 D shows the deposition of VTN produced by the cells. Fig. S1 E shows the complementary integrin αvβ5 intensity from samples analyzed for FN intensity in Fig. 1 D. Fig. S2 shows that integrin αvβ5 is necessary in the formation of FCL. Refers to Figs. 3 and 4. Fig. S2 A shows the complementing FA coverage to the RA coverage analyzed in Fig. 3, C and D. Fig. S2, B–E shows the effect of ITGB5 silencing on FCL proportions and integrin αvβ5 intensity. Fig. S2, F and G shows the efficacy of Cilengitide in dissolving RAs. Fig. S3 shows optimization experiments for visualizing FCL and RA co-assembly. Refers to Fig. 3. Fig. S3, A and B shows how Cilengitide washout was used in Fig. 5 to monitor de novo formation of FCL and RAs. Fig. S3, C and D shows how analyses were carried out in Fig. 5 B. Fig. S4 shows that silencing of integrin α5β1 promotes FCL and RA formation. Refers to Fig. 7; Fig. S4, A–C shows the integrin β1 silencing efficacy. Fig. S4, D and E shows the integrin α5 silencing efficacy. Fig. S5 shows that knockdown of focal and fibrillar adhesion components or lateral confinement promote FCL and RA formation. Refers to Fig. 8 and Fig. 9, A–C and shows the Tensin1 silencing efficacy. Fig. S5, D–F shows the Talin1 silencing efficacy. Fig. S5, G–J shows the effect of Talin1 silencing on RA coverage and on active integrin β1 intensity. Fig. S5, K and L shows representatives of both H and arrow-shaped micropatterned samples analyzed in Fig. 9, A–C. Fig. S6 explains in detail how RA coverage analyses were carried out throughout this article. IlB images and gels in the supplemental figures are provided in Source Data files. Video 1 shows representative videos of U2Os-AP2-GFP cells plated on different ECM coatings, as analyzed in Fig. 1 B; stills and kymographs of these videos are in Fig. S1 A. Video 2 shows representative video of U2Os-AP2-GFP cells plated on FN patterned dishes from Fig. 1. E–G. Video 3 is a representative video of U2Os-AP2-GFP-ITGB5-mScarlet cells showing a new RA forming; stills and analysis are shown in Fig. 5. Video 4 is a representative video of U2Os-AP2-GFP-ITGB5-mScarlet cells showing a small RA expanding from an RA/FCL puncta; stills and analysis are shown in Fig. S3 C.

## Acknowledgments

We would like to thank the Helsinki Institute of Life Science (HiLIFE) Light Microscopy Unit and the HiLIFE Flow Cytometry unit for technical assistance. We would like to thank Pekka Lappalainen, Tai Arima, and Markku Hakala for the critical and kind reading of our manuscript.

L. Almeida-Souza is supported by HiLIFE, the Academy of Finland (Research Fellow), Sigrid Juselius Foundation (Young PI grant), Finnish Diabetes Research Foundation, Magnus Ehrnrooth Foundation, and Instruct-ERIC (R&D research award). Open access funded by the University of Helsinki Library.

Author contributions: L. Hakanpää designed research and performed most experiments (CRediT: Conceptualization, investigation, formal analysis, writing). A. Abouelezz performed experiments and performed all Matlab image analysis (CRediT: Investigation, formal analysis). A.-S. Lenaerts designed and generated all knock-in cell lines (CRediT: Methodology). S. Culfa performed and analyzed all Western blots (CRediT: Investigation, formal analysis). M. Algie performed the experiments for the detection of alternative splicing of clathrin (CRediT: Investigation, formal analysis). J. Bärlund and P. Katajisto helped with the isolation of knock-in cell lines by flow cytometry and with experiments with hMECs (CRediT: Resources, methodology, supervision). H.T. McMahon supervised the project during the initial observation (CRediT: Supervision). L. Almeida-Souza designed research, conducted image analysis with ImageJ scripts, and supervised the project (CRediT: Conceptualization, investigation, formal analysis, funding acquisition, supervision, writing). L. Hakanpää and L. Almeida-Souza wrote the manuscript with input from all authors.

Disclosures: The authors declare no competing interests exist.

Submitted: 23 March 2023

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

# Supplemental material

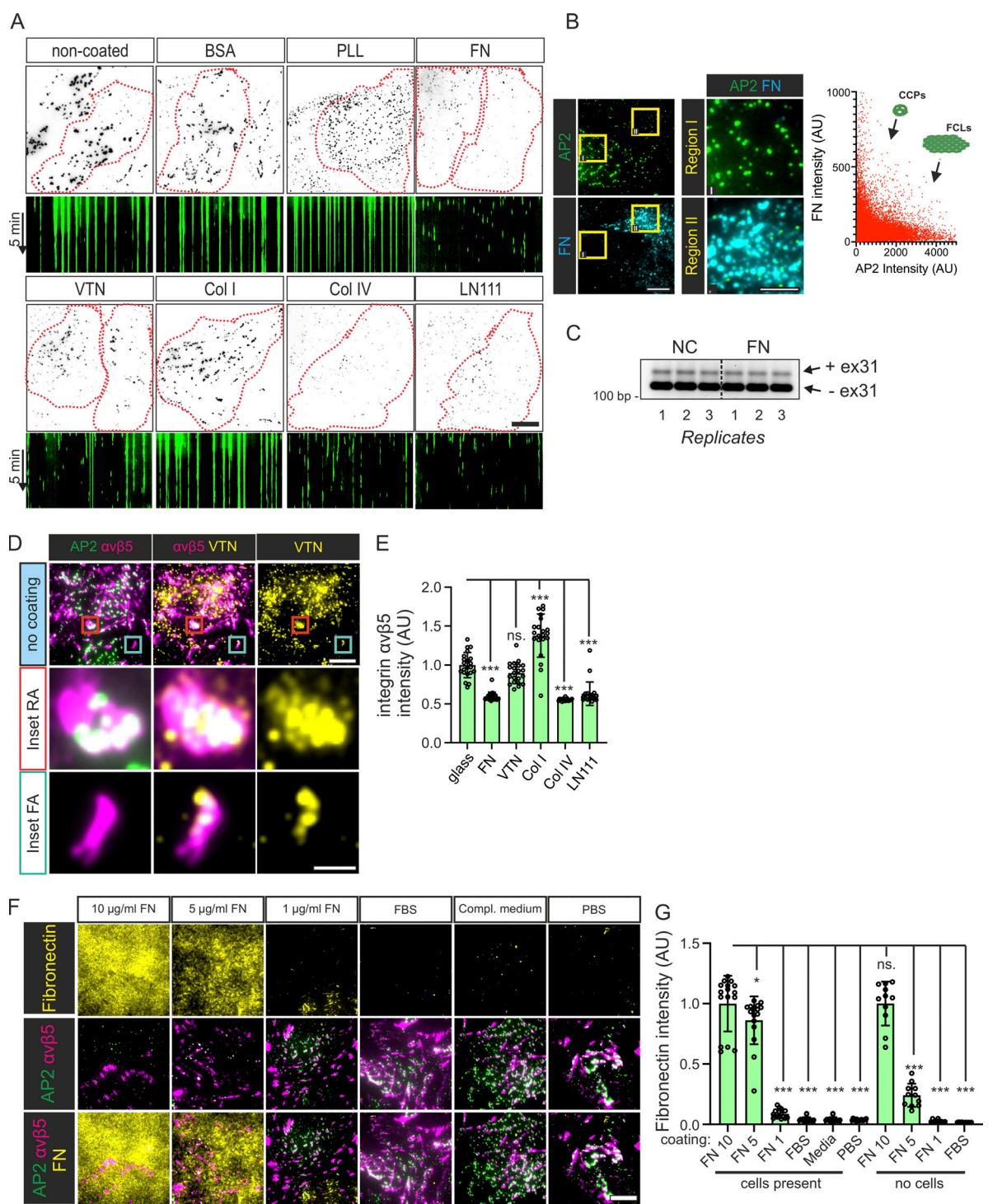

Figure S1. **FN-rich ECM inhibits the formation of FCL and RA. (A)** U2Os-AP2-GFP cells plated on PLL-, BSA-, FN-, VTN-, Col I-, Col IV-, LN111-coated, or non-coated dishes overnight. Samples were imaged with TIRF microscopy at 1-s intervals for 5 min. Representative 15-s time projections and 5-min kymographs of time-lapse videos from samples in Fig. 1 B. **(B)** Left: U2Os-AP2-GFP cells plated on non-coated dishes overnight were stained for FN. Representative TIRF images. Right: Graph showing the reduced brightness of AP2 in regions with higher FN signal (measured from a 1.5 μm × 1.5 μm region around each AP2 spot, n = 32988 AP2 spots, 27 images from one representative sample). **(C)** U2OS cells plated on non-coated (NC) or FN-coated (10 μg/ml) dishes were analyzed for clathrin exon 31 density by RT-PCR; n = 3 biological replicates. **(D)** U2Os-AP2-GFP were plated to non-coated dishes and stained for integrin αvβ5 and VTN. Representative TIRF images. **(E)** Analysis of integrin αvβ5 fluorescent intensity of samples from Fig. 2 A. N (images): FN = 15, VTN/Col I/LN111/non-coated = 21, Col IV = 17. Results were obtained from one representative experiment; similar results were observed in four individual experiments. $F_{(5, 120)}$ = 85.49, P < 0.0001. **(F)** U2Os-AP2-GFP cells plated on 10 μg/ml FN, 5 μg/ml FN, 1 μg/ml FN, FBS, or complete MEM medium-coated dishes were stained for FN and integrin αvβ5. Representative TIRF images. **(G)** FN integrated fluorescent density of dishes coated as in G, and plated or not plated with U2Os cells. N = 16-10/sample, from two independent experiments. $F_{(9, 129)}$ = 184.8, P < 0.0001. Data are the mean ± SD, ns. non-significant P value; *** P value < 0.001. Scale bars, 10 and 5 μm; insets, except in D, are 2 μm. Source data are available for this figure: SourceData FS1.

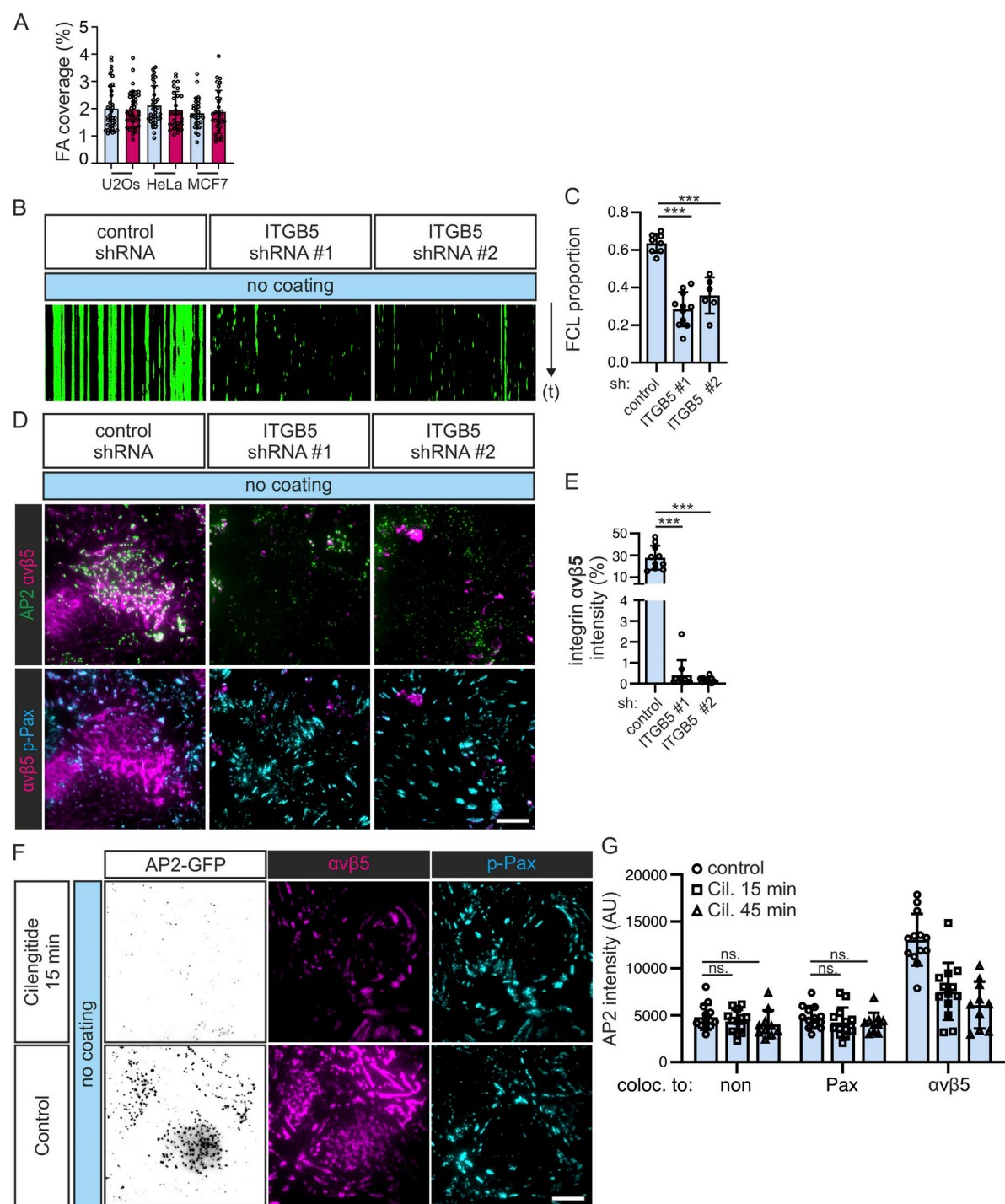

Figure S2. **Integrin αvβ5 is necessary in the formation of FCL. (A)** U2Os-AP2-GFP/halo, HeLa-AP2-GFP/halo, and MCF7-AP2-halo were plated on FN-coated or non-coated dishes, allowed to settle overnight, stained for integrin αvβ5 and p-Pax, imaged using TIRF, and analyzed for FA coverage. *N* (images): U2Os glass/FN = 37, HeLa glass = 32, Hela FN = 29, MCF7 glass = 28, and MCF7 FN = 32. **(B)** U2Os-AP2-GFP cells silenced for integrin β5 (ITGB5) with two shRNAs (shITGB5 #1, #2) or control were imaged using TIRF microscopy at 1-s intervals for 5 min. Representative 5-min kymographs. **(C)** Analysis of FCL proportions from time-lapse videos in A. *N* (videos): control = 8, shITGB5 #1 = 11, shITGB5 #2 = 6, from three independent experiments. One-way ANOVA with Tukey's multiple comparison, $F_{(2, 22)} = 44.46$, P < 0.0001. **(D)** U2Os-AP2-GFP cells silenced for integrin β5 (ITGB5) with two shRNAs (shITGB5 #1, #2) or control, were stained for integrin αvβ5 and p-Pax. Representative TIRF images. **(E)** Analysis of integrin β5 silencing efficiency, *n* = 10, from one representative experiment. Similar results were observed from three independent experiments. One-way ANOVA with Tukey's multiple comparison $F_{(2, 27)} = 63.29$, P < 0.0001. **(F)** U2Os-AP2-GFP cells plated on non-coated dishes for 20 h were treated with the integrin αvβ5 inhibitor Cilengitide (10 μM) for 15 or 45 min and stained for integrin αvβ5 and p-Pax. Representative TIRF images. **(G)** Analysis of AP2 signal colocalizing with p-Pax or integrin αvβ5 over time from samples in E. *N* (images): control = 14, Cil 15 min = 13, Cil 45 min = 10, from one representative experiment. Similar results were observed in four independent experiments. One-way ANOVA and Tukey's multiple comparison $F_{(2, 99)} = 63.38$, P < 0.0001. Data are the mean ± SD, ns. non-significant *P*-value; *** *P*-value < 0.001. Scale bars, 10 μm.

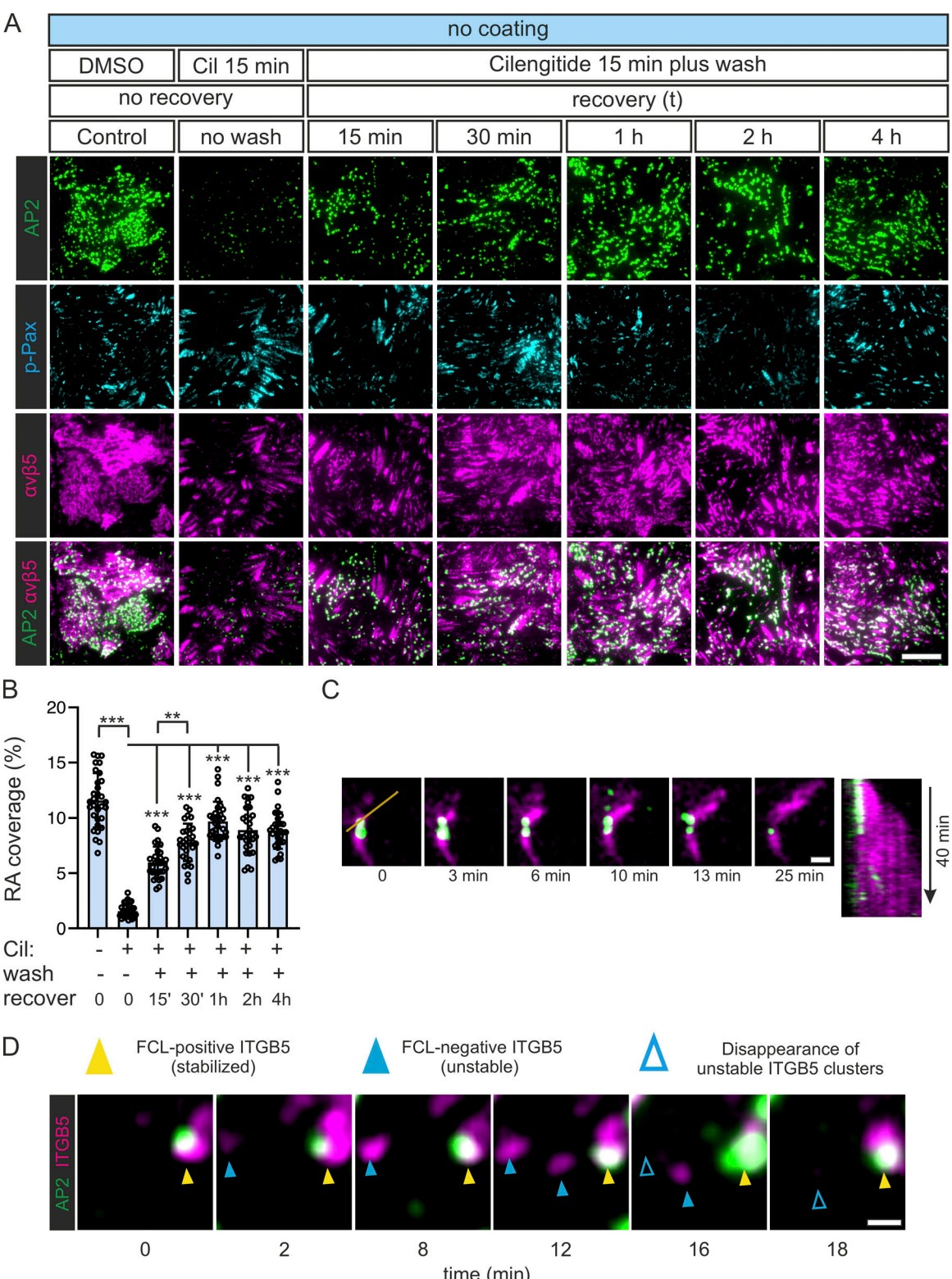

Figure S3. **Optimazition experiments for visualizing FCL and RA co-assembly. (A)** U2Os-AP2-ITGB5-mScarlet cells plated on non-coated dishes were treated with Cilengitide (10 μM) or DMSO for 25 min. Samples given Cilengitide were washed twice with fresh complete medium (except no-wash control), fixed at the respective recovery time points, and stained for p-Pax. Representative TIRF images. **(B)** Analysis of RA coverage for samples in A. *N* (images): control = 31, no wash = 32, 15-min recovery = 30, 30-min recovery = 30, 1-h recovery = 31, 2-h recovery = 30, 4-h recovery = 30, from one representative experiment. Similar results were observed in three independent experiments. **(C and D)** Additional examples of individual ITGB5 and AP2 events from experiments shown in Fig. 5. In C, an event where an RA grows from a stabilized FCL/ITGB5 cluster. A kymograph for the line on time zero is shown on the right. In D, three events are shown. One FCL-stabilized ITGB5 cluster (yellow arrowheads) and two non-stabilized ITGB5 clusters (blue arrowheads). Open blue arrows represent frames post-disappearance of ITGB5 clusters. Scale bars, 10 μm.

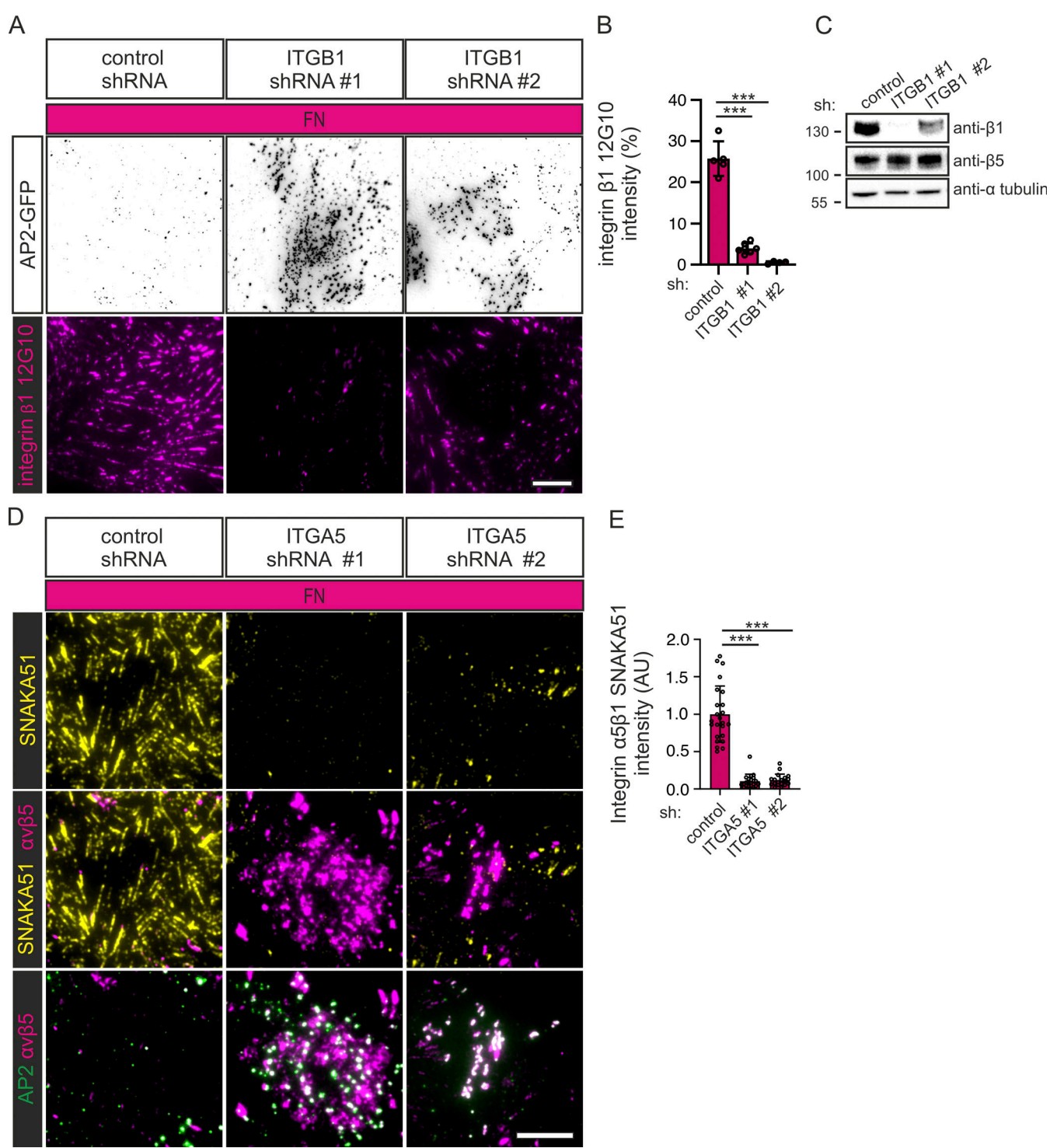

Figure S4. **Silencing of integrin α5β1 promotes FCL and RA formation. (A)** U2Os-AP2-GFP cells silenced for integrin β1 with two different shRNAs (shITGB1 #1, #2) or control shRNA were plated on FN-coated dishes and stained for active integrin β1 (12G10 antibody). Representative TIRF images. **(B)** Analysis of 12G10 fluorescent intensity. $N$ = 7 from one representative experiment. Similar results were observed from two individual experiments. One-way ANOVA with Tukey's multiple comparison $F_{(2, 13)}$ = 149.9, P < 0.0001. **(C)** Western blots showing integrin β1 silencing efficiency and the effect on integrin β5 protein levels. U2Os cells silenced for integrin β1 shRNAs (shITGB1 #1, #2) or control shRNA were blotted for integrin β1, integrin α5, and α-tubulin. Representative blots out of two individual experiments. The position of molecular weight markers (in kDa) are shown on the left. **(D)** U2Os-AP2-GFP-ITGB5-mScarlet cells silenced for integrin α5 with two different shRNAs (shITGA5 #1, #2) or control shRNA were plated on FN-coated dishes and stained for integrin α5 (SNAKA51 antibody). Representative TIRF images. **(E)** Analysis of SNAKA51 fluorescent intensity from widefield microscopic images. $N$ (images): shScr = 24, shITGA5 #1 = 20, shITGA5 #2 = 19 from one representative experiment. Similar results were observed in two individual experiments. One-way ANOVA with Tukey's multiple comparison, $F_{(2, 60)}$ = 1, P < 0.0001. Data are the mean ± SD, *** P < 0.001. Scale bars, 10 μm. Source data are available for this figure: SourceData FS4.

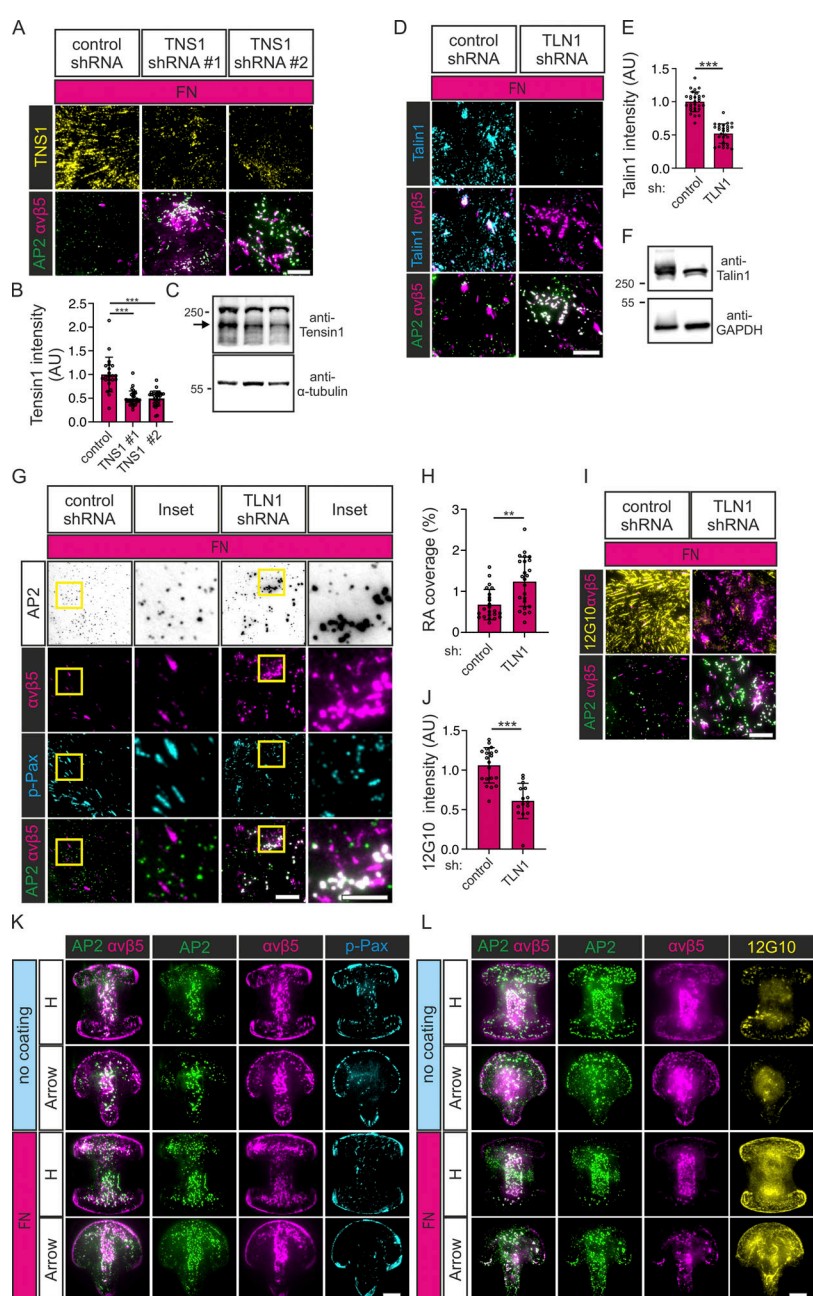

Figure S5. **Knockdown of focal or fibrillar adhesion components or lateral confinement promote FCL and RA formation. (A)** U2Os-AP2-GFP-ITGB5-mScarlet cells silenced for Tensin1 with two different shRNAs (shTNS1 #1, #2) or control shRNA were plated on FN-coated dishes and stained for Tensin1. Representative TIRF images. **(B)** Analysis of Tensin1 fluorescent intensity from samples in A. *N* (images): shScr control = 22, shTNS1 #1 = 29, shTNS1 #2 = 31, from one representative experiment. Similar results were observed in three individual experiments. One-way ANOVA with Tukey's multiple comparison $F_{(2, 79)}$ = 38.09, P < 0.0001. **(C)** Representative Western blots of Tensin1 silencing efficiency. U2Os-AP2-GFP-ITGB5-mScarlet cells silenced for Tensin1 with two different shRNAs (shTNS1 #1, #2) or control shRNA were blotted for Tensin1 and α-tubulin. The arrow marks the correct Tensin1 band. The position of molecular weight markers (in kDa) are shown on the left. **(D)** U2Os-AP2-GFP-ITGB5-mScarlet cells silenced for Talin1 with shRNA or control shRNA were plated on FN-coated dishes and stained for Talin1. Representative TIRF images. **(E)** Analysis of Talin1 fluorescent intensity from samples in D. *N* (images): shScr = 32, shTLN1 = 31, from two individual experiments. Two-tailed Student's *t* test, P < 0.0001. **(F)** Representative Western blots of Talin1 silencing efficiency. U2Os-AP2-GFP-ITGB5-mScarlet cells silenced for Talin1 shRNA or control shRNA were blotted for Talin1 and GAPDH. The position of molecular weight markers (in kDa) are shown on the left. **(G)** U2Os-AP2-GFP-ITGB5-mScarlet cells silenced for Talin1 with shRNA or control shRNA were plated on FN-coated dishes and stained for p-Pax. Representative TIRF images. **(H)** Analysis of RA coverage from samples in G. *N* (images): *n* = 23. Two-tailed Student's *t*-test, P < 0.0001. **(I)** U2Os-AP2-GFP-ITGB5-mScarlet cells silenced for Talin1 with shRNA or control shRNA were plated on FN-coated dishes and stained for integrin β1 12G10. Representative TIRF images. **(J)** Analysis of 12G10 fluorescent intensity with samples from I. *N* (images): shScr = 20, shTLN1 = 15, from one representative experiment. Similar results were observed in two individual experiments. Two-tailed Student's *t* test, P < 0.0004. **(K)** U2Os-AP2-GFP cells were grown on FN-coated or non-coated micropatterns (1,100 mm²) and stained for integrin αvβ5 and p-Pax. Representative TIRF images. **(L)** U2Os-AP2-GFP-ITGB5-mScarlet cells were grown on FN-coated or non-coated micropatterns (1,100 mm²) and stained for active integrin β1 12G10. Representative TIRF images. Data are the mean ± SD, ** P < 0.01, *** P < 0.001. Scale bars, 10 µm; insets, 5 µm. Source data are available for this figure: SourceData FS5.

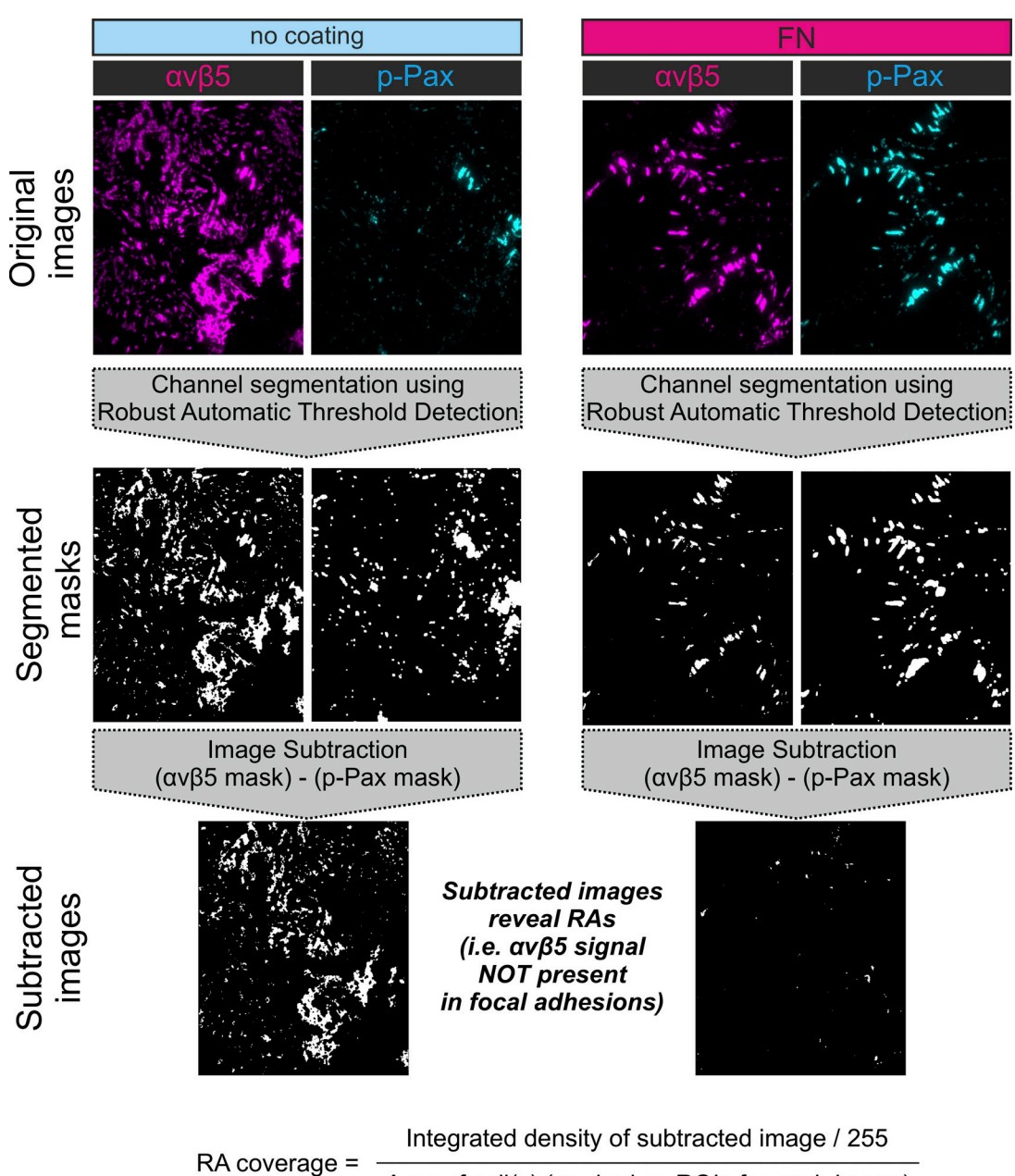

$$\text{RA coverage} = \frac{\text{Integrated density of subtracted image} / 255}{\text{Area of cell(s) (marked as ROIs for each image)}}$$

Figure S6. **Illustration of the RA coverage calculation method.** To calculate RA coverage, images of the integrin (αvβ5) and paxillin (p-Pax) channels are segmented using the Robust Automatic Threshold Selection method in ImageJ. This method generates binary masks for each channel. The paxillin channel is then subtracted from the integrin channel and results in a mask containing only the RAs. RA coverage is calculated by dividing the area covered by RAs in pixels (integrated density/255) by the area covered by the cell (or cells) in an image.

Video 1. **Representative videos of data presented in** Fig. 1 B **and** Fig. S1 A. U2OS-AP2-GFP cells plated on uncoated dishes or on FN-, VTN-, Col I-, Col IV-, LN111-, BSA-, or PLL-coated dishes were imaged using TIRF microscopy, 1 s/frame. Video played at 15 frames/s.

Video 2. **Representative video of data presented in** Fig. 1 E. U2OS-AP2-GFP cells plated on FN/uncoated patterned imaging dishes were imaged with TIRF microscopy 1 s/frame. Video played at 15 frames/s.

Video 3.   **Video used to generate** Fig. 5 A. U2OS-AP2-GFP-ITGB5-mScarlet cells plated on glass were treated with Cilengitide (10 μM) for 15 min, washed, and imaged using TIRF microscopy at 30 s/frame. Video played at 10 frames/s.

Video 4.   **Representative video of the data presented in** Fig. S3 C. U2OS-AP2-GFP-ITGB5-mScarlet cells plated on glass were treated with Cilengitide (10 μM) for 15 min, washed, and imaged using TIRF microscopy at 30 s/frame. Video played at 10 frames/s.

