## [Peer Review File · The Journal of Cell Biology]

Reticular adhesions are assembled at flat clathrin lattices and opposed by active integrin $\alpha 5\beta 1$

Laura Hakanpää, Amr Abouelezz, An-Sofie Lenaerts, Seyda Culfa, Michael Algje, Jenny Bärlund, Pekka Katajisto, Harvey McMahon, and Leonardo Almeida-Souza

Corresponding Author(s): Leonardo Almeida-Souza, University of Helsinki

Review Timeline:

Submission Date:	2023-03-23
Editorial Decision:	2023-04-24
Revision Received:	2023-05-03
Editorial Decision:	2023-05-05
Revision Received:	2023-05-10

Monitoring Editor: Martin Humphries

Scientific Editor: Andrea Marat

Transaction Report:

DOI: <https://doi.org/10.1083/jcb.202303107>

Revision 0

Review #1

1. Evidence, reproducibility and clarity:

Evidence, reproducibility and clarity (Required)

In this manuscript, Hakanpää et al explored the connection between flat clathrin lattices and reticular adhesions and the regulatory mechanism underlying the formation of these two structures in the U2OS cells. The author provided evidence that the composition of the extracellular matrix plays critical roles in their formation and concluded that fibronectin and its receptor, $\beta 1$ integrin, inhibit the assembly of FCLs and RAs. The author depleted several components of the clathrin mediated endocytosis machinery and could show that it blocked the formation of reticular adhesions.

****Major comments****

1. In Fig. 1, the author plated U2OS cells on surfaces coated with different ECM components and then measured the frequency of FCLs. How long were the cells allowed to attach to the surfaces before they were imaged? Were the cells serum-starved before seeding? Given the fact that cells attached well even on BSA-coated dishes, I guess the cells were allowed to attach and spread for at least overnight. In this case, the ECM components (vitronectin is very abundant in the serum in a concentration of 200-400 $\mu\text{g}/\text{mL}$, fibronectin is another one) in the culture medium would coat the glass surface and this has profound effects on the adhesion status of the cells. Thus, more details of this experiment need to be included and more attention should be paid regarding the data interpretation. In Fig. 1C, it seems like the mere glass surface induced the most FCL formation. However, if the cells are grown on the glass overnight or for days, the major component of the surface would actually be vitronectin and fibronectin (maybe more) rather than glass, thus it is not accurate to say that 'VTN reduced FCL frequency to some extent compared to glass' (Line 67).

2. Line 91-93. It is not accurate to claim that the reticular adhesions are the only type of cellular adhesion maintained during mitosis. In several studies, active integrin $\beta 1$ are found along the retraction fibers (Dix et al, Dev Cell, 2018; Chen et al, NCB, 2022). The importance of $\alpha V\beta 5$ integrin in the spatial memory during mitosis was only shown in the in vitro cell culture. In fact, mice lacking $\alpha V\beta 5$ integrin or its ligand vitronectin are both viable and show no major defects during embryonic development (Zheng et al, PNAS, 1995; Huang et al, Mol Cell Biol, 2000), suggesting reticular adhesions are dispensable in cell division in vivo. I advise to change it into 'RAs are composed of $\alpha V\beta 5$ integrin and are maintained during mitosis in culture'.

3. It has been shown that fibronectin and laminin coating inhibit formation of reticular adhesions (Lock et al, NCB, 2018, Fig. S7). This study should be cited in Fig. 1. I suggest to also include laminin in Fig. 1C to make the list of ECM components more complete.

4. Fig. 5C, the fluorescent intensity of $\alpha V\beta 5$ integrin is increased dramatically when integrin $\beta 1$ was depleted compared to the control shRNA. Although the images were collected in the TIRF

mode, it is important to measure $\alpha V\beta 5$ and $\beta 1$ integrin level by immunoblot to confirm the knockdown efficiency of $\beta 1$ integrin and exclude the possibility that the increase of RA formation is not due to the compensation by upregulation of $\alpha V\beta 5$.

5. Antibody-blocking or depletion of $\beta 1$ integrin both lead to accumulation of FCL and RA formation, indicating that the activation of $\beta 1$ integrin is critical in the inhibition of FCL and RA. Activation of $\beta 1$ integrin depends on talin and kindlin, which bind $\beta 5$ integrin with a much lower affinity. Would depletion of talin or kindlin cause FCL and RA formation similar to inhibition of $\beta 1$ integrin?

****Minor comments****

1. In most of the quantifications, only the number of the cells measured were mentioned in the legend. The number of replicate experiments is missing. It should be included in the legend as well.

2. A recent study (<https://doi.org/10.1242/jcs.259465>) demonstrated the molecular mechanism underlying the localization of $\alpha V\beta 5$ integrin in flat clathrin lattices. It should be mentioned in the introduction.

2. Significance:

Significance (Required)

Although it is not novel that FCLs and RAs share same localization and might actually be the different parts of the same structure (Zuidema et al, JCS, 2022), the observation that inhibiting $\beta 1$ integrin stimulates FCL and RA assembly is interesting as it indicates the counter-balance between the $\alpha V\beta 5$ and $\alpha 5\beta 1$ integrins. It is a pity that the author did not dig deeper into the mechanism underlying this interesting finding, which should greatly increase the impact of this study.

3. How much time do you estimate the authors will need to complete the suggested revisions:

Estimated time to Complete Revisions (Required)

(Decision Recommendation)

Between 1 and 3 months

4. *Review Commons* values the work of reviewers and encourages them to get credit for their work. Select 'Yes' below to register your reviewing activity at Web of Science Reviewer Recognition Service (formerly Publons); note that

the content of your review will not be visible on Web of Science.

Yes

Review #2

1. Evidence, reproducibility and clarity:

Evidence, reproducibility and clarity (Required)

****Summary:****

Hakanpää and colleagues report on the relationship between flat clathrin lattices (FCLs) and reticular adhesions, with FCLs being proposed to nucleate reticular adhesions. Overall the experimental work is high quality and the data are generally well presented.

****Major comments:****

The Introduction is very brief and doesn't cover the information required to understand the paper. There are three cellular structures to be understood: focal adhesions, reticular adhesions and FCLs. The intro jumps straight into the FCLs (the paper is written from a FCL point of view) but there is no information about the other two structures really particularly the differences between them. Furthermore, there is nothing in the intro about cellular adhesion or why this is even worth studying. The authors should fully revise this Intro, there is much room for improvement!

Fig 4D I could not understand this plot and the legend did not describe it properly. What are the units of FCL frequency? I guess it is FCLs per some distance (10 μ m?), the images need a scale bar. OK, I read the description in the methods and now I see it is the proportion of total CCSs that are FCLs; so frequency is the wrong term. The legend says that there were 32 videos and there are 32 points on the plot but what we need to know is: where $n = 1$ cell, what did the FCL frequency do over time? A line is drawn on the graph, no info on what the line is and the fit is poor ($R^2 < 0.5$). The authors should take their series of data points from individual cells and fit curves to each and describe the summary statistics of the parameters of these fits OR average the data and fit to that, using the 1 SD of the data for weighting the fit. Probably more data is required to do any meaningful fitting here. To my eye it looks like FCL frequency goes from 0.3 to a plateau of 0.5 at 10 min and that more timepoints between 0 and 10 min would have been useful.

Fig 3F/G is a nice expt. It looks as though the ITGB5 signal is already creeping up when the AP2 arrives. I agree that they accelerate together, but the prior accumulation of integrin is at odds

with the conclusion that AP2/clathrin *nucleates* the adhesion. This experiment is missing two controls: what is the behaviour of ITGB5 in AP2 negative regions? What happens to both signals in the continued presence of cilengitide? These controls are needed to conclude that AP2 is nucleating the adhesion.

****Minor comments:****

Fig 6 - several typos - "adaptor proteins" "engagement" "containing"

2. Significance:

Significance (Required)

Previously, endocytosis (clathrin-mediated) was thought to decrease cellular adhesion by removal of integrins. This paper suggests that the same machinery can be used to build adhesions. This is a surprising conclusion that will be of interest to many cell biologists; the topic of clathrin and adhesion is being actively explored by several labs either from the adhesion or the endocytosis sides. I have been following this topic from a distance and don't know the details of all the published papers, but this paper does seem to add something new over the recent work from Taraska, Sonnenberg, Montagnac, Strömblad.

3. How much time do you estimate the authors will need to complete the suggested revisions:

Estimated time to Complete Revisions (Required)

(Decision Recommendation)

Between 1 and 3 months

Yes

Review #3

1. Evidence, reproducibility and clarity:

Evidence, reproducibility and clarity (Required)

****Overview:****

This manuscript addresses the emerging nexus linking the machinery associated with clathrin endocytosis (clathrin-coated pits; CCPs), flat clathrin lattices (FCLs) and the recently discovered Reticular Adhesions (RAs). This is timely work, reflecting recent foci on the relationship between these structures and systems.

Initially clearly identifying reductions in FCL and RA formation on fibronectin, the authors sought to clarify the mechanisms that suppress or prevent FCL / RA formation on this matrix. Knock-down of integrin avb5 (core RA component) suppressed both RA and FCL formation, suggesting a dependence of FCLs on this integrin. This was supported by acute avb5 inhibition via cilengitide (avb5 and avb3 inhibitor) which caused disassembly of existing RAs and FCLs.

Notably, the inverse relationship also appears true, with suppression of core clathrin endocytic machinery (AP2 complex components) being sufficient to greatly deplete RA formation. Supporting this finding, overexpression of a dominant negative-acting protein fragment (AP180 c-terminal fragment) blocked both AP2 localisation to the plasma membrane and RA formation.

To unmix this bi-directional dependence further, the authors used acute cilengitide treatment followed by washout and post-washout incubation to first deplete RAs (cilengitide) and then allow monitoring of en masse RA formation after cilengitide washout. This is an effective experiment, however, the analysis would benefit from greater depth, particularly relating to the order of events. Analysis of this aspect seems central to the thrust of the paper, and some statistical analysis of either static co-occurrence or dynamic ordering in large numbers of FCL / RA structures (i.e. hundreds) would be of value.

The authors next focused on the observation that fibronectin suppressed both FCL and RA structures, by assessing the role of fibronectin-receptor integrin b1. Acute antibody mediated integrin b1 inhibition (mab13) and integrin b1 knock down both confirmed that in cells on fibronectin, suppression of integrin b1 is sufficient to permit massive upregulation of both FCL and RA formation. This is surprising and very interesting. It raises questions about the actual ECM requirements for avb5-mediated reticular adhesion formation. It would seem that fibronectin per se can support very efficient RA / FCL formation, but that normally concurrent integrin b1 activities would suppress this. Given this implication, it would be especially important to clarify the purity of FN ECM coating (as explored in questions 1-3 below) at the time of imaging.

The discussion addresses a number of topics, and proposes a mechanistic model to explain the results presented. I don't find the mechanism very convincing, as the directionality of the dependence between FCLs and RAs is not clearly delineated by the experiments presented, in my opinion. That there is co-dependence is convincingly shown, but whether there is

directionality, and what order of events underpins FCL then RA or RA then FCL formation, is unclear. Nonetheless, the evidence presented does generally support the idea of a shift in the way we consider the role of endocytic machinery in adhesion regulation, from a disassembly only related function to additional functions associated with adhesion formation and maintenance. Ideally, the mechanisms around this new assembly / maintenance function would be further delineated here, but regardless, this work does point in the direction of important new questions in this area.

Further discussion about the potential role of this interdependent regulatory process in, for example, mitosis, seems unwarranted and should probably be removed.

****Questions:****

1. A technical question on the replating experiments onto specific matrix proteins; after coating surfaces with the purified ECM components or controls, what media were the cells replated in? Ideally this should be serum-free media, to ensure that the ECM components of FBS / FCS are not immediately added to the purified ECM components. This should be clarified in the methods.
2. Related to above, I cannot see how long cells were plated onto the different ECM conditions. This would be relevant to know and should be clarified because cells will secrete ECM over time and thus the purity of the ECM components addressed is dependent on the length of time cells are incubated and imaged for after attachment.
3. Similarly, it is noted that cells plated on 'glass' support RA formation. It should be clarified what ECM component is then actually responsible for cell adhesion and adhesion complex (RA, FA or other) formation
- since this requires an ECM component of some type. Presumably, 'on glass' means whatever ECM is either derived from the media used during cell attachment / incubation (if that media contains serum, which is vitronectin rich), or whatever ECM is secreted by the cells themselves over the attachment / incubation period prior to imaging.
4. In the cilengitide washout experiments, the evidence shown in figures seems to suggest that AP2-positive FCLs form in locations where avb5 (probably RAs) are already present, whereas avb5 positive structures do not form from AP2-only structures. Statistical analysis of this pattern (i.e. which protein is present first) would be valuable to address directionality. Notably, in Figure 3G, it appears that avb5 is present and increasing prior to the subsequent arrival of AP2.
- a. I would suggest that the cilengitide experimental results(3E-G) be shown in a separate figure from the endocytosis inhibition results (3A-D).
5. The integrin b1 inhibition and knock down results are clear and interesting. Clarifying the ECM components present during these experiments would be valuable to interpretation of the paper.

2. Significance:

Significance (Required)

see above

3. How much time do you estimate the authors will need to complete the suggested revisions:

Estimated time to Complete Revisions (Required)

(Decision Recommendation)

Cannot tell / Not applicable

No

Reviewer #1 (Evidence, reproducibility and clarity (Required)):

In this manuscript, Hakanpää et al explored the connection between flat clathrin lattices and reticular adhesions and the regulatory mechanism underlying the formation of these two structures in the U2OS cells. The author provided evidence that the composition of the extracellular matrix plays critical roles in their formation and concluded that fibronectin and its receptor, $\beta 1$ integrin, inhibit the assembly of FCLs and RAs. The author depleted several components of the clathrin mediated endocytosis machinery and could show that it blocked the formation of reticular adhesions.

We would like to thank the reviewer for the comments on our manuscript. We have addressed all your concerns and we genuinely think they have improved our study.

As the reviewer will notice, this revised version is a significant overhaul of our initially submitted manuscript. First, we decided to simplify our message on the role of FCLs in RA formation. We have dropped the nucleation idea and now we simply claim that FCLs are essential for the formation of RAs. This focusses the attention on our strongest results and avoids the “who-comes-first” discussion, which has proven very hard to address (see below). This on itself is an exciting finding that shows a clear function for this long known and little understood clathrin structure. Moreover, we also bring a substantial amount of data on the role of fibronectin as an inhibitor of RA formation by showing that this mechanism is operational in 6 different cell lines (all lines we tested). We also further explored the interesting inter-adhesion role of integrin $\beta 1$ activation in RA inhibition and show that this integrin (and its partner $\alpha 5$) needs to translocate from focal adhesions to fibrillar adhesions to avoid RAs from forming. This activation is dependent on cell movement and as such, we show that the initiation of cell migration is coupled to the disassembly of RAs.

Major comments

1. In Fig. 1, the author plated U2OS cells on surfaces coated with different ECM components and then measured the frequency of FCLs.

How long were the cells allowed to attach to the surfaces before they were imaged? Were the cells serum-starved before seeding?

Thanks for pointing out that this was not clear. We have now clarified the protocol used for the experiments. We have now described the protocol in higher detail in the methods section and have also emphasized it in the start of the results section:

“(...)For that, dishes were coated for 16-24 h after which, cells were let to attach for 16-20h in serum-containing medium before imaging. (...)”

Given the fact that cells attached well even on BSA-coated dishes, I guess the cells were allow to attach and spread for at least overnight. In this case, the ECM components (vitronectin is very abundant in the serum in a concentration of 200-400 $\mu\text{g}/\text{mL}$, fibronectin is another one) in the culture medium would coat the glass surface and this has profound effects on the adhesion status of the cells. Thus, more details of this experiment need to be included and more attention should be pay regarding the data interpretation.

As written above, cells were indeed left to attach overnight in serum-containing media.

We agree that the potential confounding effect of the ECM proteins present in serum is an important factor to consider. We were asked the same question by a few colleagues in conferences. To reinforce our conclusions, we addressed this issue in two ways: (1) by adding additional pieces of evidence further

supporting the role of fibronectin as an inhibitor in FCL and RA formation and (2) by directly addressing the effect of ECM components in serum. Moreover, we have also added new results and discussed the possible role of vitronectin in the process (3).

(1) Pro-fibronectin evidence

In Figures 1C,D, we show that the other two ECM coatings that - alongside fibronectin - also reduce the amount of FCLs (collagen IV or laminin 111) lead to a substantial increase in fibronectin secretion.

Using a panel of 6 cell lines (U2Os, HeLa, MCF7, HDF, Caco2 and hMEC) we show that there is a direct correlation between the amount of fibronectin endogenously secreted by these cells and the amount of FCLs and RAs. Moreover, our main cell model (U2Os) deposits very little fibronectin (despite being “bathed” in serum fibronectin). These results are now shown in Figures 1C and 3.

(2) Why fibronectin in serum does not coat dishes when cells are plated overnight?

This was indeed a puzzling question. However, we knew that, somehow, serum fibronectin was not coating our dishes, otherwise we would not have such clear results in our patterning experiments. To try to understand this apparent conflict, we compared the amount of fibronectin deposited on the glass surface in different conditions: Dishes were coated for 24 h with 10, 5 and 1 $\mu\text{g}/\text{ml}$ of fibronectin (diluted in PBS), 100% fetal bovine serum (FBS), media with 10% FBS and PBS as a control. After coating, U2Os cells were plated and left to attach for 16h before being fixed and stained for FN. As additional controls, coatings without cells were also stained. Surprisingly, our results revealed that very little fibronectin was deposited on glass in dishes “coated” with full media or pure FBS (Figure S1F,G).

At this moment, I imagine one might be asking: How is this possible? Why would fibronectin in PBS deposit on dishes while fibronectin on media (or pure serum) would not? These very questions haunted us for quite some time. The answer was right in front of us, clear to see every time we prepared our cells for immunofluorescence: blocking! Bovine serum albumin is widely used as a blocking agent not only because of its cost and availability, but also because of its coating efficiency. We found out that there is a vast literature characterizing how effective BSA binds to naked and treated glass surfaces. These studies have shown that BSA readily binds to glass, within minutes of its addition (for example, see Ma et al., 2020; Sun and Zhu, 2016). Given that BSA concentrations in serum are on the range of 40mg/ml, this likely explains why fibronectin cannot deposit -passively- in significant amounts during the period of our experiment. What about vitronectin?

Vitronectin is the main ligand for integrin $\alpha\text{v}\beta\text{5}$, placing it in a likely important position on the process we are studying. However, we (Figure S1D) and others (Zuidema et al., 2022) have shown that vitronectin and integrin $\alpha\text{v}\beta\text{5}$ colocalize equally on focal adhesions and RAs, pointing to the fact that the cellular decision to use these integrins on RAs or FAs probably occurs inside-out. This point is discussed in the section “the mechanism of FCL-mediated RA formation” of the discussion. We also added to this discussion section the fact that vitronectin is known to be readily secreted by most cells during adhesion, and caution should be taken when comparing the data on non-coated and vitronectin-coated dishes.

In Fig. 1C, it seems like the mere glass surface induced the most FCL formation. However, if the cells are grown on the glass overnight or for days, the major component of the surface would actually be vitronectin and fibronectin (maybe more) rather than glass, thus it is not accurate to say that 'VTN reduced FCL frequency to some extent compared to glass' (Line 67).

This is a good point. We have now rephrased the whole manuscript to refer to “glass” as non-coated surfaces. This is more accurate and reflects the reality that other ECM components have been deposited by cells. As discussed in the previous point, this possibility is stated in the manuscript text. At the end of the day, we want to emphasize that the presence of sufficient amounts of FN inhibits FCLs and RAs, not that glass - or any other ECM component - promotes the formation of these structures.

We do indeed detect VTN in our cultures. It localizes to integrin α V β 5 equally when on RAs or FCLs (Figure S1D). However, as described above, we see significant differences in fibronectin deposition between cells and the FCL and RA phenotype nicely correlate to fibronectin abundance.

You are right, growing cells for days would indeed make any meaningful comparison of ECM identity difficult. We did notice a variable effects when cells were left on the dishes for more than 48hs. For this reason, cells were imaged within 20 h after plating for all our experiments (except the cilengitide washout and wound healing experiments). This has been further emphasized in the results section and material and methods.

2. Line 91-93. It is not accurate to claim that the reticular adhesions are the only type of cellular adhesion maintained during mitosis. In several studies, active integrin β 1 are found along the retraction fibers (Dix et al, Dev Cell, 2018; Chen et al, NCB, 2022). The importance of α V β 5 integrin in the spatial memory during mitosis was only shown in the in vitro cell culture. In fact, mice lacking α V β 5 integrin or its ligand vitronectin are both viable and show no major defects during embryonic development (Zheng et al, PNAS, 1995; Huang et al, Mol Cell Biol, 2000), suggesting reticular adhesions are dispensable in cell division in vivo. I advise to change it into 'RAs are composed of α V β 5 integrin and are maintained during mitosis in culture'.

That's a fair point. As we do not address the relationship of the effects we see with cell division, we see no reason to get into this issue. This was also suggested by reviewer #3. Thus, we have removed all the references on the participation of α V β 5 in mitosis from the revised manuscript.

3. It has been shown that fibronectin and laminin coating inhibit formation of reticular adhesions (Lock et al, NCB, 2018, Fig. S7). This study should be cited in Fig. 1. I suggest to also include laminin in Fig. 1C to make the list of ECM components more complete.

Thanks for the suggestion. We have now added Laminin 111 to the experiments in Figures 1 and 2. As you can see, it does indeed inhibit the formation of FCLs. Crucially, we go further to show that the two coatings that alongside fibronectin also inhibit the formation of FCLs and RAs - LN111 and ColIV – increased the endogenous secretion of fibronectin (Figure 1C,D). This helped us to conclude that FN is the major player in inhibiting FCLs and RAs.

4. Fig. 5C, the fluorescent intensity of $\alpha V\beta 5$ integrin is increased dramatically when integrin $\beta 1$ was depleted compared to the control shRNA. Although the images were collected in the TIRF mode, it is important to measure $\alpha V\beta 5$ and $\beta 1$ integrin level by immunoblot to confirm the knockdown efficiency of $\beta 1$ integrin and exclude the possibility that the increase of RA formation is not due to the compensation by upregulation of $\alpha V\beta 5$.

Western blots confirming shRNA depletion of integrin $\beta 1$ are now presented in figure S4C. The western blots in the same figure also revealed that the large increase in $\alpha V\beta 5$ on the cell surface is not reflected by an increase in integrin $\beta 5$ expression levels.

5. Antibody-blocking or depletion of $\beta 1$ integrin both lead to accumulation of FCL and RA formation, indicating that the activation of $\beta 1$ integrin is critical in the inhibition of FCL and RA. Activation of $\beta 1$ integrin depends on talin and kindlin, which bind $\beta 5$ integrin with a much lower affinity. Would depletion of talin or kindlin cause FCL and RA formation similar to inhibition of $\beta 1$ integrin?

This is a great suggestion. We have now added results showing that Talin knockdown does indeed induces the formation of FCLs and RAs (Figure S5D-H). We had the same results when we knocked down integrin $\alpha 5$, the main partner of $\beta 1$ in FN binding. We have also gone further and show that the role of integrin $\beta 1$ in FCL and RA inhibition occurs at fibrillar adhesions, and is dependent on another important integrin activating adaptor, Tensin1. We also showed that Talin knockdown results in the loss of fibrillar adhesions (SFig 5 I,J), complementary to the role of FAs as platforms for fibrillar adhesion formation.

In addition, we would also like to share with the reviewer, that in our hands, Kindlin 1 and 2 silencing did cause a similar effect than silencing Talin1, but milder, in line with the role of Kindlins as important accessories of Talin1-mediated integrin activation at FAs. However, with the tools available to us (we purchased two different antibodies), Kindlin silencing efficiencies were difficult to prove conclusively.

Minor comments

1. In most of the quantifications, only the number of the cells measured were mentioned in the legend. The number of replicate experiments is missing. It should be included in the legend as well.

We have added number of replicates for all experiments.

2. A recent study (<https://doi.org/10.1242/jcs.259465>) demonstrated the molecular mechanism underlying the localization of $\alpha V\beta 5$ integrin in flat clathrin lattices. It should be mentioned in the introduction.

A very relevant and elegant study indeed. The reference is now added to the manuscript.

Reviewer #1 (Significance (Required)):

Although it is not novel that FCLs and RAs share same localization and might actually be the different parts of the same structure (Zuidema et al, JCS, 2022), the observation that inhibiting $\beta 1$ integrin stimulates FCL and RA assembly is interesting as it indicates the counter-balance between the $\alpha V\beta 5$ and $\alpha 5\beta 1$ integrins. It is a pity that the author did not dig deeper into the mechanism underlying this interesting finding, which should greatly increase the impact of this study.

We fully agree with the reviewer that FCLs and RAs are part of the same structure. Our study brings a significant weight on this argument by showing - for the first time - a role for the clathrin counterpart on the biology of these adhesions.

As suggested by the reviewer, we looked deeper into the cross-adhesion mechanism of $\alpha V\beta 5$ and $\alpha 5\beta 1$ integrins. As we show in Figure 8 and 9, we found out that active integrin $\beta 1$ localization at fibrillar adhesions is crucial for its inhibitory activity. Knockdown of Tensin-1, that “slides” integrin $\beta 1$ from focal to fibrillar adhesions, induced abundant FCL and RA formation in FN coated dishes. Given the need for cells to move to extend fibrillar adhesions, we show that cells under lateral confinement cannot inhibit FCL and RAs even in the presence of FN. We went even further and showed that induction of FN-based cell migration is coupled to the dissolution of FCL and RAs.

Reviewer #2 (Evidence, reproducibility and clarity (Required)):

Summary:

Hakanpää and colleagues report on the relationship between flat clathrin lattices (FCLs) and reticular adhesions, with FCLs being proposed to nucleate reticular adhesions. Overall the experimental work is high quality and the data are generally well presented.

We would like to thank the reviewer for the constructive comments.

As the reviewer will notice, this revised version is a significant overhaul of our initially submitted manuscript. First, we decided to simplify our message on the role of FCLs on RA formation. We have dropped the nucleation idea and now we simply claim that FCLs are essential for the formation of RAs. This focusses the attention on our strongest results and avoids the “who-comes-first” discussion, which has proven very hard to address (see below). This on itself is an exciting finding that shows a clear function for this long known and little understood clathrin structure. Moreover, we also bring a substantial amount of data on the role of fibronectin as an inhibitor of RA formation by showing that this mechanism is operational in 6 different cell lines (all lines we tested). We also further explored the interesting inter-adhesion role of integrin $\beta 1$ activation in RA inhibition and show that this integrin (and its partner $\alpha 5$) needs to translocate from focal adhesions to fibrillar adhesions to avoid RAs to form. This activation is dependent on cell movement and as such, we show that the initiation of cell migration is coupled to the disassembly of RAs.

Major comments:

The Introduction is very brief and doesn't cover the information required to understand the paper. There are three cellular structures to be understood: focal adhesions, reticular adhesions and FCLs. The intro jumps straight into the FCLs (the paper is written from a FCL point of view) but there is no information about the other two structures really particularly the differences between them. Furthermore, there is nothing in the intro about cellular adhesion or why this is even worth studying. The authors should fully revise this Intro, there is much room for improvement!

We admit we had an FCL-centric approach to our submitted manuscript. This was done with the purpose to streamline the story into a short format. For this revised version, where our results have dived deeper into the crosstalk between $\alpha V\beta 5$ and $\beta 1$, we have extended the introduction to include an overview of integrin-based adhesions and their relationship. Thanks for the suggestion!

Fig 4D I could not understand this plot and the legend did not describe it properly. What are the units of FCL frequency? I guess it is FCLs per some distance ($10\mu\text{m}^2$), the images need a scale bar. OK, I read the description in the methods and now I see it is the proportion of total CCSs that are FCLs; so frequency is the wrong term. The legend says that there were 32 videos and there are 32 points on the plot but what we need to know is: where $n = 1$ cell, what did the FCL frequency do over time? A line is drawn on the graph, no info on what the line is and the fit is poor ($R^2 < 0.5$). The authors should take their series of data points from individual cells and fit curves to each and describe the summary statistics of the parameters of these fits OR average the data and fit to that, using the 1 SD of the data for weighting the fit. Probably more data is required to do any meaningful fitting here. To my eye it looks like FCL frequency goes from 0.3 to a plateau of 0.5 at 10 min and that more timepoints between 0 and 10 min would have been useful.

In the revised version we have changed the term “FCL frequency” by “FCL proportion”, as it better describes what we mean (the proportion of CCSs that are FCLs). We have also added a description of this metric to the results section.

As for the results of FCL proportion after *mab13* addition (previously figure 4D, now Figure 6D), we have now added more videos ($n=63$ videos). As you can appreciate, the data looks much clearer and the increase fits a logarithmic curve with an $R^2 = 0.656$.

As suggested, we looked at the possibility to use per frame FCL proportion data to build these curves. The major problem with this approach is that we decided to take a conservative approach to define CCPs and FCLs. For example, CCPs are defined as events that - unambiguously - last up to 119s. Therefore, all events that were present at the start or lasted beyond the end of the movies, cannot be surely counted as CCPs and end up inflating the FCL proportion count. This characteristic of our analysis can be seen in “Salvador Dali mustache”-shaped graph below.

Per frame FCL proportion. The graph shows how a conservative approach to define a CCP (<120s) lead to an inflated proportion of FCLs at the first and last 119 frames. Cells in these videos are untreated ($n = 16$ videos). Mean +/- SD

For this reason, we decided to get an average of the middle section of movies (frames 125 to 175) to define FCL proportion.

Nonetheless, if we use these “conservative frames” in the analysis of the *mab13* experiment, we reach a similar conclusion (similar R^2 , curve shape and biological effect), as you can see in the graph below. The reduction in FCL proportion in each condition (rather than the expected growth) is likely due to bleaching, which is unavoidable with our AP2-GFP cell line when imaged for 5 minutes.

Per frame FCL proportion after *mab13* treatment. FCL proportion growth after *mab13* treatment. Showing frames 100 to 175. first. Here we used the same data as in figure 6D of the manuscript. Mean +/- SEM

Fig 3F/G is a nice expt. It looks as though the ITGB5 signal is already creeping up when the AP2 arrives. I agree that they accelerate together, but the prior accumulation of integrin is at odds with the conclusion that AP2/clathrin *nucleates* the adhesion. This experiment is missing two controls: what is the behaviour of ITGB5 in AP2 negative regions?

We now show that ITGB5 signals which do not colocalize with a persistent AP2 (FCLs) signal fade away (Figure 5B). For some of these events, we could measure the appearance of blinking AP2 events (i.e. endocytic events), which complicates the analyses but beautifully demonstrate the double-role of the endocytic machinery on the biology of ITGB5-adhesions.

What happens to both signals in the continued presence of cilengitide? These controls are needed to conclude that AP2 is nucleating the adhesion.

In the presence of Cilengitide, most RAs and FCLs disappear. A few small, dot-like RAs, covered with FCLs remain (as shown in Figure S3A).

Based on the comments from reviewers and from a few colleagues, we have decided to adjust our nucleation claims in the manuscript. We now describe that FCLs are essential for the establishment of RAs. This removes the "who-comes-first" discussion and focus on the rather unique finding that the clathrin machinery is needed for the formation of an integrin-based adhesion.

The appearance of RAs is a random event and difficult to unambiguously detect, even when using the "optimal" conditions we have setup. As a matter of fact, we have been able to confidently detect their formation and growth only in a small fraction of movies we took. We still think that our hypothesis of FCLs as a nucleator of RAs is correct, so we have decided to rethink our approach and address this issue in higher detail in a next publication using a variety of techniques.

Minor comments:

Fig 6 - several typos - "adaptor proteins" "engagement" "containing"

Thanks for pointing this out. In this revised version, we have removed our model (figure 6). We received feedback from colleagues (a great advantage from pre-prints) and from reviewer #3 that it does not bring clarity for the paper. We have however, kept the summary drawings in individual figures, as those got very good feedback.

Reviewer #2 (Significance (Required)):

Previously, endocytosis (clathrin-mediated) was thought to decrease cellular adhesion by removal of integrins. This paper suggests that the same machinery can be used to build adhesions. This is a surprising conclusion that will be of interest to many cell biologists; the topic of clathrin and adhesion is being actively explored by several labs either from the adhesion or the endocytosis sides. I have been following this topic from a distance and don't know the details of all the published papers, but this paper does seem to add something new over the recent work from Taraska, Sonnenberg, Montagnac, Strömblad.

Thanks for the support. We also think that our work brings a new and surprising insights to the cell biology community.

Reviewer #3 (Evidence, reproducibility and clarity (Required)):

Overview:

This manuscript addresses the emerging nexus linking the machinery associated with clathrin endocytosis (clathrin-coated pits; CCPs), flat clathrin lattices (FCLs) and the recently discovered Reticular Adhesions (RAs). This is timely work, reflecting recent foci on the relationship between these structures and systems. Initially clearly identifying reductions in FCL and RA formation on fibronectin, the authors sought to clarify the mechanisms that suppress or prevent FCL / RA formation on this matrix. Knock-down of integrin avb5 (core RA component) suppressed both RA and FCL formation, suggesting a dependence of FCLs on this integrin. This was supported by acute avb5 inhibition via cilengitide (avb5 and avb3 inhibitor) which caused disassembly of existing RAs and FCLs. Notably, the inverse relationship also appears true, with suppression of core clathrin endocytic machinery (AP2 complex components) being sufficient to greatly deplete RA formation. Supporting this finding, overexpression of a dominant negative-acting protein fragment (AP180 c-terminal fragment) blocked both AP2 localisation to the plasma membrane and RA formation. To unmix this bi-directional dependence further, the authors used acute cilengitide treatment followed by washout and post-washout incubation to first deplete RAs (cilengitide) and then allow monitoring of en masse RA formation after cilengitide washout. This is an effective experiment, however, the analysis would benefit from greater depth, particularly relating to the order of events. Analysis of this aspect seems central to the thrust of the paper, and some statistical analysis of either static co-occurrence or dynamic ordering in large numbers of FCL / RA structures (i.e. hundreds) would be of value.

We would like to thank the reviewer for the very constructive comments. They have helped us to significantly improve our story.

As the reviewer will notice, this revised version is a significant overhaul of our initially submitted manuscript. First, we decided to simplify our message on the role of FCLs on RA formation. We have dropped the nucleation idea and now we simply claim that FCLs are essential for the formation of RAs. This focusses the attention on our strongest results and avoids the “who-comes-first” discussion, which has proven very hard to address (see below). This on itself is an exciting finding that shows a clear function for this long known and little understood clathrin structure. Moreover, we also bring a substantial amount of data on the role of fibronectin as an inhibitor of RA formation by showing that this mechanism is operational in 6 different cell lines (all lines we tested). We also further explored the interesting inter-adhesion role of integrin $\beta 1$ activation in RA inhibition and show that this integrin (and its partner $\alpha 5$) needs to translocate from focal adhesions to fibrillar adhesions to avoid RAs to form. This activation is dependent on cell movement and as such, we show that the initiation of cell migration is coupled to the disassembly of RAs.

We discuss your comments regarding the cilengitide experiments in question 4 below.

The authors next focused on the observation that fibronectin suppressed both FCL and RA structures, by assessing the role of fibronectin-receptor integrin $\beta 1$. Acute antibody mediated integrin $\beta 1$ inhibition (mab13) and integrin $\beta 1$ knock down both confirmed that in cells on fibronectin, suppression of integrin $\beta 1$ is sufficient to permit massive upregulation of both FCL and RA formation. This is surprising and very interesting. It raises questions about the actual ECM requirements for avb5-mediated reticular adhesion formation. It would seem that fibronectin per se can support very efficient RA / FCL formation, but that normally concurrent integrin $\beta 1$ activities would suppress this. Given this implication, it would be especially important to clarify the purity of FN ECM coating (as explored in questions 1-3 below) at the time of imaging.

We address these comments in questions 1-3 below.

The discussion addresses a number of topics, and proposes a mechanistic model to explain the results presented. I don't find the mechanism very convincing, as the directionality of the dependence between FCLs

and RAs is not clearly delineated by the experiments presented, in my opinion. That there is co-dependence is convincingly shown, but whether there is directionality, and what order of events underpins FCL then RA or RA then FCL formation, is unclear.

We have removed the model figure from the manuscript. We have received similar feedback from reviewer #2 and other colleagues that the model is confusing and adds little to the results (a shout to the power of pre-prints). As discussed above, we have now toned down our claims in order to focus on the strength of our results, that the clathrin machinery is essential for the formation of RAs.

Nonetheless, the evidence presented does generally support the idea of a shift in the way we consider the role of endocytic machinery in adhesion regulation, from a disassembly only related function to additional functions associated with adhesion formation and maintenance. Ideally, the mechanisms around this new assembly / maintenance function would be further delineated here, but regardless, this work does point in the direction of important new questions in this area.

As suggested by all reviewers, we have further explored the mechanism controlling the formation and disassembly of RAs and how it relates to cell movement, by focusing on the cross-adhesion interplay of integrin $\alpha V\beta 5$ RAs and the fibronectin receptor integrin $\alpha 5\beta 1$. As we show in Figure 8 and 9, we found out that active integrin $\beta 1$ localization at fibrillar adhesions is crucial for its inhibitory activity. Knockdown of Tensin-1, that promotes “sliding” of integrin $\beta 1$ from focal to fibrillar adhesions, induced abundant FCL and RA formation in FN coated dishes. Given the need for cells to move to extend fibrillar adhesions, we show that cells under lateral confinement cannot inhibit FCL and RAs even in the presence of FN. We went even further and showed that induction of cell migration is coupled to the dissolution of FCL and RAs.

Further discussion about the potential role of this interdependent regulatory process in, for example, mitosis, seems unwarranted and should probably be removed.

Fair point, we have removed the mention of mitosis in the manuscript.

Questions:

1) A technical question on the replating experiments onto specific matrix proteins; after coating surfaces with the purified ECM components or controls, what media were the cells replated in? Ideally this should be serum-free media, to ensure that the ECM components of FBS / FCS are not immediately added to the purified ECM components. This should be clarified in the methods.

We plated for 16-20h in 10% serum before imaging for all experiments. In our hands, using serum free media during replating lead to a large fraction of cells dying. Moreover, many of the surviving U2Os cells underwent an EMT-like process and start to secrete significantly more fibronectin. For this reason, we used complete medium throughout. We have made clearer how the experiments were done in the results section and in the methods.

The possibility of interference of serum components in our results is indeed a good question. This very same question was raised by reviewer #1 and by a few colleagues during conferences. The rest of this answer is a copy of the answer we gave to reviewer #1.

To reinforce our conclusions, we addressed this issue in two ways: (1) by adding additional pieces of evidence further supporting the role of fibronectin as an inhibitor in FCL and RA formation and (2) by directly addressing the effect of ECM components in serum. Moreover, we have also added new results and discussed the possible role of vitronectin in the process (3).

(1) Pro-fibronectin evidence

In Figures 1C,D, we show that the other two ECM coatings that - alongside fibronectin - also reduce the amount of FCLs (collagen IV or laminin 111) lead to a substantial increase in fibronectin secretion.

Using a panel of 6 cell lines (U2Os, HeLa, MCF7, HDF, Caco2 and hMEC) we show that there is a direct correlation between the amount of fibronectin endogenously secreted by these cells and the amount of FCLs and RAs. Moreover, our main cell model (U2Os) deposits very little fibronectin (despite being “bathed” in serum fibronectin). These results are now shown in Figures 1C and 3.

(2) Why fibronectin in serum does not coat dishes when cells are plated overnight?

This was indeed a puzzling question. However, we knew that, somehow, serum fibronectin was not coating our dishes, otherwise we would not have such clear results in our patterning experiments. To try to understand this apparent conflict, we compared the amount of fibronectin deposited on the glass surface in different conditions: Dishes were coated for 24 h with 10, 5 and 1 µg/ml of fibronectin (diluted in PBS), 100% fetal bovine serum (FBS), media with 10% FBS and PBS as a control. After coating, U2Os cells were plated and left to attach for 16h before being fixed and stained for FN. As additional controls, coatings without cells were also stained. Surprisingly, our results revealed that very little fibronectin was deposited on glass in dishes “coated” with full media or pure FBS (Figure S1F,G).

At this moment, I imagine one might be asking: How is this possible? Why would fibronectin in PBS deposit on dishes while fibronectin on media (or pure serum) would not? These very questions haunted us for quite some time. The answer was right in front of us, clear to see every time we prepared our cells for immunofluorescence: blocking! Bovine serum albumin is widely used as a blocking agent not only because of its cost and availability, but also because of its coating efficiency. We found out that there is a vast literature characterizing how effective BSA binds to naked and treated glass surfaces. These studies have shown that BSA readily binds to glass, within minutes of its addition (for example, see Ma et al., 2020; Sun and Zhu, 2016). Given that BSA concentrations in serum are on the range of 40mg/ml, this likely explains why fibronectin cannot deposit -passively- in significant amounts during the period of our experiment. What about vitronectin?

Vitronectin is the main ligand for integrin $\alpha\beta5$, placing it in a likely important position on the process we are studying. However, we (Figure S1D) and others (Zuidema et al., 2022) have shown that vitronectin and integrin $\alpha\beta5$ colocalize equally on focal adhesions and RAs, pointing to the fact that the cellular decision to use these integrins on RAs or FAs probably occurs inside-out. This point is discussed in the section “the mechanism of FCL-mediated RA formation” of the discussion. We also added to this discussion section the fact that vitronectin is known to be readily secreted by most cells during adhesion, and caution should be taken when comparing the data on non-coated and vitronectin-coated dishes.

2) Related to above, I cannot see how long cells were plated onto the different ECM conditions. This would be relevant to know and should be clarified because cells will secrete ECM over time and thus the purity of the ECM components addressed is dependent on the length of time cells are incubated and imaged for after attachment.

Cells were plated for 16-20h before imaging for all experiments. We have now made this clear in the methods and the start of the results section.

Growing cells for an extended time would indeed make any meaningful comparison of ECM identity difficult. We indeed noticed a very high variability of effects when cells were left on the dishes for more than 24 h.

Crucially, our main point of the manuscript regarding ECM components is to show that FN is the main inhibitor of FCL/RAs. The experimental conditions we use perfectly allowed us to prove our point regarding this specific property of FN. Various pieces of data in the revised manuscript (Figure 1C, 2A and 3A-E), bring support this FN property.

3) Similarly, it is noted that cells plated on 'glass' support RA formation. It should be clarified what ECM component is then actually responsible for cell adhesion and adhesion complex (RA, FA or other) formation - since this requires an ECM component of some type. Presumably, 'on glass' means whatever ECM is either derived from the media used during cell attachment / incubation (if that media contains serum, which is vitronectin rich), or whatever ECM is secreted by the cells themselves over the attachment / incubation period prior to imaging.

*Thanks for pointing this out. This was not supposed to be the message we wanted to convey. We have made many changes to the manuscript to clarify this. First, to reflect the fact that under our plating conditions cells would have time to deposit some ECM components on their own, we have changed the term "glass" to "non-coated". Second, we changed the parts where we say that "glass supports RA conditions" to "Cells were plated on non-coated dishes, a condition where we observe large RAs". At the end of the day, **we want to emphasize that the presence of sufficient amounts of FN inhibits FCLS and RAs**, not that glass - or any other ECM component - promotes the formation of these structures.*

We could not address in detail the mechanism of RA formation. As we discuss below (in question 4), the tools we currently have are not good enough to unambiguously detect these events in large enough numbers that would allow us to make manipulations and draw meaningful conclusions.

As discussed above (point 3, question 1), we discuss in the manuscript the possible influence of other ECM components (especially vitronectin) in our experiments and its possible role in the biology of RAs.

4) In the cilengitide washout experiments, the evidence shown in figures seems to suggest that AP2-positive FCLs form in locations where avb5 (probably RAs) are already present, whereas avb5 positive structures do not form from AP2-only structures. Statistical analysis of this pattern (i.e. which protein is present first) would be valuable to address directionality. Notably, in Figure 3G, it appears that avb5 is present and increasing prior to the subsequent arrival of AP2.

RA appearance is a rare and difficult to unambiguously detect them. Even under our optimized experimental conditions. As a matter of fact, we have been able to confidently detect their formation and growth only in a small fraction of movies we took. Following the suggestion of the reviewer (written in your summary), we have used cilengitide washout experiments and tried to use static co-occurrence to add statistical power to our claims. Given the variability of RA formation in terms of timing and cell-to-cell variation, we could not get any meaningful results. To bring forth the idea of FCL as a bona fide nucleator of FCLs, we realized that we need to rethink our approach and use different techniques/model system to support this idea. This effort, in our view, is a whole project of its own.

For this reason, we have decided to tone down our claims in this revised version and say that FCLs are essential for the establishment of RAs, which the bulk of our data strongly supports. To add support for this idea, we have analyzed the fate of ITGB5 signals not colocalizing to FCLs in our live-cell imaging experiments and could show that they disappear. For some of these events, we could measure the appearance of blinking AP2 events (i.e. endocytic events), which beautifully demonstrate the double-role of the endocytic machinery on the biology of ITGB5-adhesions (Figure 5 and S3).

4a. I would suggest that the cilengitide experimental results (3E-G) be shown in a separate figure from the endocytosis inhibition results (3A-D).

Following your suggestion, the former figure 3 is now presented as figures 4 and 5.

5) The integrin b1 inhibition and knock down results are clear and interesting. Clarifying the ECM components present during these experiments would be valuable to interpretation of the paper.

Discussed in questions 1, 2 and 3 above.

Reviewer #3 (Significance (Required)):

see above

References:

Ma, G. J., Ferhan, A. R., Jackman, J. A. and Cho, N. J. (2020). *Conformational flexibility of fatty acid-free bovine serum albumin proteins enables superior antifouling coatings. Communications Materials 2020 1:1 1, 1–11.*

Sun, Y. S. and Zhu, X. (2016). *Characterization of Bovine Serum Albumin Blocking Efficiency on Epoxy-Functionalized Substrates for Microarray Applications. J Lab Autom 21, 625–631.*

Zuidema, A., Wang, W., Kreft, M., Bleijerveld, O. B., Hoekman, L., Aretz, J., Böttcher, R. T., Fässler, R. and Sonnenberg, A. (2022). *Molecular determinants of α V β 5 localization in flat clathrin lattices - role of α V β 5 in cell adhesion and proliferation. J Cell Sci 135,.*

April 24, 2023

Re: JCB manuscript #202303107T

Dr. Leonardo Almeida-Souza
University of Helsinki
HiLIFE Institute of Biotechnology
Viikinkaari 5
Helsinki 00790
Finland

Dear Dr. Almeida-Souza,

Thank you for submitting your revised manuscript entitled "Reticular adhesions are assembled by flat clathrin lattices and opposed by fibrillar adhesions" from Review Commons. The manuscript has been seen by the original reviewers whose full comments are appended below. While the reviewers continue to be overall positive about the work, some important issues remain.

While we do not require further experiments to test the regulatory role of mechanical forces, the fibronectin experiments suggested by reviewer #3 should be included if you already have the data. In addition, please address all other reviewer points with appropriate edits.

Our general policy is that papers are considered through only one revision cycle; however, given that the suggested changes are relatively minor we are open to one additional short round of revision. Please note that I will expect to make a final decision without additional reviewer input upon resubmission.

Please submit the final revision within one month, along with a cover letter that includes a point by point response to the remaining reviewer comments.

Thank you for this interesting contribution to Journal of Cell Biology. You can contact me or the scientific editor listed below at the journal office with any questions, cellbio@rockefeller.edu or call (212) 327-8588.

Sincerely,

Martin Humphries
Monitoring Editor

Andrea L. Marat
Senior Scientific Editor

Journal of Cell Biology

Reviewer #1 (Comments to the Authors (Required)):

This revised version of the manuscript by Hakanpää et al has been significantly improved. The authors have clarified some of my major concerns including the coating conditions and the relationships among FCLs, RAs and CME. I also appreciate that the authors considered my advices and have now added more ligands and controls into their assays.

Major point: although it is interesting that depletion of tensin 1 induce RA formation, I do not recommend to put fibrillar adhesion in the title. It is more important to point out beta1 integrin activation inhibits RA formation as this is the major finding and topic of the manuscript. I suggest to change title to "Reticular adhesions are assembled by flat clatherin lattices and opposed by activation of $\alpha 5 \beta 1$ integrin".

Some minor points

1. In Figure, S5C, which band represents tensin1?
2. In Figure 9D right panel, there is a typos "migratioon".
3. Are tensin3 and talin2 expressed in U2OS cells? Many cells express more than one tensin and one talin, it would be more convincing to show that double knockdowns of tensin1&3 or talin 1&2 result in more obvious phenotypes.
4. Although the kindlin knockdown data was not included in the revision, I disagree with the statement that "kindlins serve as

important accessories of talin-mediated integrin activation at FAs". Kindlins are essential integrin activators, they bind the integrin tail at a different region compared to talin.

Reviewer #2 (Comments to the Authors (Required)):

I have read the revised manuscript which is a significant improvement than the version I previously reviewed at RC (I was Reviewer #2 if that is not clear). The authors have responded to all my comments in full. I appreciate the effort that the authors have put into revising and refocussing this paper.

There's a typo "Migration" in Fig 9D.

Reviewer #3 (Comments to the Authors (Required)):

Overall I think the manuscript is significantly improved, and that it contains important insights that will be of value to the field, i.e. to direct further investigation of the links between FCLs and RAs, and the coupling between adhesion and endocytic machinery. The paper points in a number of directions, and though it does not provide strong mechanistic insights, in my opinion, the gaps that it leaves are clearly delineating important areas for the field.

Despite this overall positive impression, a number of areas still need improvement, in my opinion, prior to publication. These are detailed below.

1) Whilst the authors state that they have reduced / removed their focus on the ordering of the FCL / RA formation relationship, the 3rd sentence of the abstract states that 'FCLs assemble RAs', which has a clear implication of directionality. This and similar statements should be amended, e.g. FCLs and stable RAs coincide (since unstable RAs are shown to exist without FCLs) or similar.

2) Some of the methods defined under 'Other Analyses' require additional information. Instances are detailed below. Generally speaking, some visual examples of the segmentation and colocalisation / non-colocalisation definitions should ideally be shown - i.e. segmentation boundaries around objects (FCLs, RAs etc), showing which are defined into which category according to which markers. At the moment, we simply have to trust that the segmentation and analysis processes are reasonable, which is not ideal.

a. For the analysis of AP2-ITGb5 dynamics, it seems potentially problematic that analysis is defined (initiated) by the arrival of AP2. Since it is still stated throughout that FCLs are required for the 'establishment of RAs', using a definition that starts with the arrival of FCLs seems to allow no other alternative (i.e. that RAs could be established without FCLs - since these would be invisible to an analysis predicated on AP2 arrival as the defining start point). In my opinion, an additional analysis using ITGb5 arrival as an initiating feature would be valuable. It is not clear to me how these definitions relate to the data in Fig 5B. This should be clarified.

b. The definition of 'colocalisation' is critical to interpretation of many results in this paper, and is not sufficiently defined here. For example, in the "RAs colocalising to (should be with) AP2", it is stated that "RAs were defined at ITGb5 signals not colocalizing with p-Pax". What was the quantitative definition of colocalizing? A percentage of coverage? An intensity value? Any signal above a threshold? This needs to be clarified in each relevant setting.

3) Data, especially Fig 5A, still appears to support the possibility that ITGb5 is (or can be) present first during formation, whilst the kinetic analysis in Figure 5B, right panel, suggests that the gradual loss of ITGb5 may progress in a way that is not clearly (at least in the figure) correlated with AP2 kinetics.

4) Regarding the FCL proportion questions from reviewer 2; Figure 6D still looks to be labelled as Relative FCL frequency (Y-axis). If this were changed to proportion, might it make more sense to have the proportion normalised to a maximum of 1 (or 100%) - this indicating how the real proportion of CCP vs FCLs changes, rather than the relative proportion?

5) In Figure 2C-E, and as considered in results around line 140, it is shown that FN locally suppresses RAs. In C, right upper panel, it appears that there may be avb5-positive FAs present, at similar frequency (low) to that seen on uncoated surfaces (Fig 2A, p-Pax, no-coating). It would be interesting to know if FN actually causes a shift in the balance between avb5-positive RAs and FAs, rather than simply suppressing RAs? This looks to be supported by the avb5 + p-Pax distributions in high FN environments (Fig 2A). Similarly in Fig 3C, high FN production correlates not only with suppression of RAs, but with increased FAs. Similarly in Fig 4A, C. Given these data, it would be ideal to see a similar measure to the FCL vs CCP proportion metric; using p-Pax to and avb5 (in fixed cells, you likely already have this data) to define the proportions of RAs and FAs in the presence / absence etc of FN, AP2 etc. This may add significantly to interpretation of the findings, in my opinion, since there looks to be a shift in preferred adhesion type, not just suppression of RAs (similar to the equilibrium described in Lock et al for PIP regulation, which likely impacts these same processes).

6) Fig S2 and Line 187: "the dependency of integrin b5 on FCL formation was further confirmed using Celengitide..." This may be a typo, and should possibly read "the dependency between integrin b6 and FCL formation ..." -> certainly I don't see how the loss of RAs after ITGb5 KD, or Cilengitide, can be used to argue that integrin b5 depends on FCLs. Only that FCLs depend on ITGb5 / avb5. Bidirectionality in this relationship is subsequently supported in Fig4, but this is after the statement at Line 187.

7) The language used around line 220 is by far the best and clearest exposition of the findings so far, but needs to say 'and vice versa', i.e. that ITGb5 also stabilises FCLs (FigS2). There still appears to be no merit in emphasising the role of FCLs in regulating RAs without also highlighting the reverse dependence.

- 8) Overall, as discussed briefly (Line 369 - 376) the results of Figure 8 and 9 speak to an effect of increased mechanical forces on the balance between FAs and RAs (i.e. RA suppression / FA enhancement). In my opinion, this paper is at risk of misattributing the cause of the observed effects, and would be improved by addressing the regulatory effects of mechanical forces more directly, especially since the results of Fig 9 essentially undercut the earlier evidence that FN directly mediates RA inhibition. Rather, FN seems to promote (in un-confined cells) adhesion types (FAs, FBs) compatible with force generation, which promotes both migration and RAs suppression, as reported by Zuidema et al.
- 9) Again, in the discussion, the authors (lines 310, 320, 321 and Fig 9G) use the terminology 'FCL-mediated avb5 RA formation' - but this is not a fair reflection of the data in my opinion -> RAs are similarly important to FCLs, or in other words, they are demonstrated to be co-dependent. The consistent bias in language would likely give the reader the wrong impression, especially since the data for FCL dependence on RAs is in the supplement, while the data for RA dependence on FCLs is in the main body.
- 10) Consistently throughout the paper, the authors refer to sliding of fibrillar adhesions on the membrane. It should be noted somewhere that this sliding reflects integrin (or adhesion component) assembly, turnover, disassembly, asymmetrically with respect to actin-derived mechanical forces. i.e. the adhesions are not sliding, they just appear to be.

Reviewer #1 (Comments to the Authors (Required)):

This revised version of the manuscript by Hakanpää et al has been significantly improved. The authors have clarified some of my major concerns including the coating conditions and the relationships among FCLs, RAs and CME. I also appreciate that the authors considered my advices and have now added more ligands and controls into their assays.

We would like to thank the reviewer for their support and fair comments, which proved instrumental in improving the quality of our manuscript throughout the review process.

Major point: although it is interesting that depletion of tensin 1 induce RA formation, I do not recommend to put fibrillar adhesion in the title. It is more important to point out beta1 integrin activation inhibits RA formation as this is the major finding and topic of the manuscript. I suggest to change title to "Reticular adhesions are assembled by flat clathrin lattices and opposed by activation of $\alpha 5\beta 1$ integrin".

Fair point. We have now changed the title to: "Reticular adhesions are assembled at flat clathrin lattices and opposed by active $\alpha 5\beta 1$ integrin".

Some minor points

1. In Figure, S5C, which band represents tensin1?

It is the lower band. We have added an arrow pointing to the correct band

2. In Figure 9D right panel, there is a typos "migration".

Corrected.

3. Are tensin3 and talin2 expressed in U2OS cells? Many cells express more than one tensin and one talin, it would be more convincing to show that double knockdowns of tensin1&3 or talin 1&2 result in more obvious phenotypes.

Yes, both Tensin3 and Talin2 are expressed in U2Os. mRNA expression data (from Human Protein Atlas), shows that Tensin1 and Tensin3 are expressed at similar levels while Talin1 is expressed 2.5 times more than Talin2.

We agree that double knockdowns could potentially result in stronger phenotypes. Having said that, the tensin1 KD phenotype was very clear, despite our shRNAs achieving only an incomplete knockdown. With Talin, we faced many issues while doing these experiments. To ensure the ECM effect we see is (mostly) a result of the coating and not the ECM deposited by cells, we always plated cells 16 h before fixation. This means we had to plate our cells a few days after the lentiviral transfection, when the knockdown is already happening. As expected, cells lacking these key adhesion components are not very keen to adhere to dishes after splitting. As a matter of fact, it was clear for us that the adherent-capable cells after lentiviral treatment were mostly weak knockdowns. When we plated cells on FN 48 or 72h before fixation, we ended up with a higher base line (i.e. more cells with RAs), making any gains from extra knockdown harder to detect. Double knockdowns would likely potentiate the issues we had and possibly achieve little gain. We think that the only way around these issues is to use acute degradation, such as the auxin degrading system.

4. Although the kindlin knockdown data was not included in the revision, I disagree with the statement that "kindlins serve as important accessories of talin-mediated integrin activation at FAs". Kindlins are essential integrin activators, they bind the integrin tail at a different region compared to talin.

That's a fair point. With the new paper from Pernier et al. (JCS 2023) the community should stop calling kindlin an accessory protein. It won't happen again! ☺

Reviewer #2 (Comments to the Authors (Required)):

I have read the revised manuscript which is a significant improvement than the version I previously reviewed at RC (I was Reviewer #2 if that is not clear). The authors have responded to all my comments in full. I appreciate the effort that the authors have put into revising and refocussing this paper.

Thank you for your kind words during the review process. We appreciate your feedback and are grateful for your time and effort in reviewing it.

There's a typo "Migration" in Fig 9D.

Corrected.

Reviewer #3 (Comments to the Authors (Required)):

Overall I think the manuscript is significantly improved, and that it contains important insights that will be of value to the field, i.e. to direct further investigation of the links between FCLs and RAs, and the coupling between adhesion and endocytic machinery. The paper points in a number of directions, and though it does not provide strong mechanistic insights, in my opinion, the gaps that it leaves are clearly delineating important areas for the field.

We appreciate the support and thoroughness of the reviewer. We are genuinely grateful for the comments and suggestions, which helped us to streamline our message and avoid potential misrepresentations.

Despite this overall positive impression, a number of areas still need improvement, in my opinion, prior to publication. These are detailed below.

1) Whilst the authors state that they have reduced / removed their focus on the ordering of the FCL / RA formation relationship, the 3rd sentence of the abstract states that 'FCLs assemble RAs', which has a clear implication of directionality. This and similar statements should be amended, e.g. FCLs and stable RAs coincide (since unstable RAs are shown to exist without FCLs) or similar.

Fair point. We have changed the sentence in the abstract to "RAs are assembled at FCLs". We have also changed to title to "Reticular adhesions are assembled at flat clathrin lattices". Other changes in the text to ensure we do not put too much emphasis on directionality were also made (See below).

2) Some of the methods defined under 'Other Analyses' require additional information. Instances are detailed below. Generally speaking, some visual examples of the segmentation and colocalisation / non-colocalisation definitions should ideally be shown - i.e. segmentation boundaries around objects (FCLs, RAs etc), showing which are defined into which category according to which markers. At the moment, we simply have to trust that the segmentation and analysis processes are reasonable, which is not ideal.

a. For the analysis of AP2-ITGb5 dynamics, it seems potentially problematic that analysis is defined (initiated) by the arrival of AP2. Since it is still stated throughout that FCLs are required for the 'establishment of RAs', using a definition that starts with the arrival of FCLs seems to allow no other alternative (i.e. that RAs could be established without FCLs - since these would be invisible to an analysis predicated on AP2 arrival as the defining start point). In my opinion, an additional analysis using ITGb5 arrival as an initiating feature would be valuable. It is not clear to me how these definitions relate to the data in Fig 5B. This should be clarified.

The graph in figure 5B (right) is exactly what you are asked. These are events where we simply followed appearing ITGB5 signals. In this graph, we use the disappearance of ITGB5 as time zero. We did not use AP2 as a reference. We now realized that we did not explain this appropriately. We sincerely apologize for it. We have now added clearer explanations for how these graphs were generated in the figure legends and in the methods section.

As the reviewer will see, events which were not stabilized by FCLs are destined to disassemble early. The small AP2 signal peak we detected during the disassembly appeared “without intervention” as we did not look at the AP2 channel while selecting small ITGB5 clusters.

b. The definition of 'colocalisation' is critical to interpretation of many results in this paper, and is not sufficiently defined here. For example, in the "RAs colocalising to (should be with) AP2", it is stated that "RAs were defined at ITGb5 signals not colocalizing with p-Pax". What was the quantitative definition of colocalizing? A percentage of coverage? An intensity value? Any signal above a threshold? This needs to be clarified in each relevant setting.

All the colocalization analyses in the paper were performed using binary masks generated using the RATS method, which we found to perform the best to segment ITGB5/p-PAX signals. We have now clarified in the methods how we defined colocalisation and have added an additional figure (Figure S6) illustrating how our segmentation works and how we calculate RA coverage.

Instances saying “colocalise to” were switched to “colocalise with”.

3) Data, especially Fig 5A, still appears to support the possibility that ITGb5 is (or can be) present first during formation, whilst the kinetic analysis in Figure 5B, right panel, suggests that the gradual loss of ITGb5 may progress in a way that is not clearly (at least in the figure) correlated with AP2 kinetics.

As explained above (2a), we realized we were not clear in the explanation of our results and in our methods. The results in the figure 5B (right) were not selected based on AP2 signal. We are very sorry for the confusion.

We agree that the appearance of ITGB5 clusters at the membrane are sometimes independent of the FCLs. As you will see, we have now avoided all citations of directionality in the manuscript. The interesting question on this matter is whether a small, free ITGB5 cluster can be considered an RA. In my view, the arrival of the FCL allows the ITGB5 cluster to become an RA. However, I admit that this discussion is not so relevant at the moment. Hopefully in a near future, when we crack the mechanism on how and why FCLs stabilise ITGB5 clusters, we may revisit this concept.

4) Regarding the FCL proportion questions from reviewer 2; Figure 6D still looks to be labelled as Relative FCL frequency (Y-axis). If this were changed to proportion, might it make more sense to have the proportion normalised to a maximum of 1 (or 100%) - this indicating how the real proportion of CCP vs FCLs changes, rather than the relative proportion?

The data is now presented as suggested.

5) In Figure 2C-E, and as considered in results around line 140, it is shown that FN locally suppresses RAs. In C, right upper panel, it appears that there may be avb5-positive FAs present, at similar frequency (low) to that seen on uncoated surfaces (Fig 2A, p-Pax, no-coating). It would be interesting to know if FN actually causes a shift in the balance between avb5-positive RAs and FAs, rather than simply suppressing RAs? This looks to be supported by the avb5 + p-Pax distributions in high FN environments (Fig 2A). Similarly in Fig 3C, high FN production correlates not only with suppression of RAs, but with increased FAs. Similarly in Fig 4A, C. Given these data, it would be ideal to see a similar measure to the FCL vs CCP proportion metric; using p-Pax to and avb5 (in fixed cells, you likely already have this data) to define the proportions of RAs and FAs in the presence / absence etc of FN, AP2 etc. This may add significantly to interpretation of the findings, in my opinion, since there looks to be a shift in preferred adhesion type, not just suppression of RAs (similar to the equilibrium described in Lock et al for PIP regulation, which likely impacts these same processes).

This an interesting idea. Thanks for the suggestion. With a few changes to the ImageJ scripts and re-analysing the data we already collected, we could easily calculate what is the “Focal Adhesion coverage” of our cells. For these analyses, we used the data from the experiments using U2Os, Hela and MCF7 cells plated in fibronectin and uncoated dishes, as these cell lines respond better to FN coating. These results are now presented in Figure S2A. The results reveal no effect of RA coverage on FA coverage. We went back to various experiments where we had U2Os cells on FN and uncoated dishes and the answer was the practically the same. One thing we did notice (from the start of this project) is that FAs in uncoated vs. coated dishes look very different. FAs in FN coated dishes is more “organised” in typical FA structures. These differences are, however, most likely due to the ECM itself.

In the experiment with AP2 knockdown or inhibition (AP180ct expression), there are indeed more FAs. This is, however, very likely due to the lack of integrin recycling caused by CME inhibition, which has been shown before to increase the amount of FAs (Ezratty, JCB 2009; Chao & Kunz, FEBS letters 2009).

6) Fig S2 and Line 187: “the dependency of integrin b5 on FCL formation was further confirmed using Cilengitide...” This may be a type, and should possibly read “the dependency between integrin b6 and FCL formation ...” -> certainly I don't see how the loss of RAs after ITGb5 KD, or Cilengitide, can be used to argue that integrin b5 depends on FCLs. Only that FCLs depend on ITGb5 / avb5. Bidirectionality in this relationship is subsequently supported in Fig4, but this is after the statement at Line 187.

Indeed, this sentence is misrepresenting the experiment. Thanks for pointing this out. We rephrased it to “The requirement of integrin β 5 for FCL formation was further confirmed using Cilengitide”.

7) The language used around line 220 is by far the best and clearest exposition of the findings so far, but needs to say 'and vice versa', i.e. that ITGb5 also stabilises FCLs (FigS2). There still appears to be no merit in emphasising the role of FCLs in regulating RAs without also highlighting the reverse dependence.

We have now added the “vice-versa” and have changed all other instances in the manuscript where we imply a directionality.

8) Overall, as discussed briefly (Line 369 - 376) the results of Figure 8 and 9 speak to an effect of increased mechanical forces on the balance between FAs and RAs (i.e. RA suppression / FA enhancement). In my opinion, this paper is at risk of misattributing the cause of the observed effects, and would be improved by addressing the regulatory effects of mechanical forces more directly, especially since the results of Fig 9 essentially undercut the earlier evidence that FN directly mediates RA inhibition. Rather, FN seems to promote (in un-confined cells) adhesion types (FAs, FBs) compatible with force generation, which promotes both migration and RAs suppression, as reported by Zuidema et al.

We agree that the regulation of mechanical forces is the-way-to-go to further understand this process. For this exact reason, one would need to address it properly. Our manuscript is already rather large and we prefer to tackle this issue in a separate project.

We do not think we are misattributing the effect of FN as we believe we have demonstrated the role of this ECM protein very convincingly. The fact that the action of FN requires a secondary step (i.e. mechanical forces) does not preclude the importance of FN kicking off the process. One question that remains is whether mechanical forces alone can bypass FN. With the current literature data, we cannot answer this question as, for example, the RhoA mutant used by Zuidema (2018) that increase contractility and abolish RAs does also lead to increased FN secretion (Danen, JCB 2002).

There is no doubt our results in Figure 8 and 9 nicely complements the findings from Zuidema et al. We have further addressed the connecting points in lines 410-431 of the discussion.

9) Again, in the discussion, the authors (lines 310, 320, 321 and Fig 9G) use the terminology 'FCL-mediated avb5 RA formation' - but this is not a fair reflection of the data in my opinion -> RAs are similarly important to FCLs, or in other words, they are demonstrated to be co-dependent. The consistent bias in language would likely give the reader the wrong impression, especially since the data for FCL dependence on RAs is in the supplement, while the data for RA dependence on FCLs is in the main body.

Fair point. We acknowledge the co-dependence of the process. We have changed all instances of the paper of "FCL-mediated RA formation" to either just "RA formation" or "RA formation by FCL and avB5 co-assembly".

10) Consistently throughout the paper, the authors refer to sliding of fibrillar adhesions on the membrane. It should be noted somewhere that this sliding reflects integrin (or adhesion component) assembly, turnover, disassembly, asymmetrically with respect to actin-derived mechanical forces. i.e. the adhesions are not sliding, they just appear to be.

Ok, we have now rephrased all instances of sliding by "form" or "appear".

May 5, 2023

RE: JCB Manuscript #202303107R

Dr. Leonardo Almeida-Souza
University of Helsinki
HiLIFE Institute of Biotechnology
Viikinkaari 5
Helsinki 00790
Finland

Dear Dr. Almeida-Souza:

Thank you for submitting your revised manuscript entitled "Reticular adhesions are assembled at flat clathrin lattices and opposed by active integrin $\alpha 5\beta 1$ ". We would be happy to publish your paper in JCB pending final revisions necessary to meet our formatting guidelines (see details below).

A. MANUSCRIPT ORGANIZATION AND FORMATTING:

- 1) Text limits: Character count for Articles is < 40,000, not including spaces. Count includes abstract, introduction, results, discussion, and acknowledgments. Count does not include title page, figure legends, materials and methods, references, tables, or supplemental legends.
- 2) Figures limits: Articles may have up to 10 main text figures.
- 3) Figure formatting: Scale bars must be present on all microscopy images, including inset magnifications (e.g. S2). Molecular weight or nucleic acid size markers must be included on all gel electrophoresis.
- 4) Statistical analysis: Error bars on graphic representations of numerical data must be clearly described in the figure legend. The number of independent data points (n) represented in a graph must be indicated in the legend. Statistical methods should be explained in full in the materials and methods. For figures presenting pooled data the statistical measure should be defined in the figure legends. Please also be sure to indicate the statistical tests used in each of your experiments (either in the figure legend itself or in a separate methods section) as well as the parameters of the test (for example, if you ran a t-test, please indicate if it was one- or two-sided, etc.). Also, if you used parametric tests, please indicate if the data distribution was tested for normality (and if so, how). If not, you must state something to the effect that "Data distribution was assumed to be normal but this was not formally tested."
- 5) Abstract and title: The abstract should be no longer than 160 words and should communicate the significance of the paper for a general audience. The title should be less than 100 characters including spaces. Make the title concise but accessible to a general readership.
- 6) Materials and methods: Should be comprehensive and not simply reference a previous publication for details on how an experiment was performed. Please provide full descriptions in the text for readers who may not have access to referenced manuscripts.
- 7) Please be sure to provide the sequences for all of your primers/oligos and RNAi constructs in the materials and methods. You must also indicate in the methods the source, species, and catalog numbers (where appropriate) for all of your antibodies. Please also indicate the acquisition and quantification methods for immunoblotting/western blots.
- 8) Microscope image acquisition: The following information must be provided about the acquisition and processing of images:
 - a. Make and model of microscope
 - b. Type, magnification, and numerical aperture of the objective lenses
 - c. Temperature
 - d. Imaging medium
 - e. Fluorochromes
 - f. Camera make and model

g. Acquisition software

h. Any software used for image processing subsequent to data acquisition. Please include details and types of operations involved (e.g., type of deconvolution, 3D reconstitutions, surface or volume rendering, gamma adjustments, etc.).

10) Supplemental materials: There are strict limits on the allowable amount of supplemental data. Articles may have up to 5 supplemental figures. Please also note that tables, like figures, should be provided as individual, editable files. A summary of all supplemental material should appear at the end of the Materials and methods section.

13) ORCID IDs: ORCID IDs are unique identifiers allowing researchers to create a record of their various scholarly contributions in a single place. At resubmission of your final files, please consider providing an ORCID ID for as many contributing authors as possible.

Please note that JCB now requires authors to submit Source Data used to generate figures containing gels and Western blots with all revised manuscripts. This Source Data consists of fully uncropped and unprocessed images for each gel/blot displayed in the main and supplemental figures. Since your paper includes cropped gel and/or blot images, please be sure to provide one Source Data file for each figure that contains gels and/or blots along with your revised manuscript files. File names for Source Data figures should be alphanumeric without any spaces or special characters (i.e., SourceDataF#, where F# refers to the associated main figure number or SourceDataFS# for those associated with Supplementary figures). The lanes of the gels/blots should be labeled as they are in the associated figure, the place where cropping was applied should be marked (with a box), and molecular weight/size standards should be labeled wherever possible.

Journal of Cell Biology now requires a data availability statement for all research article submissions. These statements will be published in the article directly above the Acknowledgments. The statement should address all data underlying the research presented in the manuscript. Please visit the JCB instructions for authors for guidelines and examples of statements at (<https://rupress.org/jcb/pages/editorial-policies#data-availability-statement>).

B. FINAL FILES:

**It is JCB policy that if requested, original data images must be made available to the editors. Failure to provide original images upon request will result in unavoidable delays in publication. Please ensure that you have access to all original data images prior

to final submission.**

Thank you for this interesting contribution, we look forward to publishing your paper in Journal of Cell Biology.

Sincerely,

Martin Humphries
Monitoring Editor

Andrea L. Marat
Senior Scientific Editor

Journal of Cell Biology